# LanPaint: Training-Free Diffusion Inpainting with Asymptotically Exact and Fast Conditional Sampling

**Candi ZHENG**[†*]                                                                *czhengac@connect.ust.hk*
*Department of Mathematics, Hong Kong University of Science and Technology*

**Yuan LAN**[†]                                                                        *ylanaa@connect.ust.hk*
*Independent Researcher*

**Yang Wang**                                                                      *yangwang@ust.hk*
*Department of Mathematics, Hong Kong University of Science and Technology*

**Reviewed on OpenReview:** *https://openreview.net/forum?id=JPC8JyOUSW*

[†]Equal Contribution. *Corresponding author.

## Abstract

Diffusion models excel at joint pixel sampling for image generation but lack efficient training-free methods for partial conditional sampling (e.g., inpainting with known pixels). Prior works typically formulate this as an intractable inverse problem, relying on coarse variational approximations, heuristic losses requiring expensive backpropagation, or slow stochastic sampling. These limitations preclude (1) accurate distributional matching in inpainting results, (2) efficient inference modes without gradients, and (3) compatibility with fast ODE-based samplers. To address these limitations, we propose **LanPaint**: a training-free, asymptotically exact partial conditional sampling method for ODE-based and rectified-flow diffusion models. By leveraging carefully designed Langevin dynamics, LanPaint enables fast, backpropagation-free Monte Carlo sampling. Experiments demonstrate that our approach achieves superior performance with precise partial conditioning and visually coherent inpainting across diverse tasks. Code is available on https://github.com/scraed/LanPaint.

## 1 Introduction

Denoising Diffusion Probabilistic Models (DDPMs) (Sohl-Dickstein et al., 2015; Song & Ermon, 2019a; Song et al., 2020c; Ho et al., 2020; Rombach et al., 2021; Betker et al., 2023) have emerged as powerful generative frameworks that produce high-quality outputs through iterative denoising. Subsequent advances in ODE-based deterministic samplers (Karras et al., 2022; Lu et al., 2022; Zhao et al., 2023), as well as equivalent rectified flow models (Lipman et al., 2022; Liu et al., 2022b; Gao et al., 2025) have dramatically improved the' efficiency of DDPMs, reducing the sampling steps from hundreds to dozens. These innovations, combined with model variants trained in the community, have broadened the scope and quality of the generative visual arts.

While diffusion models excel at whole-image sampling, their global denoising mechanism inherently limits partial conditional sampling given partially known pixels. The key question is

*Given $p(\mathbf{x}, \mathbf{y})$, how to sample from $p(\mathbf{x}|\mathbf{y})$?*

More rigorously, for a pretrained model $p(\mathbf{z})$ with arbitrary splitting $\mathbf{z} = (\mathbf{x}, \mathbf{y})$, sampling $\mathbf{x} \sim p(\mathbf{x}|\mathbf{y})$ in a **training-free** way remains a fundamental challenge for diffusion models. Current approaches fall into two categories: (1) Sequential Monte Carlo (SMC) methods (Trippe et al., 2022; Wu et al., 2024). They depend on stochastic DDPM sampling, making them incompatible with deterministic ODE samplers and

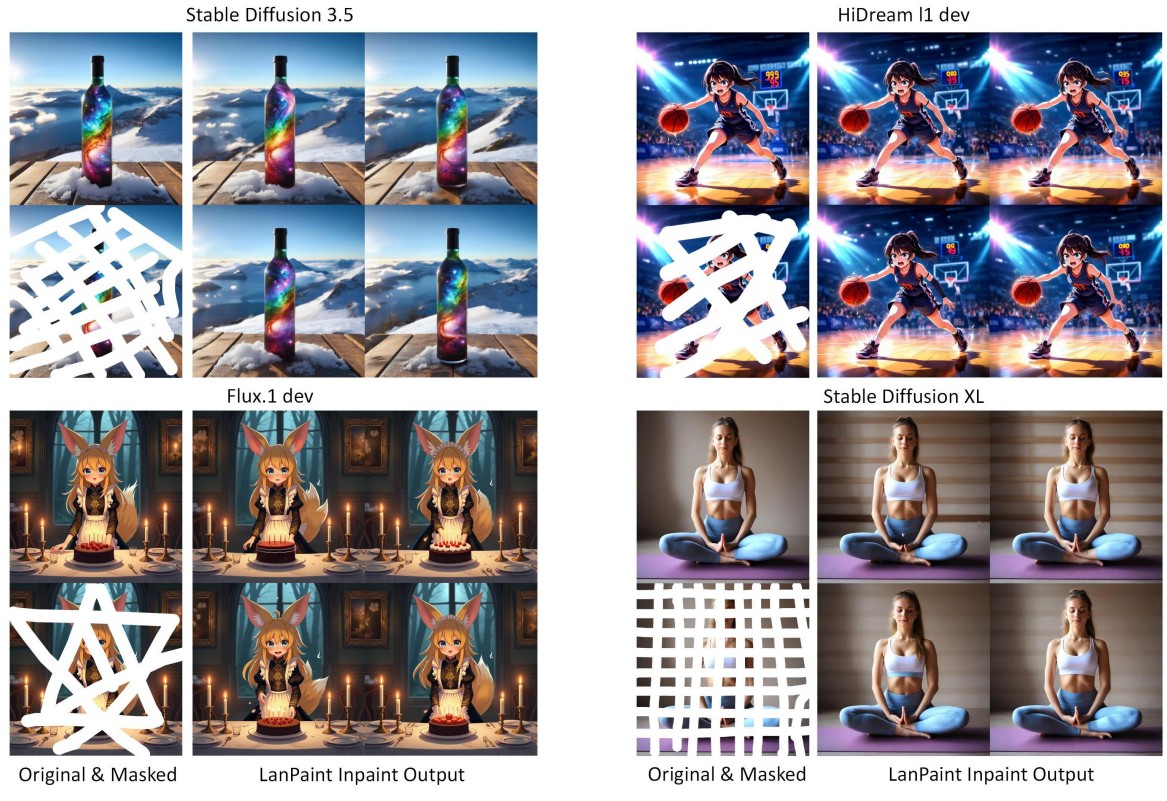

Figure 1: Demonstration of LanPaint-5 (5 inner iterations) inpainting results on HiDream-L1 (HiDream.ai, 2025), Flux.1 dev (Labs, 2024), SD 3.5 (Esser et al., 2024) and XL (Podell et al., 2023). Images are generated through ComfyUI (Comfy Org, 2025) with Euler sampler (Karras et al., 2022) (30 steps). All samples generated from a fixed seed (seed=0) producing a batch of 4 distinct random latents to avoid cherry-picking. These results demonstrate LanPaint's practical effectiveness across modern diffusion architectures, including both rectified flow (HiDream, Flux, SD 3.5) and denoising (SD XL) models.

computationally expensive; and (2) Langevin Dynamics Monte Carlo (LMC) methods (Lugmayr et al., 2022; Cornwall et al., 2024). They can be treated as iterative denoising and renoising, making them compatible with ODE samplers. But they suffer from convergence issues (Cornwall et al., 2024) and local maxima trapping - a key limitation we analyze in this work.

Another line of work formulates inpainting as a linear inverse problem, where observed pixels $\mathbf{y} = H\mathbf{z} + \epsilon$ arise from a known degenerate operator $H$ and Gaussian noise $\epsilon$. These methods approximate the intractable posterior $q(\mathbf{z}|\mathbf{y})$ using the diffusion prior $p(\mathbf{z})$ through either heuristic losses ($\|\mathbf{y} - H\mathbf{z}\|_2^2$) or DDIM-based variational inference (Chung et al., 2022a;b; Grechka et al., 2024; Kawar et al., 2022; Zhang et al., 2023a; Janati et al., 2024; Ben-Hamu et al., 2024). While linear inverse problem is applicable to other generative models (e.g., GANs (Goodfellow et al., 2014)) and tasks such as deblurring, they fundamentally address a different problem: The posterior $q(\mathbf{z}|\mathbf{y})$ is a heuristic approximation that aims to construct a visually plausible $\mathbf{z} = (\mathbf{x}, \mathbf{y})$ without requiring $\mathbf{z}$ to follow exactly the joint distribution $p(\mathbf{z})$ modeled by the pretrained diffusion model.

Training-based approaches (Zhang et al., 2023b; Mayet et al., 2024; Zhuang et al., 2024) also address conditional sampling and achieve good performance. However, these methods require training of specialized networks or modules for each model architecture, making them impractical for adoption across business and community models of diverse architectures, hindering their ecosystem development.

In this work, we propose **LanPaint**, a training-free and efficient partial conditional sampling method based on Langevin Dynamics Monte Carlo, tailored for ODE-based diffusion samplers and rectified flow models. LanPaint achieves asymptotically exact partial conditional sampling without heuristics. It introduces two core innovations: (1) *Bidirectional Guided (BiG) Score*, which enables mutual adaptation between inpainted and observed regions, and avoids local maxima traps caused by ODE-samplers with large diffusion step sizes. This significantly improves inpainting quality; and (2) *Fast Langevin Dynamics (FLD)*, an accelerated Langevin sampling scheme that yields high-fidelity results in just 5 inner iterations per step, drastically reducing computational costs compared to prior Langevin methods. Experiments confirm that LanPaint outperforms existing training-free approaches, delivering high-quality inpainting and outpainting results for both pixel-space and latent-space models.

## 2 Related Works

### 2.1 ODE-based Sampling Methods and Rectified Flow

Vanilla DDPMs are slow, requiring numerous denoising steps. Acceleration strategies like approximate diffusion processes (Song et al., 2020c; Liu et al., 2022a; Song et al., 2020a; Zhao et al., 2023) and advanced ODE solvers (Karras et al., 2022; Lu et al., 2022; Zhao et al., 2023) convert stochastic DDPM sampling into deterministic ODE flows, enabling larger time steps and faster generation, dominating current diffusion model sampling. Rectified flow (Lipman et al., 2022; Liu et al., 2022b), a recent alternative, is a reparameterization of ODE-based diffusion models with improved numerical properties. As shown by (Gao et al., 2025), it also belongs to the ODE sampling family.

### 2.2 Training-Free Partial Conditional Sampling with Diffusion Models

While DDPMs have achieved significant success, they lack inherent support for partial conditional sampling with partial observations.

**LMC**    One family of works tackling this problem is Langevin dynamics Monte Carlo (LMC). This approach was pioneered by RePaint (Lugmayr et al., 2022), which employs a "time travel" mechanism of iterative denoising and renoising steps. This mechanism was later shown by (Cornwall et al., 2024) to be equivalent to LMC. A crucial advantage of this formulation is that it enables easy switching between stochastic and ODE sampling, by simply switching the denoising step from SDE to ODE.

The original RePaint framework relies on computationally intensive DDPM sampling and suffers from convergence issues. TFG (Cornwall et al., 2024) addresses the convergence issue by reformulating RePaint's "time travel" mechanism as an independent Langevin dynamics. However, their approach remains confined to DDPMs and does not extend to more efficient ODE-based solvers. (Janati et al., 2024) also used Langevin dynamics for linear inverse problems to reduce bias in optimizing the heuristic loss $\|\mathbf{y} - H\mathbf{z}\|_2^2$, but the heuristic prevents accurate partial conditional sampling.

In this work, we first adapt both RePaint and the Langevin dynamics approach from (Cornwall et al., 2024) to an ODE sampler as our baseline. Through this implementation, we identify their common limitation—susceptibility to local maxima trapping—and subsequently address it with our proposed bidirectional guidance.

**SMC**    Alternative approaches based on Sequential Monte Carlo (Wu et al., 2024; Trippe et al., 2022) provide exact partial conditional sampling but remain computationally expensive, requiring hundreds of steps and large filtering particle sets. While (Wu et al., 2024) developed a more efficient SMC variant, their method still depends on DDPM's probabilistic framework, making it incompatible with deterministic ODE-based solvers.

**Linear Inverse Problems**    Methods based on linear inverse problems target plausible inpainting results rather than exact partial conditional sampling. These approaches infer $\mathbf{z}$ from observations $\mathbf{y}$ under the model $\mathbf{y} = H\mathbf{z} + \epsilon$, where $H$ is a known degenerate operator and $\epsilon$ represents Gaussian noise.

One approach minimizes heuristic losses $\|\mathbf{y} - H\mathbf{z}\|_2^2$, as in MCG (Chung et al., 2022b), DPS (Chung et al., 2022a), GradPaint (Grechka et al., 2024), DCPS (Janati et al., 2024), and D-Flow (Ben-Hamu et al., 2024). However, most of these methods are tailored for stochastic DDPM samplers without easy migration to ODE samplers (e.g. DCPS), requiring costly full-model differentiation or expensive optimization (e.g., line search in D-Flow). For our baselines, we select only those compatible with ODE samplers and free from line search.

Another approach uses variational inference in the DDIM framework, such as DDRM (Kawar et al., 2022). CoPaint (Zhang et al., 2023a) and MMPS (Rozet et al., 2024) also adopt variational inference and expectation-maximization to improve the optimization of heuristic losses and enhance stability. While DDIM enables fast deterministic sampling, its variational approximations introduce limitations.

While we include these inverse problem baselines for comparison, they differ fundamentally from partial conditional sampling. Linear inverse problems (effective for inpainting) do not enforce matching the joint distribution between inpainted and known regions—a core requirement of our approach. Moreover, minimizing the heuristic loss typically requires 2–4 times more GPU memory than standard inference, making it prohibitive for production-level models on consumer GPUs. Accordingly, these baselines are included as supplementary reference points rather than essential benchmarks.

### 2.3 Trained Partial Conditional Sampling with Diffusion Models

While joint diffusion models lack inherent partial conditional sampling capability, inpainting can be achieved by training conditional diffusion models with external guidance. Approaches like ControlNet (Zhang et al., 2023b) (using depth/canny maps), TD-Paint (Mayet et al., 2024), and PowerPaint (Zhuang et al., 2024) demonstrate this, but require training specialized modules for each architecture, limiting their practical adoption across diverse diffusion models. This training-dependent paradigm hinders ecosystem development around new large-scale models, highlighting the need for generalizable, architecture-agnostic inpainting solutions.

## 3 Background

### 3.1 Langevin Dynamics

Langevin Dynamics is a Monte Carlo sampling technique. For a target distribution $p(\mathbf{z})$, the dynamics is governed by the stochastic differential equation (SDE):

$$d\mathbf{z}_\tau = \mathbf{s}(\mathbf{z}_\tau)\, d\tau + \sqrt{2}\, d\mathbf{W}_\tau, \tag{1}$$

where $\mathbf{s}(\mathbf{z}) = \nabla_{\mathbf{z}} \log p(\mathbf{z})$. It asymptotically converges to the stationary distribution $\mathbf{z}_\tau \sim p(\mathbf{z})$ as $\tau \to \infty$ (Appendix B). However, this method risks trapping samples at local likelihood maxima of $p(\mathbf{z})$.

### 3.2 DDPM and ODE Based Sampling

DDPMs learn a target distribution $p(\mathbf{z})$ by reconstructing a clean data point $\mathbf{z}_0 \sim p(\mathbf{z})$ from progressively noisier versions. The forward diffusion process gradually contaminates $\mathbf{z}_0$ with Gaussian noise. A discrete-time formulation of this process is:

$$\mathbf{z}_i = \sqrt{\bar{\alpha}_{t_i}}\, \mathbf{z}_0 + \sqrt{1 - \bar{\alpha}_{t_i}}\, \bar{\boldsymbol{\epsilon}}_i, \quad 1 \leq i \leq n, \tag{2}$$

where $\bar{\boldsymbol{\epsilon}}_i \sim \mathcal{N}(\mathbf{0}, \mathbf{I})$ is Gaussian noise. This arises from discretizing the continuous-time Ornstein–Uhlenbeck (OU) process:

$$d\mathbf{z}_t = -\tfrac{1}{2}\, \mathbf{z}_t\, dt\ +\ d\mathbf{W}, \tag{3}$$

where $\bar{\alpha}_t = e^{-t}$ and $d\mathbf{W}$ is a Brownian motion increment (Appendix C). The OU process ensures $\mathbf{z}_t$ transitions smoothly from the data distribution $p(\mathbf{z})$ to pure noise $\mathcal{N}(\mathbf{0}, \mathbf{I})$.

To sample from $p(\mathbf{z})$, we reverse the diffusion process. Starting from noise $\mathbf{z}_T \sim \mathcal{N}(\mathbf{0}, \mathbf{I})$, the SDE and ODE backward diffusion processes are:

$$\text{SDE: } d\mathbf{z}_{t'} = \left(\tfrac{1}{2}\mathbf{z}_{t'} + \mathbf{s}(\mathbf{z}_{t'}, T - t')\right) dt' + d\mathbf{W}_{t'}; \quad \text{ODE: } d\mathbf{z}_{t'} = \tfrac{1}{2}\left(\mathbf{z}_{t'} + \mathbf{s}(\mathbf{z}, T - t')\right) dt'. \tag{4}$$

where $t' \in [0, T]$ is the backward time, $t = T - t'$, and $\mathbf{s}(\mathbf{z}, t) = \nabla_{\mathbf{z}} \log p_t(\mathbf{z})$ is the *score function* of the random variable $\mathbf{z}_t$, which guides noise removal. The score function is usually learnt as a denoising neural network (Appendix C and D). The recent popular flow matching model, though commonly thought as a different architecture, also falls into this category (Appendix E).

### 3.3 Inpainting as Partial Conditional Sampling

Unlike many works treating inpainting as solving an ill-posed inverse problem, we treat inpainting as a partial conditional sampling problem for diffusion models: given a joint distribution of images (DDPM), how to sample one part given the other part of the image. Partial conditional sampling in DDPMs poses a significant challenge due to the inaccessibility of the conditional score function. A DDPM is trained to model a joint distribution $p(\mathbf{z}) = p(\mathbf{x}, \mathbf{y})$, where $\mathbf{z}$ is the whole image, $\mathbf{x}$ denotes the region to be inpainted, and $\mathbf{y}$ denotes the region to be observed. But partial conditional sampling aims to generate $\mathbf{x} \sim p(\mathbf{x} \mid \mathbf{y} = \mathbf{y}_o)$, where $\mathbf{y}_o$ is the observed region of a given image. During the sampling process, DDPM's denoising network is trained to provide these joint scores:

$$\mathbf{s}_{\mathbf{x}}(\mathbf{x}, \mathbf{y}, t) = \nabla_{\mathbf{x}} \log p_t(\mathbf{x}, \mathbf{y}); \quad \mathbf{s}_{\mathbf{y}}(\mathbf{x}, \mathbf{y}, t) = \nabla_{\mathbf{y}} \log p_t(\mathbf{x}, \mathbf{y}), \tag{5}$$

for every time $t$. However, the conditional score

$$\mathbf{s}_{\mathbf{x}|\mathbf{y}_o} = \nabla_{\mathbf{x}} \log p_t(\mathbf{x} \mid \mathbf{y}_o) = \nabla_{\mathbf{x}} \log p_t(\mathbf{x}, \mathbf{y} \mid \mathbf{y}_o), \tag{6}$$

which is required for direct sampling from $p(\mathbf{x} \mid \mathbf{y}_o)$, remains inaccessible. (The second equality holds because $\mathbf{x}$ and $\mathbf{y}$ are conditionally independent given $\mathbf{y}_o$.)

**Decoupling Approximation** Rather than tracking the unknown distribution $p_t(\mathbf{x}, \mathbf{y} \mid \mathbf{y}_o)$, we can approximate it as an alternative distribution $q_t(\mathbf{x}, \mathbf{y} \mid \mathbf{y}_o)$:

$$p_t(\mathbf{x}, \mathbf{y} \mid \mathbf{y}_o) \approx q_t(\mathbf{x}, \mathbf{y} \mid \mathbf{y}_o) = p_t(\mathbf{x} \mid \mathbf{y}) \cdot p_t(\mathbf{y} \mid \mathbf{y}_o). \tag{7}$$

This decoupling introduces dependencies between $\mathbf{x}_t$ and $\mathbf{y}_t$ - unlike the original DDPM framework, where $\mathbf{x}_t$ and $\mathbf{y}_t$ are independent given $\mathbf{y}_o$. In particular, the approximation ($\approx$) becomes exact ($=$) at $t = 0$ because $p_{t=0}(\mathbf{y} \mid \mathbf{y}_o) = \delta(\mathbf{y} - \mathbf{y}_o)$, ensuring that the final output still follows precisely $p(\mathbf{x} \mid \mathbf{y} = \mathbf{y}_o)$.

Here, $p_t(\mathbf{y} \mid \mathbf{y}_o)$ is analytically known from the forward process $p_t(\mathbf{y} \mid \mathbf{y}_o) = \mathcal{N}\big(\mathbf{y} \mid \sqrt{\bar{\alpha}_t} \mathbf{y}_o, (1 - \bar{\alpha}_t)\mathbf{I}\big)$, while $p_t(\mathbf{x} \mid \mathbf{y})$ shares the same score $\mathbf{s}_{\mathbf{x}}$ as the joint distribution in Eq.5. This makes the approximation tractable in practice. **The only problem that remains is how to let the DDPM generate samples from $q_t(\mathbf{x}, \mathbf{y} \mid \mathbf{y}_o)$ instead of $p_t(\mathbf{x}, \mathbf{y})$ during the sampling process.**

**The Replace Method** During backward diffusion sampling with $t' = T - t$, (Song & Ermon, 2019b) approximately samples $\mathbf{x}_{t'}, \mathbf{y}_{t'} \sim q_{T-t'}(\mathbf{x}, \mathbf{y} \mid \mathbf{y}_o)$ by replacing unconditionally sampled $\mathbf{y}_{t'}$ with

$$\mathbf{y}_{t'} \sim p_{T-t'}(\mathbf{y} \mid \mathbf{y}_o) \tag{8}$$

for each time step of a sampling process. It correctly samples $\mathbf{y}_{t'}$, but fails to ensure $\mathbf{x}_{t'} \sim p_{T-t'}(\mathbf{x}|\mathbf{y})$. This yields a mix between unconditional and partial conditional sampling

$$(\mathbf{x}_{t'}, \mathbf{y}_{t'}) \sim \text{Between} \left[ p_{T-t'}(\mathbf{x}, \mathbf{y}) \right] \text{ and } \left[ q_{T-t'}(\mathbf{x}, \mathbf{y} \mid \mathbf{y}_o) \right]. \tag{9}$$

While straightforward, this method is inaccurate. In inpainting tasks, for example, it may create sharp boundaries between the conditioned and unconditioned regions (Lugmayr et al., 2022).

**RePaint** (Lugmayr et al., 2022) improves over the Replace Method by introducing a "time traveling" step that refines $\mathbf{x}_{t'}$. Between two backward diffusion times $t'_i = T - t_i$ and $t'_{i+1} = T - t_{i+1}$, it performs multiple inner iterations, alternating between forward and backward diffusion steps:

$$\mathbf{x}_{t'_i}^{(k+1)} = \underbrace{\text{Forward}_{t_{i+1} \to t_i}}_{\text{via Eq.3}} \circ \underbrace{\text{Backward}_{t'_i \to t'_{i+1}}}_{\text{Eq.4 (SDE) and } \mathbf{s}_{\mathbf{x}}} (\mathbf{x}_{t'_i}^{(k)}). \tag{10}$$

Meanwhile, $\mathbf{y}_{t'}$ is still replaced at each step by Eq.8. As this work focuses on ODE sampling, we replace RePaint's SDE backward steps with an ODE backward (Euler ODE sampler (Karras et al., 2022)), and name this adaptation Repaint-Euler.

By adding forward and backward together, Repaint effectively simulates a *Langevin dynamics* Eq.1 with $d\tau = t'_{i+1} - t'_i$, whose stationary distribution is $p_{t_i}(\mathbf{x} \mid \mathbf{y})$. Therefore after sufficient iterations, RePaint asymptotically produces samples from $q_t(\mathbf{x}, \mathbf{y} \mid \mathbf{y}_o)$.

RePaint's key limitation is that its step size $d\tau$ is fixed by the backward sampling schedule. A small number of sampling steps leads to overly large step sizes, causing divergence, while too many steps result in excessively small step sizes, preventing convergence to a stationary state.

**Langevin Dynamics Methods**   Recent work (Cornwall et al., 2024) replaces forward-backward iteration with Langevin dynamics, enabling flexible step sizes. However, two key limitations remain: **(1)** Samples $\mathbf{x}_{t'}$ often get trapped in local maxima of $p_t(\mathbf{x} \mid \mathbf{y})$ (Fig.3); **(2)** Slow convergence necessitates many costly inner-loop iterations per diffusion step, reducing efficiency.

## 4   Methodology

### 4.1   Bidirectional Guided (BiG) Score

In RePaint and Langevin-based approaches, the inpainted region $\mathbf{x}_{t'}$ at backward diffusion time $t'$ is iterated towards high likelihood region of $p_{T-t'}(\mathbf{x} \mid \mathbf{y})$, but the observed region $\mathbf{y}_{t'}$ remains unaware of $\mathbf{x}_{t'}$. This *one-way* dependency creates a critical flaw: if $\mathbf{x}_{t'}$ enters a suboptimal region, $\mathbf{y}_{t'}$ cannot receive corrective feedback, resulting in local maxima trapping of $\mathbf{x}_{t'}$(Fig.3). To escape such local optima, we propose to jointly optimize $\mathbf{x}_{t'}$ and $\mathbf{y}_{t'}$ through bidirectional feedback: while $\mathbf{x}_{t'}$ is refined by $\mathbf{y}_{t'}$ (as in prior work), $\mathbf{y}_{t'}$ is also updated under the guidance of $\mathbf{x}_{t'}$ while preserving observed content.

To achieve this, we observe that Eq.7 is a special case of the following equivalent but more general form

$$p_t(\mathbf{x}, \mathbf{y} \mid \mathbf{y}_o) \approx q_{\lambda,t}(\mathbf{x}, \mathbf{y} \mid \mathbf{y}_o) = \frac{1}{Z}\, p_t(\mathbf{x} \mid \mathbf{y})\, \frac{p_t(\mathbf{y}|\mathbf{y}_o)^{1+\lambda}}{p_t(\mathbf{y})^{\lambda}}, \tag{11}$$

with $\mathbf{y}_o$ as the observed region of a given clean image, $\lambda > -1$ as the guidance scale and $Z$ is a normalizing constant. **At $t = 0$, the approximation ($\approx$) still becomes exact ($=$), which means that a sampled x from $q_{\lambda,t}$ is also a sample of the desired conditional distribution $p(\mathbf{x} \mid \mathbf{y}_o)$ at $t = 0$.** The "=" holds at $t = 0$ because $p_0(\mathbf{y} \mid \mathbf{y}_o) = \delta(\mathbf{y} - \mathbf{y}_o)$ is a delta distribution that remains invariant (up to normalization) when multiplied by other functions. Therefore $\frac{p_{t=0}(\mathbf{y}|\mathbf{y}_o)^{1+\lambda}}{p_{t=0}(\mathbf{y})^{\lambda}}$ still represents $\delta(\mathbf{y} - \mathbf{y}_o)$, enforcing that the sampled $\mathbf{y}_{t=0}$ equals $\mathbf{y}_o$.

Now the only problem that remains is how to let the diffusion model generate samples from $q_{\lambda,t}$ with Langevin dynamics. The $\mathbf{x}$ component of the score function of $q_{\lambda,t}$ is still $\mathbf{s_x}$ in Eq.5, while the $\mathbf{y}$ component score can be approximated by the **BiG score**

$$\mathbf{g}_\lambda\big(\mathbf{x}, \mathbf{y}, t\big) = \Big[(1+\lambda)\ \underbrace{\frac{\sqrt{\bar{\alpha}_t}\mathbf{y}_o - \mathbf{y}}{1 - \bar{\alpha}_t}}_{\text{Score of } p(\mathbf{y}_t|\mathbf{y}_o)}\ -\lambda\ \underbrace{\mathbf{s_y}\big(\mathbf{x}, \mathbf{y}, t\big)}_{\text{Score of } p(\mathbf{y}_t|\mathbf{x}_t)}\ \Big], \tag{12}$$

which is obtained by substituting $p_t(\mathbf{y}) = \frac{p_t(\mathbf{x},\mathbf{y})}{p_t(\mathbf{x}|\mathbf{y})}$ into Eq.11 then discarding the unknown term $\nabla_{\mathbf{y}} \log p_t(\mathbf{x}|\mathbf{y})$. It successfully incorporates information of $\mathbf{x}_t$ as guidance for the $\mathbf{y}_t$ sampling process.

The BiG score's behavior depends on $\lambda$: when $\lambda = -1$, it reduces to unconditional sampling; at $\lambda = 0$, it matches RePaint-like inpainting; and for $\lambda > 0$, it enhances inpainting by penalizing unconditional scores $\mathbf{s_y}$. Larger $\lambda$ values strengthen the corrective feedback from $\mathbf{x}_t$, helping $\mathbf{y}_t$ to escape local optima more effectively.

The BiG score can be implemented by simulating the following Langevin dynamics:

$$d\mathbf{x}_{t'} = \mathbf{s}_{\mathbf{x}}(\mathbf{x}_{t'}, \mathbf{y}_{t'}, T - t')d\tau + \sqrt{2}\, d\mathbf{W}_{t'}^x, \quad d\mathbf{y}_{t'} = \underbrace{\mathbf{g}_\lambda(\mathbf{x}_{t'}, \mathbf{y}_{t'}, T - t')}_{\text{BiG score}}\, d\tau + \sqrt{2}\, d\mathbf{W}_{t'}^y, \tag{13}$$

Though the unknown $\nabla_{\mathbf{y}} \log p_t(\mathbf{x}|\mathbf{y})$ term is discarded when deriving the BiG score, it still yields asymptotically exact conditional samples given partial observation, as this term is negligible (See Fig.9, Appendix.F) near $t = 0$ compared to the score of $p_t(\mathbf{y}|\mathbf{y}_o)$, as the following theorem states:

**Theorem 4.1** (Asymptotic Exact Conditional Sampler)**.** *Under the dynamics of Eq.13, the joint state* $(\mathbf{x}_t, \mathbf{y}_t)$ *converges to the distribution*

$$\mathbf{x}_t, \mathbf{y}_t \sim \frac{1}{Z} p_t(\mathbf{x} \mid \mathbf{y}) \frac{p_t(\mathbf{y}|\mathbf{y}_o)^{1+\lambda}}{p_t(\mathbf{y})^\lambda} + \mathcal{O}(\sqrt{1 - \bar{\alpha}_t}), \tag{14}$$

*where $Z$ is a normalizing constant. Consequently, at $t = 0$, the marginal $\mathbf{x}_0 \sim p_0(\mathbf{x} \mid \mathbf{y}_o)$ is an **exact conditional sample**, provided the Langevin dynamics converges. (Proof in Appendix F)*

In summary, the BiG score enables bidirectional feedback between $\mathbf{x}_{t_i}$ and $\mathbf{y}_{t_i}$, avoiding local maxima trapping while preserving the exactness of conditional sampling.

## 4.2 Fast Langevin Dynamics (FLD)

Solving Langevin dynamics (Eq.13) is challenging: direct discretization requires trading step size against performance. Large steps accelerate convergence but introduce significant errors that yield noisy results, while small steps cause impractically slow convergence. We therefore seek an accelerated scheme with a stable solver, ensuring fast convergence to the stationary distribution while tolerating larger steps.

For fast convergence to the stationary distribution, existing approaches include Underdamped Langevin Dynamics (ULD) (Duncan et al., 2017; Cheng et al., 2018), preconditioning (AlRachid et al., 2018), and HFHR (Li et al., 2022). We exclude Metropolis-Hastings and Hamiltonian Monte Carlo, as their acceptance-rejection steps require multiple score evaluations per step—which is too expensive compared to standard Langevin dynamics' single evaluation.

After balancing interpretability, stability, and accuracy (Appendix G), we propose Fast Langevin Dynamics (FLD)—a variant of ULD defined by:

$$\begin{aligned} d\mathbf{z}_\tau &= \mathbf{q}_\tau\, d\tau \\ d\mathbf{q}_\tau &= \Gamma\left(-\mathbf{q}_\tau\, d\tau + \mathbf{s}(\mathbf{z}_\tau, t)\, d\tau + \sqrt{2}\, dW_\tau\right) \end{aligned} \tag{15}$$

where $\tau$ is the "time" of Langevin dynamics, $\mathbf{z}_\tau = (\mathbf{x}_\tau, \mathbf{y}_\tau)$ contains both the inpainted/known region, $\Gamma$ is the friction coefficient, $\mathbf{q}_\tau$ is the momentum. This dynamics is solved numerically using the FLD solver Eq.115, which computes $\mathbf{z}_{\tau+\Delta\tau}$ analytically from $\mathbf{z}_\tau$. Details about FLD and its solver are discussed in Appendix G and Algorithm 4.

The FLD and its solver incorporates two key design features: **(1)** It introduces **momentum $\mathbf{q}_\tau$** into the Langevin dynamics. Comparing with Eq.1 reveals that Eq.15 represents a time-averaged Langevin dynamics with decay rate $\Gamma$. This time averaging acts as momentum by incorporating memory of previous states, thereby accelerating convergence towards the stationary distribution. **(2)** We introduce a **diffusion damping force** when solving FLD numerically to enhance stability. The diffusion damping force is introduced by decomposing the score function as $\mathbf{s}(\mathbf{z}_\tau, t) = \mathbf{C}_t(\mathbf{z}_\tau) - A_t\mathbf{z}_\tau$ in the FLD solver. The term $\mathbf{C}_t(\mathbf{z}_\tau)$ is treated as constant over a numerical time interval $[\tau, \tau + \Delta\tau]$, while the **diffusion damping force** $-A_t\mathbf{z}_\tau$ serves as a regularization inspired by the forward diffusion process, ensuring that $\mathbf{z}_{\tau+\Delta\tau}$ remains finite and stable, even for large $\Delta\tau$.

To understand how the diffusion damping force is related to the diffusion model and enhances stability, consider $A_t = (1 - \bar{\alpha}_t)^{-1}$. As $\Delta\tau \to \infty$, $\mathbf{z}_{\tau+\Delta\tau}$ remains finite and stable, following:

$$\lim_{\Delta\tau \to \infty} \mathbf{z}_{\tau+\Delta\tau} \sim \mathcal{N}\left(\sqrt{\bar{\alpha}_t}\hat{\mathbf{z}}_0, 1 - \bar{\alpha}_t\right), \tag{16}$$

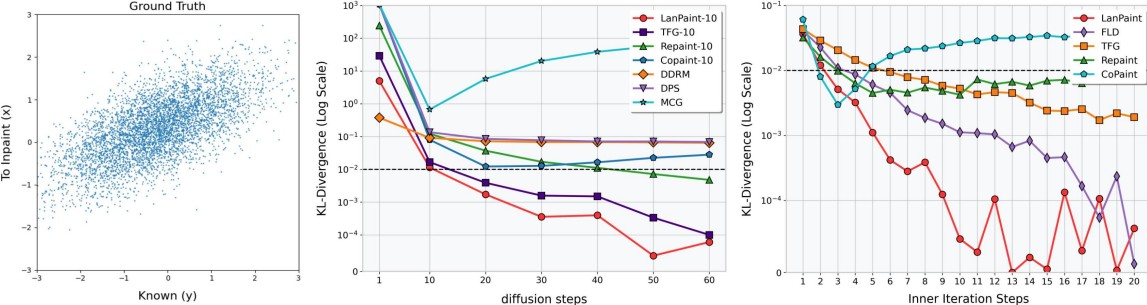

Figure 2: Comparison of inpainting methods for conditional distribution sampling (known y, inpaint x). Left: Ground truth Gaussian samples. Middle: KL divergence versus diffusion steps across methods ("-10" denotes 10 inner iterations where applicable). Right: Effect of inner iterations at 20 diffusion steps. The dashed line at KL=0.01 highlights the performance gap between asymptotically exact methods and heuristic approaches.

where $\hat{\mathbf{z}}_0 = \frac{\mathbf{z}_\tau + (1 - \bar{\alpha}_t)\mathbf{s}}{\sqrt{\bar{\alpha}_t}}$ is the Tweedie estimator for the clean image. This matches the forward diffusion process in Eq.2, ensuring stability. Thus, large time steps can accelerate convergence without compromising output stability.

A key property of FLD is that it preserves the stationary distribution of the original Langevin dynamics, as shown in the following theorem.

**Theorem 4.2** (Stationary Distribution). *Under the fast Langevin dynamics Eq.15, the joint state* $(\mathbf{z}, \mathbf{q})$ *has a stationary distribution given by*

$$(\mathbf{z}, \mathbf{q}) \sim p(\mathbf{z}) \times \mathcal{N}(\mathbf{q} \mid \mathbf{0}, \Gamma). \tag{17}$$

*Hence,* $\mathbf{z}$ *alone retains the same stationary distribution as the original Langevin dynamics Eq.1. (Proof in Appendix H)*

### 4.3 Rectified Flow Model Compatibility

We have introduced LanPaint using variance-preserving (VP) notation (Song et al., 2020c), corresponding to the forward diffusion process in Eq.2. However, LanPaint is not limited to VP notation; it is general enough to apply to other mathematically equivalent diffusion frameworks, such as variance-exploding and rectified flow notations (Liu et al., 2022b), by converting the score function into an eps-prediction or velocity prediction function, respectively. Detailed conversions among VP, variance-exploding, and rectified flow notations are provided in Appendix E.

### 4.4 Generalization to Arbitrary Dimensions

LanPaint's core formulations, the BiG score (Eq. 12) and FLD (Eq. 15)—are dimension-agnostic, operating on joint scores $\mathbf{s}(\mathbf{x}, \mathbf{y}, t)$ for arbitrary-dimensional tensors $\mathbf{z} = (\mathbf{x}, \mathbf{y}) \in \mathbb{R}^d$ $(d \geq 1)$ without 2D-specific assumptions. This enables exact conditional sampling $\mathbf{x} \sim p(\mathbf{x} \mid \mathbf{y})$ across modalities (e.g., 1D audio sequences, 2D images, 3D volumes, spatio-temporal data such as video). As an example, we extend LanPaint to video inpainting, treating video as a tensor $\mathbf{z} \in \mathbb{R}^{F \times H \times W}$ (stacked frames $\mathbf{z}^{(f)} \in \mathbb{R}^{H \times W}, f = 1, ..., F$) with static masks in Sec.5.6.

# 5   Experiments

## 5.1   Conditional Gaussian: Exactness of LanPaint

We validate the exactness of LanPaint on a synthetic 2D conditional Gaussian benchmark with an analytically known ground truth distribution and score function. This setup eliminates diffusion model training effects, allowing for an isolated comparison of sampling methods.

The task conditions on the y component to infer the x component. We compute the mean and covariance matrix of 50,000 samples and compare them with the ground truth distribution using KL divergence. Fig.2 shows the method comparisons across three plots: Ground Truth, KL Divergence vs. Diffusion Steps, and KL Divergence vs. Inner Iteration Steps. We adopt the same step size for Langevin-based methods (TFG and LanPaint).

Fig.2 also demonstrates that LanPaint achieves near-zero KL divergence with the fewest diffusion steps and inner iteration steps, outperforming other methods. Fig.2 (right) also highlights that fast Langevin dynamics (FLD) alone, without BiG score, significantly accelerates convergence compared to the TFG method adopting the original Langevin dynamics.

A key observation is that heuristic linear inverse problem approaches (MCG, DPS, CoPaint, DDRM) cannot achieve KL divergence below 0.01 (dashed line) even with a large number of steps or iterations, while exact conditional sampling methods (RePaint, TFG, LanPaint) succeed. This performance gap reflects their fundamental methodological difference: the former optimize heuristic objectives rather than the true distribution.

## 5.2   Mixture of Gaussian: Local Maxima Trapping

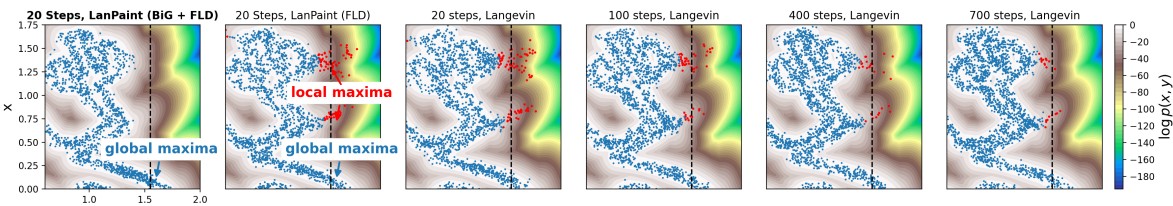

Figure 3: Local maxima trapping in inpainting $x$ given known $y$ using Euler sampler (Karras et al., 2022). Red dots show unlikely samples trapped at local maxima of $p(x|y = 1.55)$ (along the dashed line). Fewer diffusion steps (right → left) increase trapping—a key limitation of fast ODE sampler. Left panel shows LanPaint's BiG score mitigates this issue. Methods perform 10 inner iterations/step (LanPaint, Langevin)

We validate *LanPaint* on a 500-component Gaussian mixture benchmark with analytical ground truth distribution. Its samples are demonstrated in Fig.4. The task is framed as 2D inpainting: given observed $y$-coordinates, infer masked $x$-values.

The multi-modal Gaussian mixture distribution poses a significant challenge for inpainting with ODE samplers. As shown in Fig.3, Langevin-based inpainting tends to concentrate samples at the "corners" of the distribution—local maxima of $p(x|y)$—despite their low joint likelihood $(x, y)$. This trapping phenomenon is not unique to Langevin methods; Fig.4 shows that other approaches also produce samples clustered at the corners, where the distribution appears blurred.

Such trapping occurs due to insufficient information flow from the inpainted component $x$ to the observed $y$. During diffusion sampling, $x$ optimizes solely for $p(x|y)$, with no mechanism to penalize low $p(y|x)$. This motivates our BiG score Eq.12, which propagates information from $x$ to $y$, steering samples away from low joint-likelihood regions, as shown in Fig.3.

Fig.4 compares sampling results and KL divergences across methods. Due to local maxima trapping, no method achieves zero KL divergence. However, LanPaint achieves significantly lower divergence than alter-

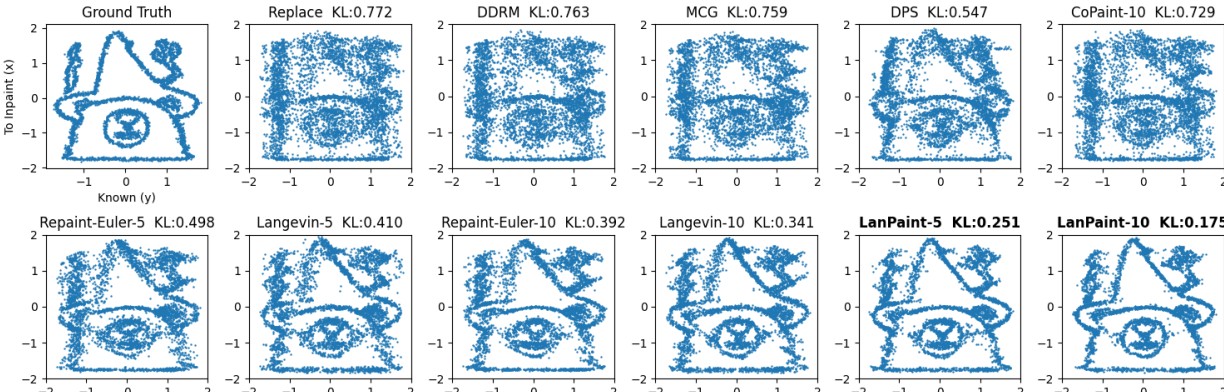

Figure 4: Inpainted samples and KL divergence for inpainting methods on a Gaussian mixture distribution. The top-left panel displays ground truth samples; other panels show inpainted samples of various methods. ("-5" and "-10" denotes 5 or 10 inner iterations where applicable)

natives, demonstrating its effectiveness in mitigating trapping and producing accurate inpainting that closely matches the ground truth distribution.

## 5.3 Latent and Pixel Space Model: CelebA and ImageNet

Table 1: LPIPS and FID comparison on CelebA-HQ-256 for various inpainting and outpainting setups. Euler Discrete Sampler, 20 steps. Lower LPIPS and FID values indicate better perceptual similarity and feature distribution similarity to the ground truth, respectively. Numerical suffixes (-5, -10) denote inner iterations (network evaluations per sampling step, except for CoPaint which requires multiple evaluations per inner iteration). Time per sample and memory overhead (extra memory required during inference) are also reported. Evaluations were conducted on a single RTX 3090.

| | Box | | Half | | Checkerboard | | Outpaint | | Time | MemOver |
|---|---|---|---|---|---|---|---|---|---|---|
| Method | LPIPS | FID | LPIPS | FID | LPIPS | FID | LPIPS | FID | s/image | MB/image |
| **Heuristic Methods** | | | | | | | | | | |
| Replace | 0.131 | 31.7 | 0.303 | **30.3** | 0.162 | 42.1 | 0.514 | 89.5 | 0.3 | 81 |
| CoPaint-2 | 0.180 | 43.6 | 0.346 | 35.7 | 0.252 | 79.8 | 0.546 | 107.6 | 1.7 | 248 |
| CoPaint-3 | 0.172 | 41.7 | 0.331 | 34.4 | 0.225 | 66.6 | 0.532 | 99.5 | 2.5 | 248 |
| DDRM | 0.128 | 32.4 | 0.308 | 33.5 | 0.148 | 30.0 | 0.537 | 94.6 | 0.3 | 81 |
| MCG | 0.130 | 31.6 | 0.302 | 30.2 | 0.162 | 42.4 | 0.513 | 80.7 | 0.8 | 248 |
| DPS | 0.181 | 44.1 | 0.345 | 35.4 | 0.275 | 90.6 | 0.534 | 99.9 | 0.8 | 247 |
| **Asymptotically Exact Methods** | | | | | | | | | | |
| Repaint-Euler-5 | 0.115 | 34.5 | 0.282 | 39.4 | 0.137 | 29.8 | 0.526 | 96.2 | 1.4 | 81 |
| Repaint-Euler-10 | 0.112 | 34.6 | 0.272 | 41.0 | 0.134 | 31.0 | 0.511 | 95.9 | 2.6 | 81 |
| TFG-5 | 0.119 | 31.9 | 0.299 | 35.9 | 0.132 | 25.5 | 0.531 | 91.0 | 1.5 | 81 |
| TFG-10 | 0.114 | 33.2 | 0.288 | 38.5 | 0.128 | 26.3 | 0.530 | 91.5 | 2.6 | 81 |
| LanPaint-5 (ours) | 0.105 | **27.9** | **0.268** | 30.4 | 0.108 | **20.5** | 0.493 | **82.3** | 1.6 | 81 |
| LanPaint-10 (ours) | **0.103** | 29.5 | 0.272 | 32.2 | **0.107** | 21.3 | **0.489** | 85.1 | 2.9 | 81 |

We evaluate the inpainting performance of LanPaint on the CelebA-HQ-256 (Liu et al., 2015) and ImageNet-256 (Deng et al., 2009) datasets, leveraging pre-trained latent (Rombach et al., 2021) and pixel space (Dhariwal & Nichol, 2021) diffusion models, respectively. The experiments assess reconstruction quality across various mask geometries, including box, half, checkerboard, and outpainting. Following the same setting as the previous works (Kawar et al., 2022; Chung et al., 2022b), perceptual fidelity is quantified through LPIPS (Zhang et al., 2018) and FID metrics, calculated on 1,000 validation images per dataset. Results are presented in Tables 1 and 2. We also provide qualitative visualization of generated samples in Fig.5 and Fig.6. All methods employ consistent parameters across tasks and masks, utilizing a 20-step Euler Discrete Sampler. Further details about parameters are provided in Appendix A.

Table 2: LPIPS and FID comparison on ImageNet for various inpainting and outpainting setups. Euler Discrete Sampler, 20 steps. Lower LPIPS and FID values indicate better perceptual similarity and feature distribution similarity to the ground truth, respectively. Numerical suffixes (-5, -10) denote inner iterations (network evaluations per sampling step, except for CoPaint, which requires multiple evaluations per inner iteration). Time per sample and memory overhead (extra memory required during inference) are also reported. Evaluations were conducted on a single RTX 3090.

| | Box | | Half | | Checkerboard | | Outpaint | | Time | MemOver |
| Method | LPIPS | FID | LPIPS | FID | LPIPS | FID | LPIPS | FID | s/image | MB/image |
|---|---|---|---|---|---|---|---|---|---|---|
| **Heuristic Methods** | | | | | | | | | | |
| Replace | 0.229 | 75.7 | 0.380 | 69.0 | 0.406 | 146.4 | 0.565 | 98.2 | 1.9 | 581 |
| CoPaint-2 | 0.234 | 85.1 | 0.379 | 63.4 | 0.319 | 186.3 | 0.565 | 102.1 | 15.7 | 5444 |
| CoPaint-3 | 0.228 | 76.9 | 0.371 | 60.8 | 0.276 | 146.9 | 0.557 | 94.8 | 22.4 | 5445 |
| DDRM | 0.216 | 67.2 | 0.385 | 60.2 | 0.214 | 58.1 | 0.570 | 81.6 | 1.9 | 583 |
| MCG | 0.225 | 83.5 | 0.378 | 82.2 | 0.429 | 152.9 | 0.561 | 106.8 | 6.4 | 5445 |
| DPS | 0.252 | 107.5 | 0.392 | 77.2 | 0.510 | 275.7 | 0.572 | 112.1 | 6.4 | 5440 |
| **Asymptotically Exact Methods** | | | | | | | | | | |
| Repaint-Euler-5 | 0.216 | 62.8 | 0.385 | 56.5 | 0.137 | 31.4 | 0.579 | 82.9 | 11.8 | 581 |
| Repaint-Euler-10 | 0.215 | 61.0 | 0.383 | 53.5 | 0.135 | 32.8 | 0.564 | 79.8 | 20.5 | 581 |
| TFG-5 | 0.235 | 69.3 | 0.418 | 67.6 | 0.317 | 78.6 | 0.654 | 92.1 | 11.9 | 595 |
| TFG-10 | 0.234 | 66.5 | 0.433 | 64.5 | 0.247 | 53.9 | 0.682 | 89.4 | 21.7 | 595 |
| LanPaint-5 (ours) | 0.180 | 49.3 | 0.323 | 49.3 | 0.127 | 24.1 | 0.508 | 68.6 | 11.3 | 599 |
| LanPaint-10 (ours) | **0.171** | **46.4** | **0.314** | **44.7** | **0.117** | **21.2** | **0.486** | **62.0** | 20.8 | 599 |

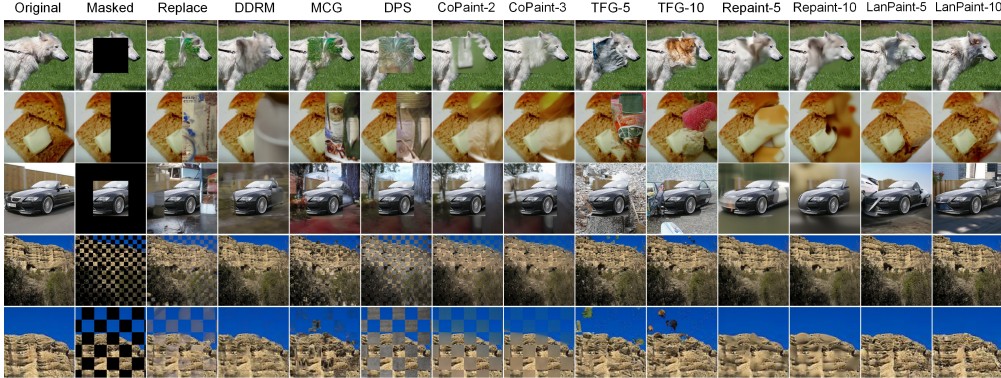

Figure 5: Visual comparisons on ImageNet-256 for center box, half, outpaint and checkerboard masks (top to bottom). Numbers 5 and 10 denote the inner iteration counts for RePaint, TFG, and LanPaint. The bottom row zooms into the checkerboard results (fourth row), highlighting LanPaint's superior coherence and texture fidelity compared to baselines, which exhibit visible checkerboard artifacts.

LanPaint consistently achieves superior LPIPS and FID scores across most test scenarios, demonstrating robustness, particularly in challenging checkerboard and outpainting tasks. In contrast, methods such as DPS (Chung et al., 2022a) and CoPaint (Zhang et al., 2023a), designed for stochastic sampling with 250–500 steps, exhibit reduced performance in the 20-step ODE setting.

Notably, DPS and CoPaint's removing of the manifold constraint from MCG (Chung et al., 2022b) compromises their stability, leading to poorer performance compared to MCG, which remains robust among heuristic methods. This contradicts prior findings in stochastic DDPM sampling, where removing the manifold constraint typically enhances performance.

On CelebA-HQ-256, Replace (Song & Ermon, 2019b) marginally outperforms LanPaint in FID for the half-mask scenario (30.3 vs. 30.4), a result attributed to FID's high variance when large inpainted regions deviate significantly from the original images. This variance is exacerbated by the use of 1,000 validation images, as

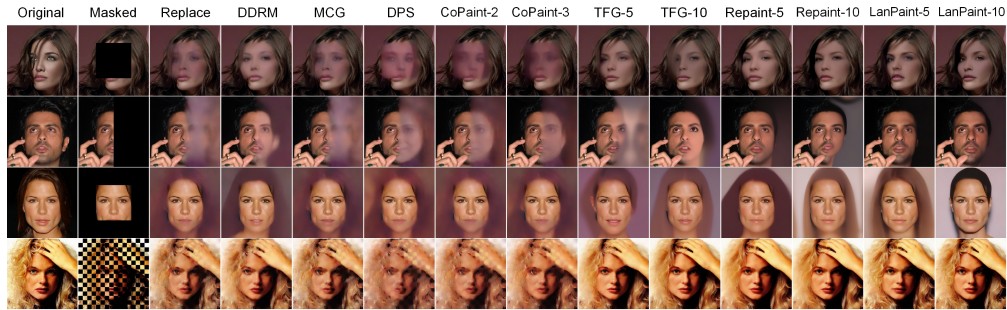

Figure 6: Comparative visualization of in-painted images in CelebA-HQ-256 dataset for center box, half, outpaint and checkerboard masks (top to bottom). Sampler: EulerDiscrete, 20 Step. For visualization purposes, masks are shown on the original pixel images, although they were applied within the $64 \times 64 \times 4$ latent space during the inpainting process.

opposed to the typical 30,000 for highly divergent sets, similarly affecting outpainting results. Consequently, FID scores for half and outpainting masks should be interpreted cautiously.

Beyond perceptual metrics, computational efficiency is critical for practical deployment. We report time per sample and memory overhead (MemOver), defined as the additional GPU memory required during inference beyond model loading (i.e., maximum GPU memory during inference minus maximum before inference). Evaluations were conducted on a single RTX 3090. On CelebA-HQ-256, LanPaint-10 delivers top-tier performance with time and memory overhead comparable to Repaint and TFG. On ImageNet-256, LanPaint-10 uses 20.8 s/image and 599 MB/image, comparable to Repaint (20.5 s, 581 MB) while achieving superior LPIPS and FID scores. Notably, LanPaint's memory overhead is low compared to heuristic methods requiring backpropagation for gradient computation, such as MCG, DPS, and CoPaint (5445 MB on ImageNet), whose overhead scales with model size, rendering them less practical for large models where loading alone nearly exhausts GPU memory. These results highlight LanPaint's effective balance of high fidelity and resource efficiency.

## 5.4 Ablation Study

Table 3 presents the ablation study of LanPaint's major components: the BiG score and FLD, along with the impact of step size on performance. The study is conducted on two datasets, CelebA-HQ-256 and ImageNet, using the box inpainting tasks (Other masks share similar trends). Results are reported for five different step sizes (0.02, 0.05, 0.1, 0.15, and 0.2) with LPIPS and FID metrics, where lower values indicate better performance. Sensitivity of other parameters is provided in Fig.8.

For the ImageNet dataset, the ablation study demonstrates that both the BiG score and FLD significantly contribute to overall performance. Without FLD, the Langevin dynamics diverge as the step size increases from 0.05 to 0.15, resulting in progressively worse performance metrics. However, incorporating FLD suppresses this divergence, enabling the use of larger step sizes and improving performance. The BiG score enhances performance by facilitating bidirectional information flow between the inpainted and known regions, while FLD supports larger step sizes, accelerating the convergence of the Langevin dynamics and yielding better results with the same number of iterations.

In contrast, for the CelebA-HQ-256 dataset, performance metrics exhibit low sensitivity to step size variations. The BiG score is the primary driver of performance improvement over the original Langevin dynamics, with the method remaining robust across step size changes. This stability is attributed to the robustness of CelebA-HQ-256's latent space, which, unlike pixel space sampling, is less affected by subtle variations in the sampling process.

Table 3: Ablation study of LanPaint-10's components on CelebA-HQ-256 and ImageNet with box inpainting. Results for different step sizes (0.02, 0.05, 0.1, 0.15, 0.2) are shown with LPIPS and FID metrics. Lower values are better.

| Method | Step Size 0.02 | | Step Size 0.05 | | Step Size 0.1 | | Step Size 0.15 | | Step Size 0.2 | |
|---|---|---|---|---|---|---|---|---|---|---|
| | LPIPS | FID | LPIPS | FID | LPIPS | FID | LPIPS | FID | LPIPS | FID |
| **CelebA-HQ-256** | | | | | | | | | | |
| None (Langevin) | 0.121 | 28.9 | 0.115 | 28.4 | 0.111 | 29.2 | 0.114 | 30.9 | 0.108 | 30.1 |
| + BiG score | 0.110 | 26.1 | 0.104 | 26.5 | 0.102 | 28.5 | 0.103 | 28.7 | 0.103 | 30.0 |
| + FLD | 0.121 | 28.6 | 0.115 | 28.9 | 0.112 | 29.9 | 0.116 | 31.7 | 0.109 | 30.5 |
| + (BiG score + FLD) | **0.111** | **26.7** | **0.105** | **26.6** | **0.103** | 28.5 | **0.103** | **29.5** | 0.103 | 30.2 |
| **ImageNet** | | | | | | | | | | |
| None (Langevin) | 0.220 | 66.7 | 0.223 | 65.1 | 0.314 | 81.6 | 0.441 | 125.9 | 0.475 | 141.1 |
| + BiG score | 0.205 | 58.6 | 0.213 | 58.0 | 0.303 | 76.0 | 0.431 | 121.4 | 0.474 | 140.0 |
| + FLD | 0.217 | 68.3 | 0.205 | 60.7 | 0.195 | 56.9 | 0.188 | 54.4 | 0.181 | 51.4 |
| + (BiG score + FLD) | **0.201** | **58.9** | **0.190** | **52.7** | **0.179** | **48.1** | **0.171** | **46.4** | **0.167** | **45.1** |

## 5.5 Production-Level Model Evaluation Across Architectures: Stable Diffusion, Flux, and HiDream

Previous evaluations of LanPaint were limited to academic benchmarks, leaving its performance on real-world production diffusion models—with their diverse architectures and higher resolutions—unexamined. Notably, to the best of our knowledge, no prior training-free inpainting methods, except variants of the replace method (built in ComfyUI), have demonstrated such validation in their publications or been implemented by third parties for this purpose.

To assess LanPaint's effectiveness on modern generative models, we implemented it on HiDream-L1 (HiDream.ai, 2025), Flux.1 Dev (Labs, 2024), Stable Diffusion 3.5 (Esser et al., 2024), and Stable Diffusion XL (Podell et al., 2023). Images were generated using ComfyUI (Comfy Org, 2025) with the Euler sampler (Karras et al., 2022) (30 steps), a fixed seed of 0, and a batch size of 4 to ensure reproducibility and avoid cherry-picking. These experiments demonstrate LanPaint's practical efficacy across diverse diffusion architectures, including rectified flow models (HiDream-L1, Flux.1, Stable Diffusion 3.5) and denoising diffusion probabilistic models (Stable Diffusion XL). Additionally, we have released LanPaint as a publicly available, plug-and-play extension for ComfyUI.

Fig.1 showcases the inpainting results. We also provide more examples in Appendix I. LanPaint consistently produces seamless inpainting across both DDPM-based (Stable Diffusion XL) and rectified flow-based (Stable Diffusion 3.5, Flux.1, HiDream-L1) architectures, highlighting its robust generalization capabilities.

## 5.6 Video Inpainting Results

We demonstrate LanPaint's dimension-agnostic formulation on video inpainting and outpainting using the Wan 2.2 T2V model Wan et al. (2025) (14B parameters, fp8-scaled). Our setup processes 40-frame, 480p clips using Euler ODE sampling with 20 steps and two LanPaint inner iterations. Fig. 7 highlights these capabilities. These results affirm LanPaint's versatility for video editing. Full videos are available at `https://github.com/scraed/LanPaint/tree/master/examples`.

## Limitation and Future work

LanPaint's exactness comes at a cost: it heavily relies on the score interpretation of diffusion models. This interpretation, while applicable to various architectures such as variance-preserving, variance-exploding, and flow matching, is valid only for models trained from scratch, not for distilled models trained without denoising or flow matching. Our experience shows that LanPaint's performance degrades with distilled models. Future studies on distillation methods that preserve the score interpretation of diffusion models are desired. A distillation method that captures LanPaint's capabilities within a model is also of interest, as it could significantly accelerate LanPaint.

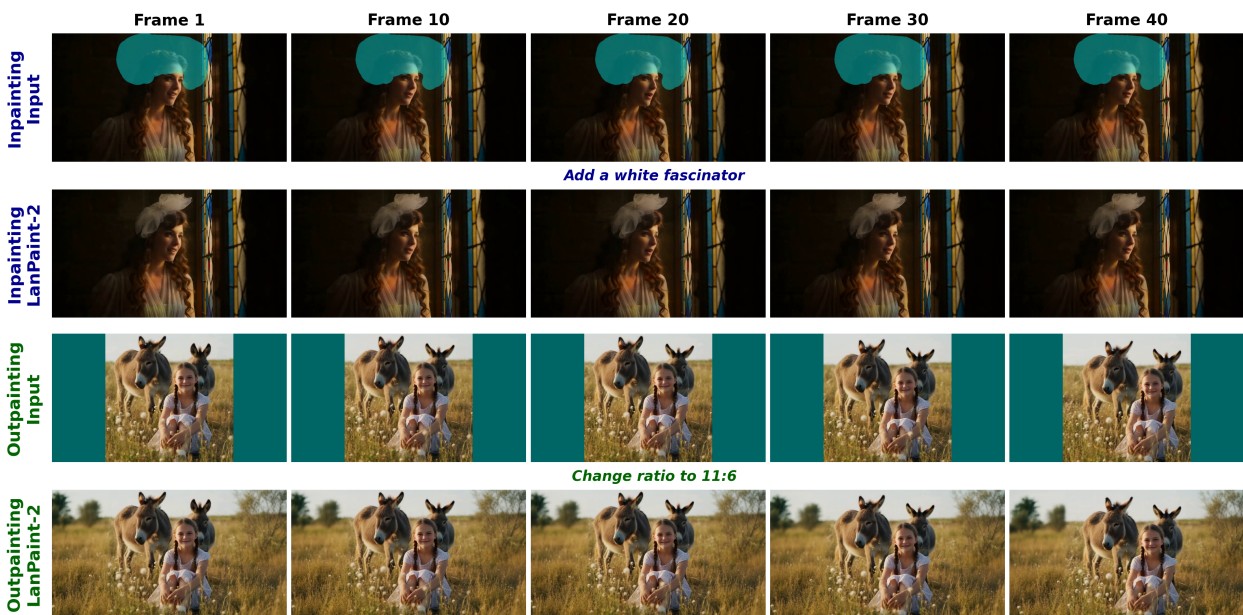

Figure 7: LanPaint-2 video inpainting and outpainting on Wan 2.2 T2V 14B models (seed=0, 480p resolution, 20 diffusion sampling steps). Top: Inpainting with prompt "Add a white fascinator" showing input with masked region (cyan) and output. Bottom: Outpainting with prompt "Change ratio to 11:6" showing input with padding regions (cyan) and output. Five keyframes displayed (frames 1, 10, 20, 30, 40).

LanPaint assumes noise-free observations $\mathbf{y}_o$. Adapting it to handle noisy observations with a specified noise level is feasible but requires modifying the conditional distribution $p_t(\mathbf{y} \mid \mathbf{y}_o)$ in Eq.11. This adaptation represents a promising direction for future research.

In this paper, we primarily focus on image inpainting. However, LanPaint, as a conditional sampling method independent of data modality, can be applied to diverse domains, including text, audio, video, and scientific applications such as protein scaffolding and fluid field reconstruction.

## Broader Impact Statement

LanPaint's efficient image inpainting boosts creative applications but risks misuse in generating deepfakes or misinformation. We advocate watermarking, provenance tracking, and community regulation to mitigate harm, as discussed in (Denton, 2021) and (Franks & Waldman, 2018).

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

# A   More Ablation and Implementation Details

**Mask Types**   The **box** mask covers a central region spanning from $1/4$ to $3/4$ of both the height ($H$) and width ($W$) of the image. The **half** mask covers the right half of the image. The **outpaint** mask covers the area outside the box mask, serving as its complement. The **checkerboard** mask forms a grid pattern with each square sized at $1/16$ of the original image dimensions. For latent space operations, these masks are applied to the encoded latent representations of the image.

**LanPaint**   We implement LanPaint using the diffusers package, following Algorithm 4. Hyperparameters are configured as follows: $\gamma = 15$, $\alpha = 0.$, and $\lambda = 8$ for all image inpainting tasks, drawing loosely from the insights gained through sensitivity analysis in Fig.8. The notation *LanPaint-5* and *LanPaint-10* denotes $N = 5$ and $N = 10$ sampling steps, respectively. The step size $\eta$ is set to 0.15 for both Celeb-A and ImageNet. The impact of step size is ablated in Table 3, with the impact of other parameters discussed in Fig.8. We have also provided impact of different samplers in Table.4.

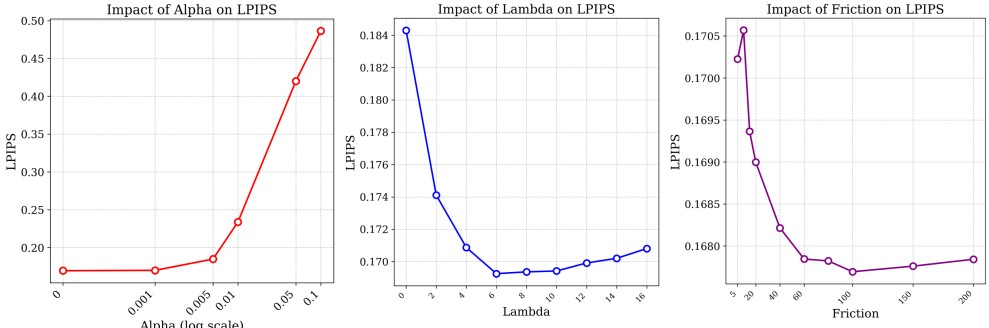

Figure 8: Impact of expected noise $\alpha$, guidance scale $\lambda$, and friction $\gamma$ on LanPaint-10's LPIPS for ImageNet box inpainting at stepsize 0.15, evaluated on a validation set of 100 images. Expected noise $\alpha$ most affects LPIPS, ideally 0 for image inpainting. Guidance scale $\lambda$ significantly improves performance from 0 (no guidance) to 4, with an optimal range of 6–10. The friction parameter $\gamma$ has a less significant effect; in practice, we use a prescribed value of $\gamma = 15$ without finetuning, sharing this value across all tasks.

Table 4: Ablation study on the impact of different diffusion samplers (Euler, DPM++Karras(Lu et al., 2025), and DDIM (Song et al., 2020b)) for various heuristic and asymptotically exact inpainting methods. Performance is evaluated using LPIPS (lower is better) and FID (lower is better) metrics on 1,000 images from the CelebA and ImageNet datasets with box mask.

| | CelebA | | | | | | ImageNet | | | | | |
|---|---|---|---|---|---|---|---|---|---|---|---|---|
| | Euler | | DPM++ | | DDIM | | Euler | | DPM++ | | DDIM | |
| Method | LPIPS | FID | LPIPS | FID | LPIPS | FID | LPIPS | FID | LPIPS | FID | LPIPS | FID |
| **Heuristic Methods** | | | | | | | | | | | | |
| Replace | 0.131 | 31.7 | 0.119 | 26.7 | 0.130 | 31.5 | 0.229 | 75.7 | 0.234 | 75.1 | 0.228 | 75.6 |
| CoPaint-2 | 0.180 | 43.6 | 0.186 | 45.8 | 0.169 | 41.1 | 0.234 | 85.1 | 0.233 | 84.5 | 0.297 | 89.1 |
| CoPaint-3 | 0.172 | 41.7 | 0.181 | 44.3 | 0.163 | 38.9 | 0.228 | 76.9 | 0.228 | 78.9 | 0.288 | 82.1 |
| DDRM | 0.128 | 32.4 | 0.135 | 34.8 | 0.127 | 32.2 | 0.216 | 67.2 | 0.215 | 68.9 | 0.211 | 65.3 |
| MCG | 0.130 | 31.6 | 0.113 | 25.9 | 0.124 | 29.7 | 0.225 | 83.5 | 0.266 | 88.1 | 0.253 | 86.7 |
| DPS | 0.181 | 44.1 | 0.166 | 40.2 | 0.179 | 43.7 | 0.252 | 107.5 | 0.248 | 103.0 | 0.249 | 108.2 |
| **Asymptotically Exact Methods** | | | | | | | | | | | | |
| Repaint-5 | 0.115 | 34.5 | 0.127 | 41.7 | 0.114 | 34.2 | 0.216 | 62.8 | 0.342 | 133.6 | 0.211 | 60.8 |
| Repaint-10 | 0.112 | 34.6 | 0.409 | 223.3 | 0.111 | 34.4 | 0.215 | 61.0 | 0.604 | 174.5 | 0.211 | 59.7 |
| TFG-5 | 0.119 | 31.9 | 0.113 | 31.9 | 0.118 | 31.4 | 0.235 | 69.3 | 0.281 | 83.7 | 0.234 | 69.7 |
| TFG-10 | 0.114 | 33.2 | 0.108 | 33.2 | 0.112 | 31.2 | 0.234 | 66.5 | 0.305 | 89.5 | 0.235 | 65.8 |
| LanPaint-5 (ours) | 0.105 | **27.9** | 0.097 | **23.7** | 0.104 | **28.0** | 0.180 | 49.3 | **0.201** | 56.2 | 0.178 | 48.6 |
| LanPaint-10 (ours) | **0.103** | 29.5 | **0.097** | 26.9 | **0.103** | 29.3 | **0.171** | **46.4** | 0.205 | **55.1** | **0.172** | **45.6** |

**TFG** TFG ((Cornwall et al., 2024)) corresponds to pure Langevin sampling. It is implemented via the Euler-Maruyama discretization, with step size schedule $d\tau = \eta\sqrt{1 - \bar{\alpha}_t}$ with $\eta = 0.04$ as reported in their paper.

**DDRM** Our implementation of DDRM follows Equations (7) and (8) in Kawar et al. (Kawar et al., 2022), with $\sigma_y = 0$, $\mathbf{V} = I$, and $s_i = 1$ for observed regions $\mathbf{y}$ and $s_i = 0$ for inpainted regions $\mathbf{x}$. The hyperparameters $\eta = 0.7$ and $\eta_b = 1$ are selected based on the optimal KID scores reported in Table 3 of the reference.

**MCG** We implement MCG as described in Algorithm 1 of Chung et al. (Chung et al., 2022b), using $\alpha = 0.1/\|\mathbf{y} - P\hat{\mathbf{x}}_0\|$ as recommended in Appendix C. An alternative choice, $\alpha = 0.1/\|\mathbf{y} - P\hat{\mathbf{x}}_0\|$, was evaluated but resulted in significantly poorer performance on ImageNet, as shown in Table 5. Consequently, we opted against using this alternative.

**DPS** We adopt DPS (Chung et al., 2022a) using $\alpha = 1/\|\mathbf{y} - P\hat{\mathbf{x}}_0\|$ as recommended in Appendix D.

Table 5: Ablation study of DPS and MCG on CelebA-HQ-256 and ImageNet with box inpainting. Results for different alpha values (0.1, 1.0) are shown with LPIPS and FID metrics. Lower values are better.

| | Alpha 0.1 | | Alpha 1.0 | |
|---|---|---|---|---|
| **Method** | **LPIPS** | **FID** | **LPIPS** | **FID** |
| **CelebA-HQ-256** | | | | |
| MCG | 0.130 | 31.6 | 0.125 | 30.1 |
| DPS | 0.190 | 41.5 | 0.181 | 44.1 |
| **ImageNet** | | | | |
| MCG | 0.226 | 83.5 | 0.240 | 79.5 |
| DPS | 0.251 | 110.7 | 0.252 | 107.5 |

**RePaint** We implement RePaint based on Algorithm 1 in Lugmayr et al. (Lugmayr et al., 2022), with modifications to accommodate fast samplers. While the original method was designed for DDPM, we adapt it by replacing the backward sampling step (Line 7) with a Euler Discrete Sample step. Additionally, we set the jump step size to 1 instead of the default 10 backward steps recommended in Appendix B of the original work. This adjustment is necessary because fast samplers typically operate with only around 20 backward sampling steps, making larger jump sizes impractical.

## B Diffusion Process and Langevin Dynamics

**Diffusion Process** forms the mathematical foundation of diffusion models, describing a system's evolution through deterministic drift and stochastic noise. Here we consider diffusion process of the following form of *stochastic differential equation (SDE)*:

$$d\mathbf{x}_t = \boldsymbol{\mu}(\mathbf{x}_t, t)\, dt + \sigma(\mathbf{x}_t, t)\, d\mathbf{W}_t, \tag{18}$$

where the drift term $\boldsymbol{\mu}(\mathbf{x}_t, t)\, dt$ governs deterministic motion, while $d\mathbf{W}_t$ adds Brownian noise.

The Brownian noise $d\mathbf{W}$ is a key characteristic of SDEs, capturing their stochastic nature. It represents a series of infinitesimal Gaussian noise. A good way to understand it is through a formally definition:

$$d\mathbf{W}_t = \sqrt{dt} \lim_{n \to \infty} \sum_{i=1}^{n} \sqrt{\frac{1}{n}} \boldsymbol{\epsilon}_i, \tag{19}$$

where $\boldsymbol{\epsilon}_i$ are independent standard Gaussian noises with mean $\mathbf{0}$ and identity covariance matrix $I$. The limit in this definition shows that $d\mathbf{W}$ is not just a single Gaussian random variable with mean $\mathbf{0}$, but rather

the cumulative effect of infinitely many independent Gaussian increments. Such cumulation allows us to compute the covariance of $d\mathbf{W}$ as vector product:

$$d\mathbf{W}_t \cdot d\mathbf{W}_t^T = \mathrm{Cov}(d\mathbf{W}_t, d\mathbf{W}_T) = I\,dt, \tag{20}$$

where $I$ is the identity matrix. When no quadratic terms of $d\mathbf{W}_t$ are involved, $d\mathbf{W}_t$ can often be roughly treated as $\sqrt{dt}\,\boldsymbol{\epsilon}$, where $\boldsymbol{\epsilon} \sim \mathcal{N}(0,1)$ is a standard Gaussian random variable.

The Brownian noise $d\mathbf{W}_t$ scales as $\sqrt{dt}$, which fundamentally alters the rules of calculus for SDEs. A change of variable in ordinary calculus has $ds = \frac{ds}{dt}dt$, but for Brownian noise it is $d\mathbf{W}_s = \sqrt{\frac{ds}{dt}}d\mathbf{W}_t$. Moreover, the differentiation of a function is $df(t, \mathbf{x}_t) = \partial_t f\,dt + \nabla_\mathbf{x} f \cdot d\mathbf{x}_t$ in ordinary calculus, but for SDE, it follows the Itô's lemma:

$$df(t, \mathbf{x}_t) = \partial_t f\,dt + \nabla_\mathbf{x} f \cdot d\mathbf{x}_t + \underbrace{\frac{\sigma^2}{2}\nabla_\mathbf{x}^2 f\,dt}_{\text{stochastic effect}}. \tag{21}$$

This is derived by differentiating $f$ using the chain rule with the help of Eq.18 and Eq.20, while keeping all terms up to order $dt$ (note that $d\mathbf{W}$ scales as $\sqrt{dt}$). The emergence of the second-order derivative term $\nabla_\mathbf{x}^2 f$ is the key distinction from ordinary calculus. We will later use this lemma to analyze the evolution of the distribution of $\mathbf{x}_t$.

**Langevin Dynamics** is a special diffusion process aims to generate samples from a distribution $p(\mathbf{x})$. It is defined as

$$d\mathbf{x}_t = \mathbf{s}(\mathbf{x}_t)dt + \sqrt{2}d\mathbf{W}_t, \tag{22}$$

where $\mathbf{s}(\mathbf{x}) = \nabla_\mathbf{x} \log p(\mathbf{x})$ is the score function.

This dynamics is often used as a Monte Carlo sampler to draw samples from $p(\mathbf{x})$, since $p(\mathbf{x})$ is its stationary distribution—the distribution that $\mathbf{x}_t$ converges to as $t \to \infty$, regardless of the initial distribution of $\mathbf{x}_0$. More precisely, this means that if an ensemble of particles $\{\mathbf{x}_t^{(i)}\}_{i=1}^N$ evolves according to the given SDE, and their initial positions $\{\mathbf{x}_0^{(i)}\}$ follow a distribution $p(\mathbf{x})$, then their positions $\{\mathbf{x}_t^{(i)}\}$ will continue to be distributed according to $p(\mathbf{x})$ at all future times $t > 0$.

To verify stationarity, we will show that after evolution from time 0 to $\Delta t$, the distribution of $\mathbf{x}_{\Delta t}$ is still $p(\mathbf{x})$. Consider a test function $f$ and initial positions $\mathbf{x}_0 \sim p(\mathbf{x})$, stationary can be assessed by tracking the change in the expectation $\mathbb{E}_{\mathbf{x}_0 \sim p(\mathbf{x})}[f(\mathbf{x}_{\Delta t})]$. Using Itô's lemma and note that $\mathbb{E}_\mathbf{x}[d\mathbf{W}] = \mathbf{0}$ for any distribution of $\mathbf{x}$, we compute:

$$\begin{aligned}
\mathbb{E}_{\mathbf{x}_0 \sim p(\mathbf{x})}\left[f(\mathbf{x}_{\Delta t}) - f(\mathbf{x}_0)\right] &\approx \Delta t \int p(\mathbf{x})\left(\nabla_\mathbf{x} f \cdot \mathbf{s} + \nabla_\mathbf{x}^2 f\right)d\mathbf{x} \\
&= \Delta t \int f(\mathbf{x})\left(-\nabla_\mathbf{x} \cdot (p\mathbf{s}) + \nabla_\mathbf{x}^2 p\right)d\mathbf{x} \quad \text{(integration by parts)} \\
&= \Delta t \int f(\mathbf{x})\nabla_\mathbf{x} \cdot (-p\mathbf{s} + \nabla_\mathbf{x} p)\,d\mathbf{x} \\
&= 0,
\end{aligned} \tag{23}$$

where 0 is obtained by substituting $\mathbf{s} = \nabla_\mathbf{x} \log p$. Because $\mathbb{E}_{\mathbf{x}_0 \sim p(\mathbf{x})}\left[f(\mathbf{x}_{\Delta t}) - f(\mathbf{x}_0)\right] = 0$ for any test function $f$, this means the distribution of $\mathbf{x}_{\Delta t}$ must have been kept the same as $\mathbf{x}_0$.

**Langevin Dynamics as 'Identity'** The stationary of $p(\mathbf{x})$ is very important: The Langevin dynamics for $p(\mathbf{x})$ acts as an "identity" operation on the distribution, transforming samples from $p(\mathbf{x})$ into new samples from the same distribution. This property enables efficient derivations of both forward and backward diffusion processes for diffusion models.

## C   The Denoising Diffusion Probabilistic Model (DDPMs)

Langevin dynamics can be used to generate samples from a distribution $p(\mathbf{x})$, given its score function $\mathbf{s}$. But its success hinges on two critical factors. First, the method is highly sensitive to initialization - a poorly chosen $\mathbf{x}_0$ may trap the sampling process in local likelihood maxima, failing to explore the full distribution. Second, inaccuracies in the score estimation, particularly near $\mathbf{x}_0$, can prevent convergence altogether. These limitations led to the development of diffusion models, which use a unified initialization process: all samples are generated by gradually denoising pure Gaussian noise.

DDPMs (Ho et al., 2020) are models that generate high-quality images from noise via a sequence of denoising steps. Denoting images as random variable $\mathbf{x}$ of the probabilistic density distribution $p(\mathbf{x})$, the DDPM aims to learn a model distribution that mimics the image distribution $p(\mathbf{x})$ and draw samples from it. The training and sampling of the DDPM utilize two diffusion process: the forward and the backward diffusion process.

**The forward diffusion process** of the DDPM provides necessary information to train a DDPM. It gradually adds noise to existing images $\mathbf{x}_0 \sim p(x)$ using the Ornstein–Uhlenbeck diffusion process (OU process) (Uhlenbeck & Ornstein, 1930) within a finite time interval $t \in [0, T]$. The OU process is defined by the stochastic differential equation (SDE):

$$d\mathbf{x}_t = -\frac{1}{2}\mathbf{x}_t dt + d\mathbf{W}_t, \tag{24}$$

in which $t$ is the forward time of the diffusion process, $\mathbf{x}_t$ is the noise contaminated image at time $t$, and $\mathbf{W}_t$ is a Brownian noise.

The forward diffusion process has the standard Gaussian $\mathcal{N}(\mathbf{0}, I)$ as its stationary distribution. Moreover, regardless of the initial distribution $p_0(\mathbf{x})$ of positions $\{\mathbf{x}_0^{(i)}\}_{i=1}^N$, their probability density $p_t(\mathbf{x})$ at time $t$ converges to $\mathcal{N}(\mathbf{x}|\mathbf{0}, I)$ as $t \to \infty$.

**The backward diffusion process** is the conjugate of the forward process. While the forward process evolves $p_t$ toward $\mathcal{N}(\mathbf{0}, I)$, the backward process reverses this evolution, restoring $\mathcal{N}(\mathbf{0}, I)$ to $p_t$. To derive it, we know from previous section that Langevin dynamics Eq.22 acts as an "identity" operation on a distribution. Thus, the composition of forward and backward processes, at time $t$, must yield the Langevin dynamics for $p_t(\mathbf{x})$.

To formalize this, consider the Langevin dynamics for $p_t(\mathbf{x})$ with a distinct time variable $\tau$, distinguished from the forward diffusion time $t$. This dynamics can be decomposed into forward and backward components as follows:

$$\begin{aligned} d\mathbf{x}_\tau &= \mathbf{s}(\mathbf{x}_\tau, t)d\tau + \sqrt{2}\, d\mathbf{W}_\tau, \\ &= \underbrace{-\frac{1}{2}\mathbf{x}_\tau d\tau + d\mathbf{W}_\tau^{(1)}}_{\text{Forward}} + \underbrace{\left(\frac{1}{2}\mathbf{x}_t + \mathbf{s}(\mathbf{x}_\tau, t)\right)d\tau + d\mathbf{W}_\tau^{(2)}}_{\text{Backward}}, \end{aligned} \tag{25}$$

where $\mathbf{s}(\mathbf{x}, t) = \nabla_{\mathbf{x}} \log p_t(\mathbf{x})$ is the score function of $p_t(\mathbf{x})$. The "Forward" part corresponds to the forward diffusion process Eq.24, effectively increasing the forward diffusion time $t$ by $d\tau$, bringing the distribution to $p_{t+d\tau}(\mathbf{x})$. Since the forward and backward components combine to form an "identity" operation, the "Backward" part in Eq.25 must reverse the forward process—decreasing the forward diffusion time $t$ by $d\tau$ and restoring the distribution back to $p_t(\mathbf{x})$.

Now we can define the backward process according to the backward part in Eq.25, and a backward diffusion time $t'$ different from the forward diffusion time $t$:

$$d\mathbf{x}_{t'} = \left(\frac{1}{2}\mathbf{x}_{t'} + \mathbf{s}(\mathbf{x}_{t'}, t)\right)dt' + d\mathbf{W}_{t'}. \tag{26}$$

It remains to determine the relation between the forward diffusion time $t$ and backward diffusion time $t'$. Since $dt'$ is interpreted as "decrease" the forward diffusion time $t$, we have

$$dt = -dt' \tag{27}$$

which means the backward diffusion time is the inverse of the forward. To make $t'$ lies in the same range $[0, T]$ of the forward diffusion time, we define $t = T - t'$. In this notation, the backward diffusion process (Anderson, 1982) is

$$d\mathbf{x}_{t'} = \left(\frac{1}{2}\mathbf{x}_{t'} + \mathbf{s}(\mathbf{x}_{t'}, T - t')\right) dt' + d\mathbf{W}_{t'}, \tag{28}$$

in which $t' \in [0, T]$ is the backward time, $\mathbf{s}(\mathbf{x}, t) = \nabla_{\mathbf{x}} \log p_t(\mathbf{x})$ is the score function of the density of $\mathbf{x}_t$ in the forward process.

**Forward-Backward Duality** The forward and backward processes form a dual pair, advancing the time $t'$ means receding time $t$ by the same amount. We define the densities of $\mathbf{x}_t$ (forward) as $p_t(\mathbf{x})$, the densities of $\mathbf{x}_{t'}$ (backward) as $q_{t'}(\mathbf{x})$. If we initialize

$$q_0(\mathbf{x}) = p_T(\mathbf{x}), \tag{29}$$

then their evolution are related by

$$q_{t'}(\mathbf{x}) = p_{T-t'}(\mathbf{x}) \tag{30}$$

For large $T$, $p_T(\mathbf{x})$ converges to $\mathcal{N}(\mathbf{x}|\mathbf{0}, I)$. Thus, the backward process starts at $t' = 0$ with $\mathcal{N}(\mathbf{0}, I)$ and, after evolving to $t' = T$, generates samples from the data distribution:

$$q_T(\mathbf{x}) = p_0(\mathbf{x}). \tag{31}$$

This establishes an exact correspondence between the forward diffusion process and the backward diffusion process.

**Numerical Implementations** In practice, the forward OU process Eq.24 is numerically discretized into the variance-preserving (VP) form (Song et al., 2020c):

$$\mathbf{x}_i = \sqrt{1 - \beta_{i-1}}\mathbf{x}_{i-1} + \sqrt{\beta_{i-1}}\boldsymbol{\epsilon}_{i-1}, \tag{32}$$

where $i = 1, \cdots, n$ is the number of the time step, $\beta_i$ is the step size of each time step, $\mathbf{x}_i$ is image at $i$th time step with time $t_i = \sum_{j=0}^{i-1} \beta_j$, $\boldsymbol{\epsilon}_i$ is standard Gaussian random variable. The time step size usually takes the form $\beta_i = \frac{i(b_2 - b_1)}{n-1} + b_1$ where $b_1 = 10^{-4}$ and $b_2 = 0.02$. Note that our interpretation of $\beta$ differs from that in (Song et al., 2020c), treating $\beta$ as a varying time-step size to solve the autonomous SDE Eq.24 instead of a time-dependent SDE. Our interpretation holds as long as every $\beta_i^2$ is negligible and greatly simplifies future analysis. The discretized OU process Eq.32 adds a small amount of Gaussian noise to the image at each time step $i$, gradually contaminating the image until $\mathbf{x}_n \sim \mathcal{N}(\mathbf{0}, I)$.

Training a DDPM aims to recover the original image $x_0$ from one of its contaminated versions $x_i$. In this case Eq.32 could be rewritten into the **forward diffusion process**

$$\mathbf{x}_i = \sqrt{\bar{\alpha}_i}\mathbf{x}_0 + \sqrt{1 - \bar{\alpha}_i}\bar{\boldsymbol{\epsilon}}_i; \quad 1 \le i \le n, \tag{33}$$

where $\bar{\alpha}_i = \prod_{j=0}^{i-1}(1 - \beta_j)$ is the weight of contamination and $\bar{\boldsymbol{\epsilon}}_i$ is a standard Gaussian random noise to be removed.

An useful property we shall exploit later is that for **infinitesimal** time steps $\beta$, the contamination weight $\bar{\alpha}_i$ is the exponential of the diffusion time $t_i$

$$\lim_{\max_j \beta_j \to 0} \bar{\alpha}_i \to e^{-t_i}. \tag{34}$$

The **backward diffusion process** is used to sample from the DDPM by removing the noise of an image step by step. It is the time reversed version of the OU process, starting at $x_{0'} \sim \mathcal{N}(\mathbf{x}|\mathbf{0}, I)$, using the reverse of the OU process Eq.28. In practice, the backward diffusion process is discretized into

$$\mathbf{x}_{i'+1} = \frac{\mathbf{x}_{i'} + \mathbf{s}(\mathbf{x}_{i'}, T - t'_{i'})\beta_{n-i'}}{\sqrt{1 - \beta_{n-i'}}} + \sqrt{\beta_{n-i'}}\boldsymbol{\epsilon}_{i'}, \tag{35}$$

where $i' = 0, \cdots, n$ is the number of the backward time step, $\mathbf{x}_{i'}$ is image at $i'$th backward time step with time $t'_{i'} = \sum_{j=0}^{i'-1} \beta_{n-1-j} = T - t_{n-i'}$. This discretization is consistent with Eq.28 as long as $\beta_i^2$ are negligible. The score function $\mathbf{s}(\mathbf{x}_{i'}, T - t'_{i'})$ is generally modeled by a neural network and trained with a denoising objective.

**Training the score function** requires a training objective. We will show that the score function could be trained with a denoising objective.

DDPM is trained to removes the noise $\bar{\epsilon}_i$ from $\mathbf{x}_i$ in Eq.33, by training a denoising neural network $\epsilon_\theta(\mathbf{x}, t_i)$ to predict and remove the noise $\bar{\epsilon}_i$. This means that DDPM minimizes the **denoising objective** (Ho et al., 2020):

$$L_{denoise}(\epsilon_\theta) = \frac{1}{n} \sum_{i=1}^n \mathbf{E}_{\mathbf{x}_0 \sim p_0(\mathbf{x})} \mathbf{E}_{\bar{\epsilon}_i \sim \mathcal{N}(\mathbf{0}, I)} \|\bar{\epsilon}_i - \epsilon_\theta(\mathbf{x}_i, t_i)\|_2^2. \tag{36}$$

Now we show that $\epsilon_\theta$ trained with the above objective is proportional to the score function $\mathbf{s}$. Note that the Eq.33 tells us that the distribution of $\mathbf{x}_i$ given $\mathbf{x}_0$ is a Gaussian distribution

$$p(\mathbf{x}_i | \mathbf{x}_0) = \mathcal{N}(\mathbf{x}_i | \sqrt{\bar{\alpha}_i} \mathbf{x}_0, (1 - \bar{\alpha}_i)I), \tag{37}$$

and the noise $\bar{\epsilon}_i$ in Eq.33 is directly proportional to the score function

$$\bar{\epsilon}_i = -\sqrt{1 - \bar{\alpha}_i} \mathbf{s}(\mathbf{x}_i | \mathbf{x}_0, t_i), \tag{38}$$

where $\mathbf{s}(\mathbf{x}_i | \mathbf{x}_0, t_i) = \nabla_{\mathbf{x}_i} \log p(\mathbf{x}_i | \mathbf{x}_0)$ is the score of the conditional probability density $p(\mathbf{x}_i | \mathbf{x}_0)$ at $\mathbf{x}_i$.

The Eq.38 is an important property. It tells us that the noise $\bar{\epsilon}_i$ is directly related to a conditional score function. This conditional score function is connected to the score function $\mathbf{s}(\mathbf{x}, t)$ through the following equation:

$$\mathbf{E}_{\mathbf{x}_i \sim p_{t_i}(\mathbf{x})} f(\mathbf{x}_i) \mathbf{s}(\mathbf{x}, t_i) = \mathbf{E}_{\mathbf{x}_0 \sim p_0(\mathbf{x})} \mathbf{E}_{\mathbf{x}_i \sim p(\mathbf{x}_i | \mathbf{x}_0)} f(\mathbf{x}_i) \mathbf{s}(\mathbf{x}_i | \mathbf{x}_0) \tag{39}$$

where $f$ is an arbitrary function and $\mathbf{s}(\mathbf{x}, t) = \nabla_{\mathbf{x}} \log p_t(\mathbf{x})$ is the score function of the probability density of $\mathbf{x}_t$.

Substituting Eq.38 into Eq.36 and utilizing Eq.39, we could derive that Eq.36 is equivalent to a denoising score matching objective

$$L_{denoise}(\epsilon_\theta) = \frac{1}{n} \sum_{i=1}^n \mathbf{E}_{\mathbf{x}_i \sim p_{t_i}(\mathbf{x})} \|\sqrt{1 - \bar{\alpha}_i} \mathbf{s}(\mathbf{x}_i, t_i) + \epsilon_\theta(\mathbf{x}_i, t_i)\|_2^2, \tag{40}$$

This objectives says that the denoising neural network $\epsilon_\theta(\mathbf{x}, t_i)$ is trained to approximate a scaled score function $\epsilon(\mathbf{x}, t_i)$ (Yang et al., 2022)

$$\epsilon_\theta(\mathbf{x}, t_i) \approx -\sqrt{1 - \bar{\alpha}_i} \mathbf{s}(\mathbf{x}, t_i). \tag{41}$$

Therefore the denoising neural network is actually a scaled estimate of the score function $\mathbf{s}(\mathbf{x}, t)$, hence could be inserted into the backward sampling process Eq.35 to generate images.

## D   The ODE Based Backward Diffusion Process

The backward diffusion process Eq.26 is not the only reverse process for the forward process Eq.24. We can derive a deterministic ordinary differential equation (ODE) as an alternative, removing the stochastic term $d\mathbf{W}$ in the reverse process.

To obtain this ODE reverse process, consider the Langevin dynamics Eq.25 with a rescaled time $(d\tau \to \frac{1}{2} d\tau)$:

$$
\begin{aligned}
d\mathbf{x}_\tau &= \frac{1}{2} \mathbf{s}(\mathbf{x}_\tau, t) d\tau + d\mathbf{W}_\tau, \\
&= \underbrace{-\frac{1}{2} \mathbf{x}_\tau d\tau + d\mathbf{W}_\tau}_{\text{Forward}} + \underbrace{\frac{1}{2} \mathbf{x}_\tau d\tau + \frac{1}{2} \mathbf{s}(\mathbf{x}_\tau, t) d\tau}_{\text{Backward}},
\end{aligned}
\tag{42}
$$

Following the same logic used to derive the backward diffusion process Eq.28, we extract from this splitting the backward ODE (known as the probability flow ODE (Song et al., 2020c)):

$$d\mathbf{x}_{t'} = \frac{1}{2}\left(\mathbf{x}_{t'} + \mathbf{s}(\mathbf{x}, T - t')\right)dt', \tag{43}$$

where $t' \in [0, T]$ is backward time, and $\mathbf{s}(\mathbf{x}, t) = \nabla_{\mathbf{x}_t} \log p_t(\mathbf{x})$ is the score function of the density of $\mathbf{x}_t$ in the forward process. This ODE maintains the same forward-backward duality as the SDE reverse process Eq.28.

Since the ODE is deterministic, it enables faster sampling than the SDE version. Established ODE solvers—such as higher-order methods and exponential integrators—can further reduce computational steps while maintaining accuracy.

## E  Three Notations of Diffusion Models

In this section, we discuss three common formulations of diffusion models: variance-preserving (VP), variance-exploding (VE), and rectified flow (RF). We demonstrate their mathematical equivalence and show how they can be transformed into one another.

To simplify notation, we now use continuous time $t$ and its corresponding state $\mathbf{x}_t$ (as in Eq.24), rather than discrete notations like $t_i$ and $\mathbf{x}_i$.

**Variance Preserving (VP)**  The DDPMs introduced in the previous section are called 'variance-preserving' models. This name originates from the forward process Eq.33: if the clean images $\mathbf{x}_0$ are normalized such that $\mathrm{Cov}(\mathbf{x}_0, \mathbf{x}_0) = I$, then this covariance is preserved at any time $t_i$, with $\mathrm{Cov}(\mathbf{x}_i, \mathbf{x}_i) = I$.

The forward diffusion process Eq.33 in continuous time $t$ is:

$$\mathbf{x}_t = \sqrt{\bar{\alpha}_t}\mathbf{x}_0 + \sqrt{1 - \bar{\alpha}_t}\bar{\boldsymbol{\epsilon}}_t, \tag{44}$$

where $\bar{\alpha}_t = e^{-t}$ (from Eq.34) and $\bar{\boldsymbol{\epsilon}}_t \sim \mathcal{N}(0, \mathbf{I})$.

The continuous-time processes are:

- **Forward SDE** (Ornstein-Uhlenbeck process):

$$d\mathbf{x}_t = -\frac{1}{2}\mathbf{x}_t dt + d\mathbf{W}_t \tag{45}$$

- **Backward ODE** (Probability flow):

$$d\mathbf{x}_{t'} = \frac{1}{2}\left(\mathbf{x}_{t'} + \mathbf{s}(\mathbf{x}_{t'}, T - t')\right)dt', \tag{46}$$

  where $t' \in [0, T]$ is reversed time, and the score function $\mathbf{s}(\mathbf{x}, t) = \nabla_{\mathbf{x}} \log p_t(\mathbf{x})$ is learned via the denoising objective Eq.36 and Eq.41.

While we previously trained the denoising network $\boldsymbol{\epsilon}_\theta$ using objective Eq.36 (related to the score function via Eq.41), we can alternatively model the score function $\mathbf{s}_\theta$ directly. By substituting $\boldsymbol{\epsilon}_\theta$ with $\mathbf{s}_\theta$ and dropping the $\sqrt{1 - \bar{\alpha}_t}$ scaling factor, we obtain the equivalent score-based objective:

$$L_{score}(\mathbf{s}_\theta) = \mathbf{E}_{t \sim \mathcal{U}[0,1]}\mathbf{E}_{\mathbf{x}_0 \sim p_0(\mathbf{x})}\mathbf{E}_{\bar{\boldsymbol{\epsilon}}_t \sim \mathcal{N}(\mathbf{0}, I)}\left\|\frac{\bar{\boldsymbol{\epsilon}}_t}{\sqrt{1 - \bar{\alpha}_t}} + \mathbf{s}_\theta(\mathbf{x}_t, t)\right\|_2^2, \tag{47}$$

where $\mathbf{x}_t$ follows Eq.44. This represents an equivalent but reweighted version of the original denoising objective Eq.36.

**Variance Exploding (VE)**   The variance exploding formulation provides an alternative to variance preserving. Define:

$$\sigma = \sqrt{\frac{1 - \bar{\alpha}_t}{\bar{\alpha}_t}}; \quad \sigma' = \sqrt{\frac{1 - \bar{\alpha}_{T-t'}}{\bar{\alpha}_{T-t'}}}; \quad \mathbf{z}_\sigma = \frac{\mathbf{x}_t}{\sqrt{\bar{\alpha}_t}}; \quad \mathbf{z}_{\sigma'} = \frac{\mathbf{x}_{t'}}{\sqrt{\bar{\alpha}_{T-t'}}}; \quad \boldsymbol{\epsilon}(\mathbf{z}_\sigma, \sigma) = -\sqrt{1 - \bar{\alpha}_t}\mathbf{s}(\mathbf{x}_t, t), \quad (48)$$

Rewriting the VP forward Eq.44 in VE notation yields:

$$\mathbf{z}_\sigma = \mathbf{z}_0 + \sigma\bar{\boldsymbol{\epsilon}}_\sigma, \tag{49}$$

where $\mathbf{z}_0$ is the clean image corrupted by noise of magnitude $\sigma$.

The continuous-time processes become:

- **Forward SDE** (from Eq.45):

$$d\mathbf{z}_\sigma = \sqrt{2\sigma}d\mathbf{W}_\sigma, \quad \sigma \in \left[0, \sqrt{\frac{1-\bar{\alpha}_T}{\bar{\alpha}_T}}\right] \tag{50}$$

- **Backward ODE** (from Eq.46):

$$d\mathbf{z}_{\sigma'} = \boldsymbol{\epsilon}(\mathbf{z}_{\sigma'}, \sigma')d\sigma', \quad \sigma' \in \left[\sqrt{\frac{1-\bar{\alpha}_T}{\bar{\alpha}_T}}, 0\right] \tag{51}$$

To directly model $\boldsymbol{\epsilon}_\theta(\mathbf{z}, \sigma)$, we adapt the denoising objective Eq.36 to VE coordinates by replacing $\mathbf{x}_t$ with $\mathbf{z}_\sigma$:

$$L_{denoise}(\boldsymbol{\epsilon}_\theta) = \mathbf{E}_{\sigma \sim \mathcal{U}[0,\sigma_{max}]}\mathbf{E}_{\mathbf{z}_0 \sim p_0(\mathbf{x})}\mathbf{E}_{\bar{\boldsymbol{\epsilon}}_\sigma \sim \mathcal{N}(\mathbf{0},I)}\|\bar{\boldsymbol{\epsilon}}_\sigma - \boldsymbol{\epsilon}_\theta(\mathbf{z}_\sigma, \sigma)\|_2^2, \tag{52}$$

where $\sigma_{max} = \sqrt{(1 - \bar{\alpha}_T)/\bar{\alpha}_T}$ and $\mathbf{z}_\sigma$ follows Eq.49. This preserves the denoising objective's structure while operating in VE space.

**Rectified Flow (RF)**   While often presented as a distinct framework from DDPMs, rectified flows are mathematically equivalent (Gao et al., 2025) to DDPMs. We now provide a much simpler proof via the transformations:

$$s = \frac{\sigma}{1+\sigma}; \quad s' = \frac{\sigma'}{1+\sigma'}; \quad \mathbf{r}_s = \frac{\mathbf{z}_\sigma}{1+\sigma}; \quad \mathbf{r}_{s'} = \frac{\mathbf{z}_{\sigma'}}{1+\sigma'}; \quad \mathbf{v}(\mathbf{r}_s, s) = \frac{\boldsymbol{\epsilon}(\mathbf{z}_\sigma, \sigma) - \mathbf{r}_s}{1-s} \tag{53}$$

Rewriting the VE forward process Eq.49 in RF coordinates yields:

$$\mathbf{r}_s = (1-s)\mathbf{r}_0 + s\bar{\boldsymbol{\epsilon}}_s, \tag{54}$$

which linearly interpolates between clean data ($\mathbf{r}_0$) and noise.

The continuous-time dynamics become:

- **Forward SDE (from Eq.50)**:

$$d\mathbf{r}_s = -\frac{\mathbf{r}_s}{1-s}ds + \sqrt{\frac{2s}{1-s}}d\mathbf{W}_s, \quad s \in [0,1] \tag{55}$$

- **Backward ODE (from Eq.51)**:

$$d\mathbf{r}_{s'} = \mathbf{v}(\mathbf{r}_{s'}, s')ds', \quad s' \in [1,0] \tag{56}$$

To directly model $\mathbf{v}_\theta(\mathbf{r}_{s'}, s')$, we transform the denoising objective Eq.52 by substituting $\boldsymbol{\epsilon}_\theta$ with $\mathbf{v}_\theta$ and inserting the RF forward process Eq.54 into Eq.52, while removing a constant scaling factor $(1 - s)$. This yields the flow matching objective:

$$L_{flow}(\mathbf{v}_\theta) = \mathbf{E}_{s\sim\mathcal{U}[0,1]}\mathbf{E}_{\mathbf{r}_0\sim p_0(\mathbf{x})}\mathbf{E}_{\bar{\boldsymbol{\epsilon}}_s\sim\mathcal{N}(\mathbf{0},I)}\|\bar{\boldsymbol{\epsilon}}_s - \mathbf{r}_0 - \mathbf{v}_\theta(\mathbf{r}_s,s)\|_2^2, \tag{57}$$

where $\mathbf{r}_s$ follows Eq.54. This represents a re-weighted equivalent of the denoising objective Eq.36, interpreted in the flow matching framework where $\bar{\boldsymbol{\epsilon}}$ corresponds to the endpoint $\mathbf{r}_1$ and $\mathbf{v}_\theta$ models the velocity field transporting $\mathbf{r}_0$ to $\mathbf{r}_1$.

In summary, the three notations (VP, VE, and RF) are mutually transformable through the mappings defined in Eq.48 and Eq.53. This equivalence enables a practical LanPaint implementation strategy: we can design LanPaint using any single notation (such as VP) and automatically extend it to other frameworks by applying these transformations.

## F   Stationary Distribution of Langevin Dynamics with the BiG score

In this appendix, we prove that the BiG score Langevin Dynamics defined in Eq.13 converges to the target distribution

$$\pi_t(\mathbf{x}, \mathbf{y}) \propto p_t(\mathbf{x} \mid \mathbf{y}) \frac{p_t(\mathbf{y} \mid \mathbf{y}_o)^{1+\lambda}}{p_t(\mathbf{y})^\lambda} + o(\sqrt{1 - \bar{\alpha}_t}). \tag{58}$$

with a negligible deviation as $t \to 0$. The joint dynamics of $\mathbf{x}_t$ and $\mathbf{y}_t$ are governed by the following Langevin dynamics:

$$\begin{aligned} d\mathbf{x}_t &= \mathbf{s_x}(\mathbf{x}_t, \mathbf{y}_t, t)\, d\tau + \sqrt{2}\, d\mathbf{W}_\tau^{\mathbf{x}}, \\ d\mathbf{y}_t &= \mathbf{g}_\lambda(\mathbf{x}_t, \mathbf{y}_t, t)\, d\tau + \sqrt{2}\, d\mathbf{W}_\tau^{\mathbf{y}}, \end{aligned} \tag{59}$$

where the drift term $\mathbf{g}_\lambda$ is defined as

$$\mathbf{g}_\lambda(\mathbf{x}, \mathbf{y}, t) = -\left( (1+\lambda) \frac{\mathbf{y} - \sqrt{\bar{\alpha}_t}\, \mathbf{y}_o}{1 - \bar{\alpha}_t} + \lambda\, \mathbf{s_y}(\mathbf{x}, \mathbf{y}, t) \right). \tag{60}$$

### F.1   Idealized SDE for the Target Distribution

To establish the convergence of BiG score, we first consider an idealized Langevin dynamics whose invariant distribution is the exact target distribution

$$\pi^*(\mathbf{x}_t, \mathbf{y}_t) \propto p(\mathbf{x}_t \mid \mathbf{y}_t) \frac{p(\mathbf{y}_t \mid \mathbf{y}_o)^{1+\lambda}}{p_t(\mathbf{y})^\lambda}. \tag{61}$$

The score function used in this idealized Langevin dynamics is given by:

$$\mathbf{s}^*(\mathbf{x}, \mathbf{y}, t) = \nabla_{\mathbf{x},\mathbf{y}} \left[ \log p_t(\mathbf{x} \mid \mathbf{y}) + (1+\lambda) \log p_t(\mathbf{y} \mid \mathbf{y}_o) - \lambda \log p_t(\mathbf{y}) \right]. \tag{62}$$

Expanding this and note that $p_t(\mathbf{y}) = \frac{p_t(\mathbf{x},\mathbf{y})}{p_t(\mathbf{x}|\mathbf{y})}$, we obtain:

$$\begin{aligned} d\mathbf{x}_t &= \mathbf{s_x}^*(\mathbf{x}_t, \mathbf{y}_t, t)\, d\tau + \sqrt{2}\, d\mathbf{W}_\tau^{\mathbf{x}}, \\ d\mathbf{y}_t &= \mathbf{s_y}^*(\mathbf{x}_t, \mathbf{y}_t, t)\, d\tau + \sqrt{2}\, d\mathbf{W}_\tau^{\mathbf{y}}, \end{aligned} \tag{63}$$

where
$$\begin{aligned} \mathbf{s_x}^*(\mathbf{x}, \mathbf{y}, t) &= \mathbf{s_x}(\mathbf{x}, \mathbf{y}, t), \\ \mathbf{s_y}^*(\mathbf{x}, \mathbf{y}, t) &= (1+\lambda)\, \nabla_{\mathbf{y}} \log p(\mathbf{x} \mid \mathbf{y}, t) + (1+\lambda)\, \nabla_{\mathbf{y}} \log p(\mathbf{y} \mid \mathbf{y}_o, t) - \lambda\, \nabla_{\mathbf{y}} \log p(\mathbf{x}, \mathbf{y}, t). \end{aligned} \tag{64}$$

Using the fact that $p(\mathbf{y}_t \mid \mathbf{y}_o) = \mathcal{N}(\mathbf{y}_t \mid \sqrt{\bar{\alpha}_t}\, \mathbf{y}_o, (1 - \bar{\alpha}_t)\mathbf{I})$, we simplify $\mathbf{s_y}^*$ to:

$$\mathbf{s_y}^* = (1+\lambda)\, \nabla_{\mathbf{y}_t} \log p(\mathbf{x}_t \mid \mathbf{y}_t) - (1+\lambda) \frac{\mathbf{y}_t - \sqrt{\bar{\alpha}_t}\, \mathbf{y}_o}{1 - \bar{\alpha}_t} - \lambda\, \nabla_{\mathbf{y}_t} \log p(\mathbf{x}_t, \mathbf{y}_t). \tag{65}$$

### F.2 Comparison with BiG score Drift Term

To connect the idealized SDE with the BiG score dynamics, we analyze the relationship between $\mathbf{s}_\mathbf{y}^*$ and $\mathbf{g}_\lambda$. Define the following quantities:

$$
\begin{aligned}
r_t &= \mathbb{E}_{p(\mathbf{x}_t, \mathbf{y}_t)} \left\| \frac{\mathbf{y}_t - \sqrt{\bar{\alpha}_t}\, \mathbf{y}_o}{1 - \bar{\alpha}_t} \right\|_2, \\
s_t &= \mathbb{E}_{p(\mathbf{x}_t, \mathbf{y}_t)} \left\| \nabla_{\mathbf{y}_t} \log p(\mathbf{x}_t \mid \mathbf{y}_t) \right\|_2, \\
\mathbf{s}_{\text{cond}} &= \frac{r_t}{s_t} \nabla_{\mathbf{y}_t} \log p(\mathbf{x}_t \mid \mathbf{y}_t).
\end{aligned}
\tag{66}
$$

Here, $\mathbf{s}_{\text{cond}}$ is a rescaled version of $\nabla_{\mathbf{y}_t} \log p(\mathbf{x}_t \mid \mathbf{y}_t)$ that matches the order of magnitude of $\frac{\mathbf{y}_t - \sqrt{\bar{\alpha}_t}\, \mathbf{y}_o}{1 - \bar{\alpha}_t}$. Substituting these definitions into $\mathbf{s}_\mathbf{y}^*$, we obtain:

$$
\mathbf{s}_\mathbf{y}^* = (1 + \lambda) \frac{s_t}{r_t} \mathbf{s}_{\text{cond}} + \mathbf{g}_\lambda(\mathbf{x}_t, \mathbf{y}_t, t).
\tag{67}
$$

Thus, the idealized score function $\mathbf{s}^*(\mathbf{x}, \mathbf{y}, t)$ can be expressed as:

$$
\mathbf{s}^*(\mathbf{x}, \mathbf{y}, t) = \Big( \mathbf{s}_\mathbf{x}(\mathbf{x}, \mathbf{y}, t),\ \mathbf{g}_\lambda(\mathbf{x}, \mathbf{y}, t) \Big) + \frac{s_t}{r_t} \big( \mathbf{0},\ \mathbf{s}_{\text{cond}} \big).
\tag{68}
$$

### F.3 Perturbation Between Ideal and BiG score Scores

To quantify the deviation between the idealized score $\mathbf{s}_\mathbf{y}^*$ and the BiG score score $\mathbf{g}_\lambda$, we analyze the scaling relationship between the terms $\frac{s_t}{r_t}$ and $\sqrt{1 - \bar{\alpha}_t}$. Recall that $s_t$ and $r_t$ are defined as:

$$
\begin{aligned}
r(t) &= \mathbb{E}_{p_t(\mathbf{x}, \mathbf{y})} \left\| \frac{\mathbf{y} - \sqrt{\bar{\alpha}_t}\, \mathbf{y}_o}{1 - \bar{\alpha}_t} \right\|_2, \\
s(t) &= \mathbb{E}_{p_t(\mathbf{x}, \mathbf{y})} \left\| \nabla_\mathbf{y} \log p_t(\mathbf{x} \mid \mathbf{y}) \right\|_2.
\end{aligned}
\tag{69}
$$

From the relationship between the score function and the noise term in diffusion models (see Eq.41), we have:

$$
\boldsymbol{\epsilon}(\mathbf{x}, t) = -\sqrt{1 - \bar{\alpha}_t}\, \mathbf{s}(\mathbf{x}, t),
\tag{70}
$$

where $\bar{\alpha}_t = e^{-t}$. This implies that the score function $\mathbf{s}(\mathbf{x}, t)$ can be expressed as:

$$
\mathbf{s}(\mathbf{x}, t) = -\frac{\boldsymbol{\epsilon}(\mathbf{x}, t)}{\sqrt{1 - \bar{\alpha}_t}}.
\tag{71}
$$

To proceed, we make the following assumption about the noise term $\boldsymbol{\epsilon}(\mathbf{x}, t)$:

**Assumption F.1.** The expected $L_2$ norm of the noise term $\boldsymbol{\epsilon}(\mathbf{x}, t)$ is a positive bounded value, i.e., there exists a constant $C > 0$ such that

$$
\mathbb{E}_{p(\mathbf{x}, t)} \left\| \boldsymbol{\epsilon}(\mathbf{x}, t) \right\|_2 = C.
\tag{72}
$$

Under Assumption F.1, we can derive the scaling behavior of $s_t$ and $r_t$:

1. Scaling of $s_t$: Since $\nabla_\mathbf{y} \log p_t(\mathbf{x} \mid \mathbf{y})$ is a score function, it follows the same scaling as $\mathbf{s}(\mathbf{x}, t)$. Thus,

$$
s(t) = \mathbb{E}_{p_t(\mathbf{x}, \mathbf{y})} \left\| \nabla_\mathbf{y} \log p_t(\mathbf{x} \mid \mathbf{y}) \right\|_2 \sim \frac{C}{\sqrt{1 - \bar{\alpha}_t}}.
\tag{73}
$$

2. Scaling of $r_t$: The term $\frac{\mathbf{y} - \sqrt{\bar{\alpha}_t}\, \mathbf{y}_o}{1 - \bar{\alpha}_t}$ represents the deviation of $\mathbf{y}$ from its conditional mean. For small $1 - \bar{\alpha}_t$, this scales as:

$$
r(t) = \mathbb{E}_{p_t(\mathbf{x}, \mathbf{y})} \left\| \frac{\mathbf{y} - \sqrt{\bar{\alpha}_t}\, \mathbf{y}_o}{1 - \bar{\alpha}_t} \right\|_2 \sim \frac{C'}{1 - \bar{\alpha}_t},
\tag{74}
$$

where $C' > 0$ is a constant proportional to the standard deviation of $\mathbf{y}$.

3. Ratio $\frac{s_t}{r_t}$: Combining the scaling behaviors of $s_t$ and $r_t$, we obtain:

$$\frac{s_t}{r_t} \sim \frac{C/\sqrt{1 - \bar{\alpha}_t}}{C'/(1 - \bar{\alpha}_t)} = \frac{C}{C'}\sqrt{1 - \bar{\alpha}_t}. \tag{75}$$

Thus, $\frac{s_t}{r_t}$ scales as $\mathcal{O}(\sqrt{1 - \bar{\alpha}_t})$.

Now we have shown that the idealized Langevin dynamics Eq.63 and the BiG score Langevin dynamics Eq.59 differ only by an $\mathcal{O}(\sqrt{1 - \bar{\alpha}_t})$ perturbation. We also provide numerical verification of this analysis in Fig.9.

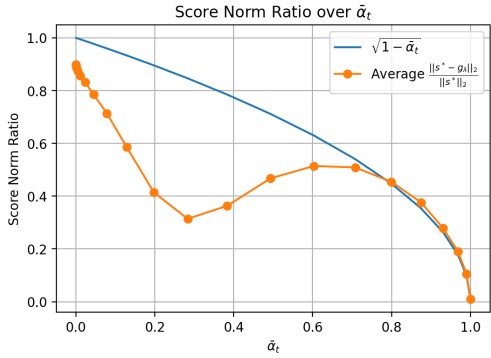

Figure 9: We analyze the average norm ratio between the ideal score $\mathbf{s}_{\mathbf{y}}^*$ (Eq.65) and the component discarded by the BiG score, $\mathbf{s}_{\mathbf{y}}^* - \mathbf{g}_\lambda$, relative to $\bar{\alpha}_t$, in the conditional Gaussian case (Section 5.1). We focus on the right part of the image when $\bar{\alpha}_t$ approaches 1 (i.e., $t \to 0$), which determines the distribution of generated clean image. This ratio decreases at the same rate as $\sqrt{1 - \bar{\alpha}_t}$, indicating that the BiG score $\mathbf{g}_\lambda$ closely approximates the ideal score $\mathbf{s}_{\mathbf{y}}^*$ with negligible error as $t \to 0$. This confirms that the error scales as $\mathcal{O}(\sqrt{1 - \bar{\alpha}_t})$, laying the foundation for Theorem 4.1.

### F.4 Fokker-Planck Equation Analysis

We now show that an $\mathcal{O}(\sqrt{1 - \bar{\alpha}_t})$ deviation from the idealized score function translates to an $\mathcal{O}(\sqrt{1 - \bar{\alpha}_t})$ deviation in the stationary distribution of Langevin dynamics.

The Fokker–Planck equation Eq.147 describes the time evolution of the probability density $\rho(\mathbf{z}, \tau)$ associated with a stochastic process governed by a stochastic differential equation (SDE). For a general SDE of the form:

$$dz_i = h_i(\mathbf{z})\,d\tau + \gamma_{ij}(\mathbf{z})\,dW_j, \tag{76}$$

the corresponding Fokker–Planck equation can be written in operator form as:

$$\frac{\partial \rho(\mathbf{z}, \tau)}{\partial \tau} = \mathcal{L}\rho(\mathbf{z}, \tau), \tag{77}$$

where $\mathcal{L}$ is the Fokker-Planck operator, defined as:

$$\mathcal{L} = -\nabla \cdot \left[\mathbf{h}(\mathbf{z}) \cdot\right] + \frac{1}{2}\nabla \cdot \left[\mathbf{D}(\mathbf{z})\nabla\cdot\right]. \tag{78}$$

Here, $\mathbf{h}(\mathbf{z})$ is the drift vector, $\mathbf{D}(\mathbf{z}) = \gamma(\mathbf{z})\gamma(\mathbf{z})^\top$ is the diffusion matrix, and $\nabla\cdot$ denotes the divergence operator.

#### F.4.1 Fokker-Planck Operator for BiG score

The BiG score dynamics are governed by the SDE Eq.59. The corresponding Fokker-Planck operator for the BiG score is:

$$\mathcal{L}_{\text{BiG score}} = -\nabla_{\mathbf{x}} \cdot \left[\mathbf{s}_{\mathbf{x}}(\mathbf{x}, \mathbf{y}, t) \cdot\right] - \nabla_{\mathbf{y}} \cdot \left[\mathbf{g}_\lambda(\mathbf{x}, \mathbf{y}, t) \cdot\right] + \nabla^2, \tag{79}$$

where $\nabla^2$ is the Laplacian operator.

The corresponding Fokker-Planck operator for the idealized SDE Eq.63 is:

$$\mathcal{L}_{\text{ideal}} = -\nabla_{\mathbf{x}_t} \cdot \left[ \mathbf{s}_{\mathbf{x}}(\mathbf{x}_t, \mathbf{y}_t, t) \cdot \right] - \nabla_{\mathbf{y}_t} \cdot \left[ \mathbf{s}_{\mathbf{y}}^*(\mathbf{x}_t, \mathbf{y}_t, t) \cdot \right] + \nabla^2. \tag{80}$$

### F.4.2 Deviation Between BiG score and Idealized SDE

The key difference between the BiG score and the idealized SDE lies in their score functions. Specifically, the score function $\mathbf{s}_{\mathbf{y}}^*$ for the idealized SDE can be expressed in terms of the BiG score score function $\mathbf{g}_\lambda$ as:

$$\mathbf{s}_{\mathbf{y}}^* = (1 + \lambda) \frac{s_t}{r_t} \mathbf{s}_{\text{cond}} + \mathbf{g}_\lambda, \tag{81}$$

where $\mathbf{s}_{\text{cond}} = \frac{r_t}{s_t} \nabla_{\mathbf{y}_t} \log p(\mathbf{x}_t \mid \mathbf{y}_t)$. Thus, the difference between the Fokker-Planck operators for the BiG score and the idealized SDE is:

$$\mathcal{L}_{\text{BiG score}} - \mathcal{L}_{\text{ideal}} = -\nabla_{\mathbf{y}_t} \cdot \left[ (1 + \lambda) \frac{s_t}{r_t} \mathbf{s}_{\text{cond}} \cdot \right]. \tag{82}$$

This additional term represents the perturbation between the BiG score and the idealized SDE.

### F.4.3 Analysis Using Dyson's Formula

To analyze the deviation of the solutions to the Fokker-Planck equations, we use Dyson's Formula (Evans & Morriss, 2008), a powerful tool in the study of perturbed differential equations. Dyson's Formula expresses the solution to a perturbed differential equation in terms of the unperturbed solution. Specifically, for a differential equation of the form

$$\frac{\partial \rho(\mathbf{z}, \tau)}{\partial \tau} = (\mathcal{L}_0 + \mathcal{L}_1) \rho(\mathbf{z}, \tau), \tag{83}$$

where $\mathcal{L}_0$ is the unperturbed operator and $\mathcal{L}_1$ is a perturbation, the solution can be written as

$$\rho(\mathbf{z}, \tau) = e^{(\mathcal{L}_0 + \mathcal{L}_1)\tau} \rho(\mathbf{z}, 0) = e^{\mathcal{L}_0 \tau} \rho(\mathbf{z}, 0) + \int_0^\tau e^{\mathcal{L}_0(\tau - s)} \mathcal{L}_1 \rho(\mathbf{z}, s) \, ds. \tag{84}$$

This formula allows us to express the solution to the perturbed Eq.83 as the sum of the unperturbed solution and a correction term due to the perturbation.

In our case, the Fokker-Planck equation for the BiG score can be viewed as a perturbation of the Fokker-Planck equation for the idealized SDE. Let $\rho_{\text{ideal}}(\mathbf{x}_t, \mathbf{y}_t, \tau)$ be the solution to the idealized Fokker-Planck equation:

$$\frac{\partial \rho_{\text{ideal}}}{\partial \tau} = \mathcal{L}_{\text{ideal}} \rho_{\text{ideal}}. \tag{85}$$

The solution to the BiG score Fokker-Planck equation can then be written using Dyson's Formula as:

$$\rho_{\text{BiG score}}(\mathbf{x}_t, \mathbf{y}_t, \tau) = \rho_{\text{ideal}}(\mathbf{x}_t, \mathbf{y}_t, \tau) + \int_0^\tau e^{\mathcal{L}_{\text{ideal}}(\tau - s)} \left( \mathcal{L}_{\text{BiG score}} - \mathcal{L}_{\text{ideal}} \right) \rho_{\text{BiG score}}(\mathbf{x}_t, \mathbf{y}_t, s) \, ds. \tag{86}$$

Substituting the perturbation term, we obtain:

$$\rho_{\text{BiG score}}(\mathbf{x}_t, \mathbf{y}_t, \tau) = \rho_{\text{ideal}}(\mathbf{x}_t, \mathbf{y}_t, \tau) - (1 + \lambda) \frac{s_t}{r_t} \int_0^\tau e^{\mathcal{L}_{\text{ideal}}(\tau - s)} \nabla_{\mathbf{y}_t} \cdot \left[ \mathbf{s}_{\text{cond}} \, \rho_{\text{BiG score}}(\mathbf{x}_t, \mathbf{y}_t, s) \right] \, ds. \tag{87}$$

From the analysis in Section F.3, we know that $\frac{s_t}{r_t} \sim \sqrt{1 - \bar{\alpha}_t}$. Thus, the deviation term is proportional to $\sqrt{1 - \bar{\alpha}_t}$.

### F.5 Conclusion

As $\tau \to \infty$, the idealized solution $\rho_{\text{ideal}}(\mathbf{x}_t, \mathbf{y}_t, \tau)$ converges exponentially to the stationary distribution $\pi^*(\mathbf{x}_t, \mathbf{y}_t)$, driven by the contractive nature of $\mathcal{L}_{\text{ideal}}$. The BiG score dynamics, governed by $\rho_{\text{BiG score}}$, deviate from $\pi^*$ by a term proportional to $\sqrt{1 - \bar{\alpha}_t}$. Thus, the BiG score dynamics converge to $\pi^*(\mathbf{x}_t, \mathbf{y}_t)$ with negligible error as $t \to 0$, controlled by the vanishing perturbation of the order $\sqrt{1 - \bar{\alpha}_t}$. This argument hence proves Theorem 4.1, as long as the term

$$P = \int_0^\tau e^{\mathcal{L}_{\text{ideal}}(\tau - s)} \nabla_{\mathbf{y}_t} \cdot \left[ \mathbf{s}_{\text{cond}} \, \rho_{\text{BiG score}}(\mathbf{x}_t, \mathbf{y}_t, s) \right] ds \tag{88}$$

remains bounded for all $\tau > 0$.

### F.6 Supp: P is bounded

This section aims to show that the term

$$P = \int_0^\tau e^{\mathcal{L}_{\text{ideal}}(\tau - s)} \nabla_{\mathbf{y}_t} \cdot \left[ \mathbf{s}_{\text{cond}} \, \rho_{\text{BiG score}}(\mathbf{x}_t, \mathbf{y}_t, s) \right] ds \tag{89}$$

is bounded.

**Assumptions**

1. Let $\mathbf{S}(\mathbf{x}_t, \mathbf{y}_t, s) = \mathbf{s}_{\text{cond}} \, \rho_{\text{BiG score}}(\mathbf{x}_t, \mathbf{y}_t, s)$, which is well-defined, bounded for all $\mathbf{x}_t$ and $s$, and vanishes as $\|\mathbf{y}_t\| \to \infty$.

2. The idealized Fokker-Planck equation (with generator $\mathcal{L}_{\text{ideal}}$), has a unique stationary solution for each initial condition and a finite relaxation time.

**Key Properties**

1. **Spectral Gap**: Assumption 2 implies that $\mathcal{L}_{\text{ideal}}$ has a spectral gap $L > 0$, or equivalently, $-L$ is the largest non-zero eigenvalue.

2. **Exponential Damping**: The term $e^{\mathcal{L}_{\text{ideal}}(\tau - s)} \nabla_{\mathbf{y}_t} \cdot \mathbf{S}$ represents the evolution of the initial condition $\nabla_{\mathbf{y}_t} \cdot \mathbf{S}$ from time 0 to $\tau - s$. Suppose

$$c = \lim_{\tau - s \to \infty} e^{\mathcal{L}_{\text{ideal}}(\tau - s)} \nabla_{\mathbf{y}_t} \cdot \mathbf{S} \tag{90}$$

Due to the spectral gap, this term behaves like $e^{-L(\tau - s)} + c$.

**Integral Evaluation**   If the integrand behaves like $e^{-L(\tau - s)}$ (suppose $c = 0$), then the integral

$$\int_0^\tau e^{-L(\tau - s)} \, ds = \frac{1 - e^{-L\tau}}{L} \tag{91}$$

is finite for $\tau > 0$, which completes the argument.

**Proof of $c = 0$**   Under **Assumption 1**, since $\mathbf{S}$ vanishes as $\|\mathbf{y}_t\| \to \infty$, the divergence theorem yields:

$$\int \nabla_{\mathbf{y}_t} \cdot \mathbf{S} \, d\mathbf{y}_t = 0. \tag{92}$$

Furthermore, because the Fokker-Planck operator $\mathcal{L}_{\text{ideal}}$ preserves probability mass, we have:

$$\int e^{\mathcal{L}_{\text{ideal}}(\tau - s)} \nabla_{\mathbf{y}_t} \cdot \mathbf{S} \, d\mathbf{y}_t = 0 \tag{93}$$

for all $\tau - s \geq 0$. As $\tau - s \to \infty$, this quantity must relax to zero—its unique stationary solution—without a constant term $C$.

**QED**   The integral is bounded:

$$|P| \leq \frac{\text{constant}}{L}.$$ (94)

## G   Fast Langevin Dynamics (FLD) with Momentum

In this section, we introduce how we design the solver for FLD step by step.

**The Original Langevin Dynamics**   Suppose we wish to solve the Langevin dynamics for LanPaint to perform a conditional sampling task:

$$d\mathbf{x} = \mathbf{s}(\mathbf{x})dt + \sqrt{2}dW_t,$$ (95)

where $\mathbf{s}(\mathbf{x}) = \nabla_{\mathbf{x}} \log p(\mathbf{x})$ is the score function modeled by the diffusion model. Our goal is to simulate the dynamics of $\mathbf{x}$ until it converges to its stationary distribution $p(\mathbf{x})$.

The simplest approach is to use the Euler-Maruyama scheme:

$$\mathbf{x}(t + \Delta t) = \mathbf{x}(t) + \mathbf{s}(\mathbf{x}(0))\Delta t + \sqrt{2\Delta t}\xi,$$ (96)

where $\xi$ is a standard Gaussian noise. This is a first-order scheme with a total numerical error scaling as $\mathcal{O}(\Delta t)$. However, this method has two drawbacks:

- **Slow convergence**: It requires many time steps for $\mathbf{x}$ to reach the stationary distribution unless we use large $\Delta t$.

- **White Noite Issue**: The numerical error scales with $\Delta t$. If $\Delta t$ is large, it will add too much noise to $\mathbf{x}$ within one step, manifesting as either visible white noise (pixel space) or blurriness (latent space) in the generated images.

To address these issues, we aim to:

- **Accelerate convergence** to reduce the number of required steps.

- **Use more accurate numerical scheme** to suppress the white noise artifacts.

**The Underdamped Langevin Dynamics (ULD)**   To accelerate convergence, we adopt the underdamped Langevin dynamics, which introduces momentum to the original Langevin dynamics:

$$\begin{aligned} d\mathbf{x} &= \frac{1}{m}\mathbf{v}\, dt, \\ d\mathbf{v} &= -\frac{\gamma}{m}\mathbf{v}\, dt + \mathbf{s}(\mathbf{x})\, dt + \sqrt{2\gamma}\, dW_{\mathbf{v}}, \end{aligned}$$ (97)

where $\gamma$ is the friction coefficient, $m$ is the mass, and $\mathbf{v}$ is an auxiliary momentum variable. The stationary distribution of this system is:

$$\rho(\mathbf{x}, \mathbf{v}) = p(\mathbf{x})\, \mathcal{N}(\mathbf{v}|\mathbf{0}, m\mathbf{I}).$$ (98)

Momentum is well-known to accelerate convergence in optimization problems, and the same holds for Langevin dynamics. While ULD provides faster convergence, it has two key drawbacks:

- **Interpretability**: The parameters of $\gamma$ and $m$ is non-intuitive and challenging to understand.

- **Momentum cannot be switched off**: There exists no parameter choice ($\gamma$ or $m$) that recovers the original Langevin dynamics. This makes it difficult to isolate whether the acceleration stems from momentum effects or other factors (e.g., larger effective step sizes) in comparative studies.

**The Fast Langevin Dynamics**  To address these limitations, we reformulate the ULD by introducing the transformations: $\mathbf{q} = \frac{\gamma}{m}\mathbf{v}$, $\tau = \frac{t}{\gamma}$, and $\Gamma = \frac{\gamma^2}{m}$, yielding the following system:

$$
\begin{aligned}
d\mathbf{x} &= \mathbf{q}\,d\tau \\
d\mathbf{q} &= \Gamma\left(-\mathbf{q}\,d\tau + \mathbf{s}(\mathbf{x})\,d\tau + \sqrt{2}\,dW_\tau\right)
\end{aligned}
\tag{99}
$$

This formulation provides key advantages that resolve our previous concerns:

- **Improved interpretability**: The $(\mathbf{s}(\mathbf{x})\,d\tau + \sqrt{2}\,dW_\tau)$ term directly correspond to the original dynamics, with $\mathbf{q}$ representing an exponentially weighted moving average of these terms with decay rate $\Gamma$.

- **Exact recovery of original dynamics**: By taking $\Gamma \to \infty$, we recover the original Langevin dynamics exactly, as the momentum equation reduces to:

$$
\mathbf{q}\,d\tau = \mathbf{s}(\mathbf{x})\,d\tau + \sqrt{2}\,dW_\tau
\tag{100}
$$

  This allows direct comparison between the momentum and non-momentum cases.

- **Parameter reduction**: The reformulation combines the two original parameters $(\gamma,\ m)$ into a single parameter $\Gamma$, revealing that one parameter is redundant. This allows us to set $m = 1$ in the original ULD without loss of generality, simplifying both analysis and implementation.

It remains to design an accurate numerical scheme for this dynamics. For LanPaint, we aim to perform conditional sampling with minimal computational cost. The most computationally expensive operation is the evaluation of the score function $\mathbf{s}(\mathbf{x})$. We therefore limit the number of function evaluations (NFE) of the score function to 1 per time step.

This constraint eliminates many traditional high-order numerical schemes that could potentially improve numerical accuracy with more NFE per time step. Moreover, in practice, we have found that traditional high-order schemes (which assume smooth second- or third-order derivatives of $\mathbf{s}$) performs poorly.

Given these considerations, the most accurate approach—while using only one score function evaluation per step—appears to be solving the dynamics analytically under the assumption that $\mathbf{s}(\mathbf{x})$ remains constant during each time step.

**A Naive Solver of FLD**  Within a single time step, the dynamics simplifies to:

$$
\begin{aligned}
d\mathbf{x} &= \mathbf{q}\,d\tau \\
d\mathbf{q} &= \Gamma\left(-\mathbf{q}\,d\tau + \mathbf{s}\,d\tau + \sqrt{2}\,dW_\tau\right)
\end{aligned}
\tag{101}
$$

However, this approach has an important limitation: When we take $\Gamma \to \infty$ to recover the original Langevin dynamics, the system reduces to:

$$
d\mathbf{x} = \mathbf{s}\,d\tau + \sqrt{2}\,dW_\tau
\tag{102}
$$

Solving this exactly over time interval $[0, \Delta\tau]$ with constant $\mathbf{s}$ yields:

$$
\mathbf{x}(\Delta\tau) = \mathbf{x}(0) + \mathbf{s}\Delta\tau + \sqrt{2\Delta\tau}\,\xi,
\tag{103}
$$

where $\xi \sim \mathcal{N}(0, 1)$. This solution is identical to the Euler-Maruyama scheme, meaning it inherits the same white noise problems we aimed to avoid.

While the previous approach represents the best we can do without prior knowledge of the diffusion model, we can fortunately leverage such knowledge to develop a better scheme. In variance-preserving notation, the diffusion model adopts the forward process:

$$\mathbf{x}_t = \sqrt{\bar{\alpha}_t}\mathbf{x}_0 + \sqrt{1 - \bar{\alpha}_t}\boldsymbol{\epsilon} \tag{104}$$

where $\mathbf{x}_0$ is the clean image, $\mathbf{x}_t$ is the noise-contaminated image at diffusion time $t$ (distinct from Langevin dynamics time), $\bar{\alpha}_t$ follows the diffusion schedule (typically approximating $\exp(-t)$), and $\boldsymbol{\epsilon} \sim \mathcal{N}(0, \mathbf{I})$.

The model trains a denoising network $\boldsymbol{\epsilon}(\mathbf{x}_t, t)$ to predict the noise, which relates to the score function through:

$$\boldsymbol{\epsilon}(\mathbf{x}_t, t) = -\sqrt{1 - \bar{\alpha}_t}\nabla_{\mathbf{x}_t} \log p(\mathbf{x}_t) = -\sqrt{1 - \bar{\alpha}_t}\mathbf{s}(\mathbf{x}_t) \tag{105}$$

This relationship enables estimation of the clean image:

$$\hat{\mathbf{x}}_0(\mathbf{x}_t) = \frac{\mathbf{x}_t + (1 - \bar{\alpha}_t)\mathbf{s}(\mathbf{x}_t)}{\sqrt{\bar{\alpha}_t}} \tag{106}$$

This estimator provides a simple way to the clean image $\hat{\mathbf{x}}_0$ without noise.

**The FLD Solver**   We propose a solution to address the white noise issue: by engineering the FLD dynamics such that for large $\Delta\tau$, $\mathbf{x}$ asymptotically converges to the form specified in Eq.104. In contrast to Eq.103, which introduces infinitely large noise to $\mathbf{x}$ as $\Delta\tau \to \infty$, the asymptotic behavior for large $\Delta\tau$ should satisfy:

$$\mathbf{x} \sim \mathcal{N}\left(\mathbf{x} \mid \sqrt{\bar{\alpha}_t}\hat{\mathbf{x}}_0,\, 1 - \bar{\alpha}_t\right) \tag{107}$$

This design ensures that the Langevin dynamics neither introduces excessive noise nor improperly suppresses it when $\Delta\tau$ is large.

Such asymptotic behavior can be technically achieved by treating $\hat{\mathbf{x}}_0(\mathbf{x})$ as constant within each time step, rather than $\mathbf{s}(\mathbf{x})$, leading to the Fast Langevin Dynamics (FLD) equations:

$$\begin{aligned}
d\mathbf{x} &= \mathbf{q}\, d\tau \\
d\mathbf{q} &= \Gamma\left(-\mathbf{q}\, d\tau + \frac{\sqrt{\bar{\alpha}_t}\hat{\mathbf{x}}_0 - \mathbf{x}}{1 - \bar{\alpha}_t}\, d\tau + \sqrt{2}\, dW_\tau\right)
\end{aligned} \tag{108}$$

where the clean image estimate is:

$$\hat{\mathbf{x}}_0(\mathbf{x}) = \frac{\mathbf{x} + (1 - \bar{\alpha}_t)\mathbf{s}(\mathbf{x})}{\sqrt{\bar{\alpha}_t}} \tag{109}$$

Analyzing the limiting case $\Gamma \to \infty$ reveals an Ornstein-Uhlenbeck process:

$$d\mathbf{x} = \frac{\sqrt{\bar{\alpha}_t}\hat{\mathbf{x}}_0 - \mathbf{x}}{1 - \bar{\alpha}_t}\, d\tau + \sqrt{2}\, dW_\tau \tag{110}$$

This process has the desired stationary distribution $\mathcal{N}(\sqrt{\bar{\alpha}_t}\hat{\mathbf{x}}_0, 1 - \bar{\alpha}_t)$, rigorously satisfying Eq.107 and maintaining the correct noise characteristics. Now, let's design an exact solver for the FLD equation Eq.108, treating $\hat{\mathbf{x}}_0$ as a constant.

The FLD equation is a special case of the following general form of a stochastic harmonic oscillator:

$$\begin{aligned}
d\mathbf{x} &= \mathbf{q}\, d\tau \\
d\mathbf{q} &= \Gamma\left(-\mathbf{q}\, d\tau - A\,\mathbf{x}\, d\tau + \mathbf{C}\, d\tau + D\, dW_\tau\right),
\end{aligned} \tag{111}$$

where $\Gamma, A >= 0$, $\mathbf{q} = \frac{d\mathbf{x}}{d\tau}$ and $\boldsymbol{\eta}_\tau = \frac{d\mathbf{W}_\tau}{d\tau}$. This system can be rewritten as a second-order stochastic differential equation:

$$\frac{d^2\mathbf{x}}{d\tau^2} + \Gamma\frac{d\mathbf{x}}{d\tau} + \Gamma A\,\mathbf{x} - \Gamma\mathbf{C} = \Gamma D\,\boldsymbol{\eta}_\tau. \tag{112}$$

Here, $\boldsymbol{\eta}_\tau = \frac{d\mathbf{W}_\tau}{d\tau}$ is the formal derivative of a Wiener process $\mathbf{W}_\tau$, representing white noise. If we formally express the Wiener increment as $d\mathbf{W}_\tau = \sqrt{d\tau}\,\boldsymbol{\epsilon}$, where $\boldsymbol{\epsilon}$ is a standard Gaussian noise, then $\boldsymbol{\eta}_\tau$ can be interpreted as $\boldsymbol{\eta}_\tau = \frac{\boldsymbol{\epsilon}}{\sqrt{d\tau}}$. This defines a singular stochastic process with zero mean and a delta-correlated covariance:

$$\mathbb{E}[\boldsymbol{\eta}_\tau] = 0, \quad \mathbb{E}[\boldsymbol{\eta}_\tau \boldsymbol{\eta}_{\tau'}^T] = \delta(\tau - \tau')\,I. \tag{113}$$

Equation Eq.112 describes a damped harmonic oscillator with noise, whose behavior is governed by the competition between restoring force $A$ and damping $\Gamma$. The key parameter is the discriminant:

$$\Delta = 1 - \frac{4\,A}{\Gamma}, \tag{114}$$

which emerges when solving $\ddot{x} + \Gamma\dot{x} + \Gamma A x = 0$ via the exponential test solution $x = e^{\lambda\tau}$. This yields characteristic roots $\lambda = [-\Gamma \pm \Gamma\sqrt{\Delta}]/2$, revealing three distinct regimes according to square roots of $\Delta$:

- **Underdamped** ($\Delta < 0$): Complex roots cause oscillatory decay (like a swinging pendulum coming to rest).

- **Critically damped** ($\Delta = 0$): A repeated real root enables fastest non-oscillatory return to equilibrium.

- **Overdamped** ($\Delta > 0$): Distinct real roots lead to sluggish, non-oscillatory decay.

The exact solution to the FLD equation Eq.108 with initial conditions $\mathbf{x}(0)$ and $\mathbf{q}(0)$ follows a multivariate normal distribution at time $\tau$:

$$\{\mathbf{x}(\tau), \mathbf{q}(\tau)\} \sim \mathcal{N}(\boldsymbol{\mu}, \Sigma) \tag{115}$$

The mean $\boldsymbol{\mu}$ and covariance $\Sigma$ are given by:

$$\boldsymbol{\mu} = \begin{pmatrix} \mathbf{x} + \Big[\mathbf{q}(0)\zeta_2(\Gamma\tau, \Delta) + (\mathbf{C} - A\mathbf{x})(1 - \zeta_1(\Gamma\tau, \Delta))\Big]\tau \\ \mathbf{q}(0)\Big[E(\Gamma\tau, \Delta) - A(1 - \zeta_1(\Gamma\tau, \Delta))\tau\Big] + (\mathbf{C} - A\mathbf{x})(1 - E(\Gamma\tau, \Delta)) \end{pmatrix} \tag{116}$$

$$\Sigma = D^2 \begin{pmatrix} \tau\sigma_{22}(\Gamma\tau, \Delta) & \frac{1}{2}[\Gamma\tau\zeta_2(\Gamma\tau, \Delta)]^2 \\ \frac{1}{2}[\Gamma\tau\zeta_2(\Gamma\tau, \Delta)]^2 & \frac{\Gamma}{2}\sigma_{11}(\Gamma\tau, \Delta) \end{pmatrix} \tag{117}$$

where the auxiliary functions are defined as:

$$\begin{aligned}
\zeta_1(\Gamma\tau, \Delta) &= 1 - \frac{1 - e^{-\frac{1}{2}\Gamma\tau}\left(\frac{\sinh\left(\frac{1}{2}\Gamma\tau\sqrt{\Delta}\right)}{\sqrt{\Delta}} + \cosh\left(\frac{1}{2}\Gamma\tau\sqrt{\Delta}\right)\right)}{\frac{1}{4}\Gamma\tau(1 - \Delta)} \\
\zeta_2(\Gamma\tau, \Delta) &= \frac{2e^{-\frac{1}{2}\Gamma\tau}\sinh\left(\frac{1}{2}\Gamma\tau\sqrt{\Delta}\right)}{\Gamma\tau\sqrt{\Delta}} \\
E(\Gamma\tau, \Delta) &= 1 - \Gamma\tau\zeta_2(\Gamma\tau, \Delta) \\
\sigma_{11}(\Gamma\tau, \Delta) &= (1 - e^{-\Gamma\tau}) + e^{-\Gamma\tau}\left(\frac{1 - \cosh[\Gamma\tau\sqrt{\Delta}]}{\Delta} + \frac{\sinh[\Gamma\tau\sqrt{\Delta}]}{\sqrt{\Delta}}\right) \\
\sigma_{22}(\Gamma\tau, \Delta) &= \frac{2}{\Gamma\tau(1 - \Delta)}\left[1 - e^{-\Gamma\tau}\left(1 + \frac{\sinh(\Gamma\tau\sqrt{\Delta})}{\sqrt{\Delta}} + \frac{\cosh(\Gamma\tau\sqrt{\Delta}) - 1}{\Delta}\right)\right]
\end{aligned} \tag{118}$$

The solution captures all three damping regimes through the discriminant $\Delta = 1 - 4A/\Gamma$, with the hyperbolic functions smoothly transitioning between oscillatory ($\Delta < 0$), critical ($\Delta = 0$), and overdamped ($\Delta >$

0) behavior. The covariance structure reflects the coupling between position and momentum fluctuations induced by the stochastic forcing.

---

**Algorithm 1:** Stochastic Harmonic Oscillator

---

**Input:** Initial position $\mathbf{x}_0 \in \mathbb{R}^n$, initial momentum $\mathbf{q}_0 \in \mathbb{R}^n$ (optional), time step $\tau > 0$, friction $\Gamma > 0$, scalar $A \in \mathbb{R}$, vector $\mathbf{C} \in \mathbb{R}^n$, scalar $D \in \mathbb{R}$

**Output:** Final position $\mathbf{x}_\tau \in \mathbb{R}^n$, final momentum $\mathbf{q}_\tau \in \mathbb{R}^n$

**if** $\mathbf{q}_0$ *is None* **then**

    Sample $\mathbf{z} \sim \mathcal{N}(0, I_m)$ ;                     `// Standard normal vector in` $\mathbb{R}^m$

    Set $\mathbf{q}_0 \leftarrow \sqrt{\frac{\Gamma}{2}} \cdot D \cdot \mathbf{z}$ ;       `// Initialize the velocity according to stationary`

    `distribution`

**end**

Sample $\begin{bmatrix} \mathbf{x}_\tau \\ \mathbf{q}_\tau \end{bmatrix} \sim \mathcal{N}(\boldsymbol{\mu}, \Sigma)$ according to Eq.115;

**return** $\mathbf{x}_\tau, \mathbf{q}_\tau$ ;

---

**Parameter Schedule** For the FLD equation Eq.108, the coefficients take specific forms:

$$A = \frac{1}{1 - \bar{\alpha}_t}, \quad \mathbf{C} = \frac{\sqrt{\bar{\alpha}_t}\hat{\mathbf{x}}_0(\mathbf{x})}{1 - \bar{\alpha}_t} = \mathbf{s}(\mathbf{x}) + A\,\mathbf{x}, \quad D = \sqrt{2} \tag{119}$$

where $\mathbf{s}(\mathbf{x})$ is the score function. Substituting these into the general solution Eq.115 yields an exact analytical solver for the FLD dynamics ($\hat{\mathbf{x}}_0$ treated as constant).

More generally, we have the freedom to choose the coefficients $\mathbf{C}$ and $A$, as long as they add up to the score function

$$\mathbf{s}(\mathbf{x}) = \mathbf{C}(\mathbf{x}) - A\,\mathbf{x}, \tag{120}$$

where $\mathbf{C}$ is treated as a constant during a single time step. This freedom allow us to do the following modification to Eq.119

1. The coefficients can take the following forms:

$$A = \frac{1}{1 - \bar{\alpha}_t + \bar{\alpha}_t\,\alpha}, \quad \mathbf{C} = \mathbf{s}(\mathbf{x}) + A\,\mathbf{x}, \quad D = \sqrt{2}. \tag{121}$$

The hyperparameter $\alpha > 0$ can be tuned based on the task. It represents the expected noise level of the sampling target, derived according to the forward diffusion process Eq.2, under which a Gaussian random variable with standard deviation $\alpha$ follows the distribution $\mathcal{N}(0, 1 - \bar{\alpha}_t + \bar{\alpha}_t\alpha)$, whose score has linear term of the form $\frac{-1}{1 - \bar{\alpha}_t + \bar{\alpha}_t\,\alpha}\mathbf{x}$. It is particularly useful when the assumption that $\hat{\mathbf{x}}_0(\mathbf{x})$ is constant does not hold. For instance, when sampling from a Gaussian distribution with standard deviation $\sigma$, $\hat{\mathbf{x}}_0(\mathbf{x})$ varies, and setting $\alpha = \sigma$ optimizes $A$. Thus, $\alpha$ can be interpreted as the "noise level" of the target distribution. For image generation tasks, set $\alpha = 0$.

2. The coefficients can alternatively be expressed as:

$$A = \frac{1 + \lambda}{1 - \bar{\alpha}_t}, \quad \mathbf{C} = \mathbf{s}(\mathbf{x}) + A\,\mathbf{x} \quad D = \sqrt{2} \tag{122}$$

This formulation incorporates the guidance scale $\lambda$ from Eq.12, where $A$ scales proportionally with $\lambda$. The proportional relationship ensures stable solutions even at large $\lambda$ values. In practice, we adopt this set of parameter for the **masked** ($\mathbf{y}$, known) part of LanPaint.

Substituting these coefficients into the general solution (FLD Gaussian solution) yields exact analytical expressions for the FLD dynamics over an arbitrary time interval $[0, \tau]$. In practice, this can be adapted to a shifted interval $[\tau, \tau + \Delta\tau]$.

Note that the parameters $A$ and $\mathbf{C}$ depend on the diffusion time $t$ (or say noise level), meaning the FLD dynamics vary throughout the diffusion process. Consequently, both the time step $\Delta\tau$ for each iteration and

the friction coefficient $\Gamma$ must be adjusted accordingly to adapt to these changing dynamics and maintain solution stability.

**FLD Solver Summary**  As a summary of previous discussion. We now show one time step of the FLD solver

---

**Algorithm 2:** 1st-order FLD solver

---

**Input:** $\mathbf{x}_0$, $\mathbf{q}_0$, $\Delta\tau$, $\Gamma$, $A$, $D$, function $Ccoef$
**Output:** $\mathbf{x}_\tau$, $\mathbf{q}_\tau$
`// Compute coefficients C via Eq.121 or Eq.122`
$\mathbf{C} \leftarrow Ccoef(\mathbf{x}_{\tau/2})$
`// Advance time with Algorithm 1`
$\mathbf{x}_\tau, \mathbf{q}_\tau \leftarrow \mathrm{StochasticHarmonicOscillator}(\mathbf{x}_0, \mathbf{q}_0, \Delta\tau, \Gamma, A, \mathbf{C}, D)$
**return** $\mathbf{x}_\tau$, $\mathbf{q}_\tau$, $\mathbf{C}$,

---

The solver can be made second-order accurate in time step $\tau$ by introducing midpoint states $\mathbf{x}_{\tau/2}$, $\mathbf{q}_{\tau/2}$ without requiring additional neural network evaluations, maintaining the same computational efficiency. The idea is to do the following operator splitting:

$$
\begin{aligned}
d\mathbf{x} &= \mathbf{q}\, d\tau \\
d\mathbf{q} &= \Gamma\left(-\mathbf{q}\, d\tau - A\, \mathbf{x}\, d\tau + \mathbf{C}(\mathbf{x})\, d\tau + D\, dW_\tau\right),
\end{aligned}
\tag{123}
$$

split into

$$
\begin{aligned}
d\mathbf{x} &= \mathbf{q}\, d\tau \\
d\mathbf{q} &= \Gamma\left(-\mathbf{q}\, d\tau - A\, \mathbf{x}\, d\tau + \mathbf{C}_0\, d\tau + D\, dW_\tau\right),
\end{aligned}
\tag{124}
$$

and

$$
\begin{aligned}
d\mathbf{x} &= \mathbf{0} \\
d\mathbf{q} &= \Gamma\left(\mathbf{C}(\mathbf{x}) - \mathbf{C}_0\right) d\tau,
\end{aligned}
\tag{125}
$$

then do a 2nd order Strang splitting. The resulting method is

---

**Algorithm 3:** 2nd-order FLD solver

---

**Input:** $\mathbf{x}_0$, $\mathbf{q}_0$, $\Delta\tau$, $\Gamma$, $A$, $\mathbf{C}_0$, $D$, function $Ccoef$
**Output:** $\mathbf{x}_\tau$, $\mathbf{q}_\tau$, $\mathbf{C}_\tau$
`// Advance time with Algorithm 1`
$\mathbf{x}_{\tau/2}, \mathbf{q}_{\tau/2} \leftarrow \mathrm{StochasticHarmonicOscillator}(\mathbf{x}, \mathbf{q}, \Delta\tau/2, \Gamma, A, \mathbf{C}_0, D)$
`// Compute C via Eq.121 or Eq.122`
$\mathbf{C}_\tau \leftarrow Ccoef(\mathbf{x}_{\tau/2})$
$\mathbf{q}_{\tau/2} \leftarrow \mathbf{q}_{\tau/2} + \Gamma(\mathbf{C}_\tau - \mathbf{C}_0)\Delta\tau$
`// Advance time with Algorithm 1`
$\mathbf{x}_\tau, \mathbf{q}_\tau \leftarrow \mathrm{StochasticHarmonicOscillator}(\mathbf{x}_{\tau/2}, \mathbf{q}_{\tau/2}, \Delta\tau/2, \Gamma, A, \mathbf{C}_0, D)$
**return** $\mathbf{x}_\tau$, $\mathbf{q}_\tau$, $\mathbf{C}_\tau$

---

**Friction Schedule**  The friction schedule is designed based on a core principle: the FLD dynamics should maintain its characteristic damping behavior consistently throughout the entire diffusion process. Whether the system is underdamped or overdamped, this state should remain invariant for all time $t$.

To achieve this, we ensure that the discriminant $\Delta$ remains constant across all diffusion times $t$. A straightforward choice is

$$
\Gamma_t = \Gamma_0 A_t
\tag{126}
$$

This schedule preserves a constant discriminant for each time $t$. It uniformly control the global damping behavior - transitioning between underdamped and overdamped regimes - while ensuring consistent dynamics at every diffusion step.

**Time Step Schedule**  We employ the time step schedule inversely proportional to $A$

$$\Delta\tau_t = \Delta\tau_0 \frac{A_T}{A_t} \tag{127}$$

to maintain a constant product $\Gamma\Delta\tau$ across all diffusion times ($A_T$ acts as a normalization constant). This design is motivated by the fundamental principle that the step size should adapt to the system's rate of change: faster-evolving dynamics (large $\Gamma$) require smaller steps, while slower dynamics (small $\Gamma$) permit larger ones.

The key insight comes from examining the exponential terms in Eq.118. When $\tau = \Delta\tau$, most terms scale like $e^{-\Gamma\Delta\tau}$, where the product $\Gamma\Delta\tau$ directly determines the decay rate. By keeping $\Gamma\Delta\tau$ constant, we ensure a consistent "amount of change" per step—effectively balancing step size with the system's intrinsic timescale. This approach automatically adjusts $\Delta\tau_t$ to be smaller when $\Gamma_t$ is large (fast dynamics) and larger when $\Gamma_t$ is small (slow dynamics), yielding stable and efficient numerical step across all diffusion time.

**Extension of FLD: Hessian-Free High Resolution(HFHR) Dynamics**  The HFHR technique accelerates the convergence of Underdamped Langevin Dynamics (ULD) by introducing a new parameter $\alpha$ into the ULD dynamics:

$$
\begin{aligned}
d\mathbf{x} &= \frac{1}{m}\mathbf{v}dt + \alpha\,\mathbf{s}(\mathbf{x})dt + \sqrt{2\alpha}dW_{\mathbf{x}} \\
d\mathbf{v} &= -\frac{\gamma}{m}\mathbf{v}dt + \mathbf{s}(\mathbf{x})dt + \sqrt{2\gamma}dW_{\mathbf{v}}
\end{aligned}
\tag{128}
$$

The additional term, $\alpha\,\mathbf{s}(\mathbf{x})dt + \sqrt{2\alpha}dW_{\mathbf{x}}$, corresponds to the original Langevin dynamics. Empirically, setting $\alpha > 0$ accelerates convergence, but it remains unclear whether this improvement stems from an increased effective step size in ULD or an inherent acceleration due to the added terms. To analyze the dynamics, we perform a parameter transformation. Let

$$\Psi = \alpha\gamma + 1, \quad \mathbf{q} = \frac{\gamma}{\Psi}\frac{\mathbf{v}}{m}, \quad \tau = \Psi\frac{t}{\gamma}, \quad \Gamma = \frac{\gamma^2}{m\Psi}, \tag{129}$$

which transforms the system into:

$$
\begin{aligned}
d\mathbf{x} &= \mathbf{q}\,d\tau + \frac{\Psi-1}{\Psi}\,\mathbf{s}(\mathbf{x})\,d\tau + \sqrt{2\frac{\Psi-1}{\Psi}}\,dW_{\mathbf{x}}, \\
d\mathbf{q} &= \Gamma\left(-\mathbf{q}\,d\tau + \frac{1}{\Psi}\,\mathbf{s}(\mathbf{x})\,d\tau + \sqrt{\frac{2}{\Psi}}\,dW_{\mathbf{v}}\right).
\end{aligned}
\tag{130}
$$

In this form, two key observations emerge:

1. **Limit behavior**: As $\Gamma \to \infty$, the dynamics reduces to the original Langevin dynamics:

$$d\mathbf{x} = \mathbf{s}(\mathbf{x})\,d\tau + \sqrt{2}\,dW_\tau. \tag{131}$$

2. **Linear combination**: The HFHR dynamics is a weighted combination of Langevin dynamics and ULD, with weighting factor $\frac{1}{\Psi}$ and $\frac{\Psi-1}{\Psi}$.

This reveals that HFHR does not introduce inherent acceleration beyond ULD. Instead, its convergence improvement stems primarily from an increased effective step size. For this reason, we do not adopt HFHR in the FLD sampler.

**Extension of FLD: Pre-conditioned Langevin Dynamics**  The stationary distribution $\pi(\mathbf{x})$ of the original Langevin dynamics

$$d\mathbf{x} = \mathbf{s}(\mathbf{x})\,d\tau + \sqrt{2}\,dW_\tau, \tag{132}$$

---

**Algorithm 4:** LanPaint, Variance Perserving Notation

---

**Input:** Input image $\mathbf{y}$, mask $\mathbf{m}$, text embeddings $\mathbf{e}$, step size $\eta$, steps $N$, friction $\gamma$, expected noise $\alpha$, guidance scale $\lambda$

**Output:** Inpainted image $\mathbf{z}$

**Initialize:**

$\mathbf{x} \leftarrow$ Random noise ;             `// Latent variable`

$\{\mathbf{y}_t\} \leftarrow$ ForwardDiffuse($\mathbf{y}$) ;         `// Pre-diffused inputs`

**for** *each timestep t* **do**

     $\sigma \leftarrow$ scheduler.sigma($t$) ;     `// Get the noise level (VE notation) from scheduler`

     $\bar{\alpha}_t \leftarrow 1/(1 + \sigma^2)$ ;           `// Compute alpha bar (VP notation)`

     `// Prepare parameters for x and y regions`

     $A_x \leftarrow 1/(1 - \bar{\alpha}_t + \bar{\alpha}_t \alpha)$

     $A_y \leftarrow (1 + \lambda)/(1 - \bar{\alpha}_t)$

     $\Gamma_x \leftarrow \gamma^2 A_x$ ;             `// Friction coefficient for x`

     $\Gamma_y \leftarrow \gamma^2 A_y$ ;             `// Friction coefficient for y`

     $D \leftarrow \sqrt{2}$ ;        `// Diffusion coefficient, assumed equal for x and y`

     `// Set step sizes based on sigma functions`

     $\sigma_x \leftarrow (1 - \bar{\alpha}_t + \bar{\alpha}_t \alpha)$

     $\sigma_y \leftarrow (1 - \bar{\alpha}_t)$

     $d\tau \leftarrow \eta$ ;              `// Base step size`

     $\mathbf{q}, \mathbf{C} \leftarrow None, None$

     **Function** `Ccoef(`$\mathbf{x}$`)`:

         `// Compute score`

         $\epsilon \leftarrow$ UNet($\mathbf{x}, t$)

         $\mathbf{s} \leftarrow -\epsilon/\sqrt{1 - \bar{\alpha}_t}$

         `// Compute BiG score`

         $\mathbf{s}_\lambda \leftarrow \mathbf{s} \odot (1 - \mathbf{m}) + \left( \frac{(1+\lambda)(\sqrt{\bar{\alpha}_t}\mathbf{y} - \mathbf{x})}{(1 - \bar{\alpha}_t)} - \lambda \mathbf{s} \right) \odot \mathbf{m}$

         `// Compute masked` $\mathbf{C}$

         $\mathbf{C} \leftarrow (\mathbf{s}_\lambda + A_x \mathbf{x}) \odot (1 - \mathbf{m}) + (\mathbf{s}_\lambda + A_y \mathbf{x}) \odot \mathbf{m}$

         **return** $\mathbf{C}$

     `// FLD dynamics with stochastic harmonic oscillator`

     **for** $k = 1$ *to* $N$ **do**

         **if** $\mathbf{q}$ *is None* **then**

             `// Advance time with FLD 1st order algorithm 2`

             $\mathbf{x}, \mathbf{q}, \mathbf{C} \leftarrow$ FLD_1st($\mathbf{x}, \mathbf{q}, d\tau, \Gamma, A, D, \text{Ccoef}$)

         **else**

             `// Advance time with FLD 2nd order algorithm 3`

             $\mathbf{x}, \mathbf{q}, \mathbf{C} \leftarrow$ FLD_2nd($\mathbf{x}, \mathbf{q}, d\tau, \Gamma, A, \mathbf{C}, D, \text{Ccoef}$)

         **end**

     **end**

     `// After LanPaint steps, use scheduler to step`

     $\epsilon \leftarrow$ UNet($\mathbf{z}, t$)

     $\mathbf{z} \leftarrow$ SchedulerStep($\mathbf{z}, \epsilon, t$)

**end**

$\mathbf{z} \leftarrow \mathbf{z} \odot (1 - m) + \mathbf{y} \odot m$

**return** $\mathbf{z}$

---

where $\mathbf{s}(\mathbf{x}) = \nabla_\mathbf{x} \log \pi(\mathbf{x})$, is also the stationary distribution of the pre-conditioned dynamics

$$d\mathbf{x} = P\,\mathbf{s}(\mathbf{x})\,d\tau + \sqrt{2P}\,dW_\tau, \tag{133}$$

with $P$ a positive definite symmetric matrix. A distribution is stationary if it remains unchanged after a small time step $\Delta\tau$, i.e., $\pi'(\mathbf{x}') = \pi(\mathbf{x}')$.

The transition probability for the pre-conditioned dynamics over a small time step is given by a Gaussian integral:

$$\pi'(\mathbf{x}') = \int \pi(\mathbf{x}) \, \mathcal{N}(\mathbf{x}'; \mathbf{x} + P\mathbf{s}(\mathbf{x})\Delta\tau, 2P\Delta\tau) \, d\mathbf{x}, \tag{134}$$

where $\mathcal{N}(\mathbf{x}'; \mathbf{m}, \Sigma)$ denotes a multivariate Gaussian with mean $\mathbf{m} = \mathbf{x} + P\mathbf{s}(\mathbf{x})\Delta\tau$ and covariance $\Sigma = 2P\Delta\tau$. Since the time step is small, the Gaussian is sharply peaked near $\mathbf{x}'$, allowing us to simplify the integral.

To evaluate this, we approximate the score function near $\mathbf{x}'$, assuming $\mathbf{s}(\mathbf{x}) \approx \mathbf{s}(\mathbf{x}')$ for $\mathbf{x}$ close to $\mathbf{x}'$, as the dynamics involve small steps. We introduce a change of variables, defining $\mathbf{y} = \mathbf{x} + P\mathbf{s}(\mathbf{x})\Delta\tau$, which we approximate as:

$$\mathbf{y} \approx \mathbf{x} + P\mathbf{s}(\mathbf{x}')\Delta\tau. \tag{135}$$

The inverse transformation is:

$$\mathbf{x} = \mathbf{y} - P\mathbf{s}(\mathbf{x}')\Delta\tau. \tag{136}$$

The Jacobian determinant of this transformation, to first order, is approximately $1 - P\nabla \cdot \mathbf{s}(\mathbf{x}')\Delta\tau$, so the volume element transforms as:

$$d\mathbf{x} = \left(1 - P\nabla \cdot \mathbf{s}(\mathbf{x}')\Delta\tau\right)d\mathbf{y}. \tag{137}$$

Using the symmetry of the Gaussian, $\mathcal{N}(\mathbf{x}'; \mathbf{y}, 2P\Delta\tau) = \mathcal{N}(\mathbf{y}; \mathbf{x}', 2P\Delta\tau)$, the integral becomes:

$$\pi'(\mathbf{x}') = \left(1 - P\nabla \cdot \mathbf{s}(\mathbf{x}')\Delta\tau\right) \int \pi(\mathbf{y} - P\mathbf{s}(\mathbf{x}')\Delta\tau) \, \mathcal{N}(\mathbf{y}; \mathbf{x}', 2P\Delta\tau) \, d\mathbf{y}. \tag{138}$$

Define the deviation $\Delta\mathbf{x} = \mathbf{y} - \mathbf{x}' - P\mathbf{s}(\mathbf{x}')\Delta\tau$, so that $\mathbf{y} - P\mathbf{s}(\mathbf{x}')\Delta\tau = \mathbf{x}' + \Delta\mathbf{x}$. Since $\mathbf{y}$ follows a Gaussian distribution centered at $\mathbf{x}'$ with covariance $2P\Delta\tau$, we compute the moments:

$$\mathbb{E}[\Delta\mathbf{x}] = -P\mathbf{s}(\mathbf{x}')\Delta\tau, \quad \mathbb{E}[\Delta\mathbf{x}\Delta\mathbf{x}^T] = 2P\Delta\tau. \tag{139}$$

We approximate the density at the shifted point using a Taylor expansion:

$$\pi(\mathbf{x}' + \Delta\mathbf{x}) \approx \pi(\mathbf{x}') + \Delta\mathbf{x}^T\nabla\pi(\mathbf{x}') + \frac{1}{2}\Delta\mathbf{x}^T\nabla\nabla\pi(\mathbf{x}')\Delta\mathbf{x}. \tag{140}$$

Taking the expectation over the Gaussian, the integral evaluates to:

$$\int \pi(\mathbf{x}' + \Delta\mathbf{x})\mathcal{N} \, d\mathbf{y} \approx \pi(\mathbf{x}') - \Delta\tau\mathbf{s}(\mathbf{x}')^T P\nabla\pi(\mathbf{x}') + \Delta\tau P : \nabla\nabla\pi(\mathbf{x}'). \tag{141}$$

Multiplying by the Jacobian factor and collecting terms up to order $\Delta\tau$, we obtain:

$$\pi'(\mathbf{x}') \approx \pi(\mathbf{x}') - \Delta\tau\left[\nabla \cdot (P\mathbf{s}(\mathbf{x}')\pi(\mathbf{x}')) - \nabla \cdot (P\nabla\pi(\mathbf{x}'))\right] + \mathcal{O}(\Delta\tau^2). \tag{142}$$

Since the score function satisfies $\mathbf{s}(\mathbf{x}') = \nabla\log\pi(\mathbf{x}') = \frac{\nabla\pi(\mathbf{x}')}{\pi(\mathbf{x}')}$, we have:

$$P\mathbf{s}(\mathbf{x}')\pi(\mathbf{x}') = P\nabla\pi(\mathbf{x}'). \tag{143}$$

Substituting this into the expression, the divergence terms cancel. Thus, the updated distribution simplifies to:

$$\pi'(\mathbf{x}') = \pi(\mathbf{x}') + \mathcal{O}(\Delta\tau^2). \tag{144}$$

As the time step approaches zero, the higher-order terms vanish, yielding $\pi'(\mathbf{x}') = \pi(\mathbf{x}')$. Therefore, $\pi(\mathbf{x})$ is the stationary distribution of the pre-conditioned dynamics. QED.

The FLD dynamics Eq.101 can also be preconditioned by a positive definite symmetric matrix $P$, yielding:

$$\begin{aligned} d\mathbf{x} &= P\mathbf{q} \, d\tau, \\ d\mathbf{q} &= \Gamma\left(-P\mathbf{q} \, d\tau + P\mathbf{s} \, d\tau + \sqrt{2P} \, dW_\tau\right). \end{aligned} \tag{145}$$

When $P$ is a diagonal matrix, its diagonal elements $P_{ii}$ act as scaling factors for each dimension of the system. This effectively assigns a distinct time step $\Delta\tau_i = P_{ii}\Delta\tau$ to the dynamics of each dimension, allowing independent control over the rate of evolution along each coordinate. For example, a larger $P_{ii}$ accelerates the dynamics in the $i$-th dimension, equivalent to a larger time step, while a smaller $P_{ii}$ slows it down. This flexibility enables tailored convergence speed for each dimension without altering the system's stationary distribution.

# H A General Form of Langevin Dynamics and Its Stationary Distribution

In this section, we present a unified proof demonstrating that ULD, FLD, pre-conditioned, and HFHR dynamics all share the same stationary distribution as the original Langevin dynamics.

## 1. General Relation Between SDEs and the Fokker–Planck Equation

The Fokker–Planck equation describes the time evolution of the probability density $\rho(\mathbf{z}, t)$ associated with a stochastic process governed by a stochastic differential equation (SDE). For a general SDE of the form:

$$dz_i = h_i(\mathbf{z}) \, dt + \gamma_{ij}(\mathbf{z}) \, dW_j, \tag{146}$$

the corresponding Fokker–Planck equation is:

$$\frac{\partial \rho(\mathbf{z}, t)}{\partial t} = -\frac{\partial}{\partial z_i} \big[ h_i(\mathbf{z}) \rho(\mathbf{z}, t) \big] + \frac{1}{2} \frac{\partial^2}{\partial z_j \partial z_k} \big[ \gamma_{ji}(\mathbf{z}) \gamma_{ki}(\mathbf{z}) \rho(\mathbf{z}, t) \big], \tag{147}$$

where $h_i(\mathbf{z})$ is the drift term, $\gamma_{ij}(\mathbf{z})$ is the diffusion matrix, and $dW_j$ are independent Wiener processes.

## 2. Fokker–Planck Equation and Stationary Distribution of the Langevin Dynamics

Consider the SDE:

$$d\mathbf{z} = \nabla_{\mathbf{z}} \log p(\mathbf{z}) \, dt + \sqrt{2} \, d\mathbf{W}_{\mathbf{z}}. \tag{148}$$

The drift term is $h(\mathbf{z}) = \nabla_{\mathbf{z}} \log p(\mathbf{z})$, and the diffusion matrix is constant with $\gamma_{ij} = \sqrt{2}\delta_{ij}$. The Fokker–Planck equation becomes:

$$\frac{\partial \rho(\mathbf{z}, t)}{\partial t} = -\frac{\partial}{\partial z_i} \Big[ \Big( \frac{\partial \log p(\mathbf{z})}{\partial z_i} \Big) \rho \Big] + \frac{\partial^2 \rho}{\partial z_i^2}. \tag{149}$$

At stationarity, $\frac{\partial \rho}{\partial t} = 0$, leading to:

$$0 = -\frac{\partial}{\partial z_i} \Big[ \Big( \frac{\partial \log p(\mathbf{z})}{\partial z_i} \Big) \rho \Big] + \frac{\partial^2 \rho}{\partial z_i^2}. \tag{150}$$

Solving this equation shows that the stationary distribution is:

$$\rho(\mathbf{z}) = p(\mathbf{z}), \tag{151}$$

where $p(\mathbf{z})$ is the target probability distribution.

## 3. Fokker–Planck Equation and Stationary Distribution of Fast Langevin Dynamics

Now consider the SDE system:

$$\begin{aligned}
d\mathbf{z} &= \frac{1}{m} P\mathbf{v} \, dt + \alpha P \nabla_{\mathbf{z}} \log p(\mathbf{z}) \, dt + \sqrt{2\alpha P} \, d\mathbf{W}_{\mathbf{z}}, \\
d\mathbf{v} &= -\frac{\gamma}{m} P\mathbf{v} \, dt + P \nabla_{\mathbf{z}} \log p(\mathbf{z}) \, dt + \sqrt{2\gamma P} \, d\mathbf{W}_{\mathbf{v}},
\end{aligned} \tag{152}$$

where $P$ is a symmetric positive semidefinite matrix. This system is the general form of underdamped Langevin dynamics, with preconditioning and the HFHR technique introduced in Appendix G. The Fokker–Planck equation for this system is:

$$\begin{aligned}
\frac{\partial \rho(\mathbf{z}, \mathbf{v}, t)}{\partial t} &= -\frac{\partial}{\partial z_i} P_{ij} \Big[ \Big( \frac{1}{m} v_j + \alpha \frac{\partial \log p(\mathbf{z})}{\partial z_j} \Big) \rho - \alpha \frac{\partial \rho}{\partial z_j} \Big] \\
&\quad - \frac{\partial}{\partial v_i} P_{ij} \Big[ \Big( -\frac{\gamma}{m} v_j + \frac{\partial \log p(\mathbf{z})}{\partial z_j} \Big) \rho - \gamma \frac{\partial \rho}{\partial v_j} \Big].
\end{aligned} \tag{153}$$

To determine the stationary distribution, assume:

$$\rho(\mathbf{z}, \mathbf{v}) = p(\mathbf{z})\mathcal{N}(\mathbf{v}|\mathbf{0}, m\mathbf{I}), \tag{154}$$

where $p(\mathbf{z})$ is the marginal distribution of $\mathbf{z}$, and $\mathcal{N}(\mathbf{v}|\mathbf{0}, m\mathbf{I})$ is a Gaussian distribution with zero mean and covariance $m\mathbf{I}$. Substituting into the Fokker–Planck equation, we find:

$$\frac{\partial \rho(\mathbf{z}, \mathbf{v}, t)}{\partial t} = -\frac{\partial}{\partial z_i} P_{ij} \Big[ \frac{1}{m} v_j p(\mathbf{z})\mathcal{N}(\mathbf{v}|\mathbf{0}, m\mathbf{I}) \Big] - \frac{\partial}{\partial v_i} P_{ij} \Big[ \frac{\partial p(\mathbf{z})}{\partial z_j} \mathcal{N}(\mathbf{v}|\mathbf{0}, m\mathbf{I}) \Big]. \tag{155}$$

Note that $\partial_{v_i}\mathcal{N}(\mathbf{v}|\mathbf{0}, m\mathbf{I}) = -\frac{v_i}{m}\mathcal{N}(\mathbf{v}|\mathbf{0}, m\mathbf{I})$, therefore these terms cancel, confirming:

$$\rho(\mathbf{z}, \mathbf{v}) = p(\mathbf{z})\mathcal{N}(\mathbf{v}|\mathbf{0}, m\mathbf{I}) \tag{156}$$

is a stationary solution. This implies: 1. $\mathbf{z}$ follows the target distribution $p(\mathbf{z})$, 2. $\mathbf{v}$ is independent of $\mathbf{z}$ and thermalized around zero with variance proportional to $m$.

Thus, the stationary distribution is a decoupled joint distribution where $\mathbf{z}$ governs the spatial distribution, and $\mathbf{v}$ represents a Gaussian thermal velocity.

**The Fast Langevin Dynamics (FLD)**   The FLD reparametrizes Eq.152 through the transformations: $\mathbf{q} = \frac{\gamma}{m}\mathbf{v}$, $\tau = \frac{t}{\gamma}$, $\Gamma = \frac{\gamma^2}{m}$, $m = 1$, $\alpha = 0$, and $P = I$, resulting in the following system:

$$
\begin{aligned}
d\mathbf{z} &= \mathbf{q}\, d\tau \\
d\mathbf{q} &= \Gamma\left(-\mathbf{q}\, d\tau + \mathbf{s}(\mathbf{z})\, d\tau + \sqrt{2}\, dW_\tau\right)
\end{aligned}
\tag{157}
$$

where $\mathbf{s}(\mathbf{z}) = \nabla_{\mathbf{z}} \log p(\mathbf{z})$. Transforming $\mathbf{v}$ to $\mathbf{q}$, we have the stationary distribution:

$$\rho(\mathbf{z}, \mathbf{q}) = p(\mathbf{z})\mathcal{N}(\mathbf{q}|\mathbf{0}, \Gamma) \tag{158}$$

This proves Theorem 4.2.

# I   More Production-Level Model Evaluations Across Architectures

This section offers a qualitative analysis of LanPaint's performance across diverse models, in comparison to ComfyUI's built-in inpainting functionality (ComfyUI Wiki, 2025), which is a variant of the Replace method. The evaluation demonstrates LanPaint's strong generalization capabilities, effectively handling various mask types and models from different communities and companies, across a range of architectures.

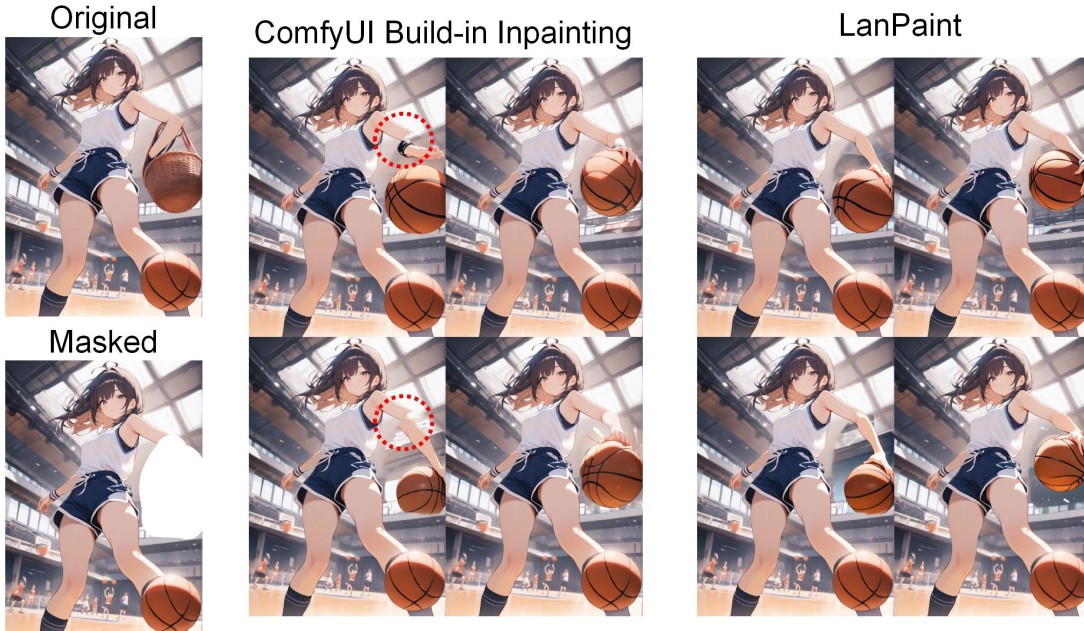

Figure 10: Model: animagineXL40_v4Opt, Prompt: "basketball, masterpiece, high score, great score, absurdres", Steps: 30, CFG Scale: 5.0, Sampler: Euler, Scheduler: Karras, LanPaint Iteration Steps: 2, Seed: 0, Batch Size: 4

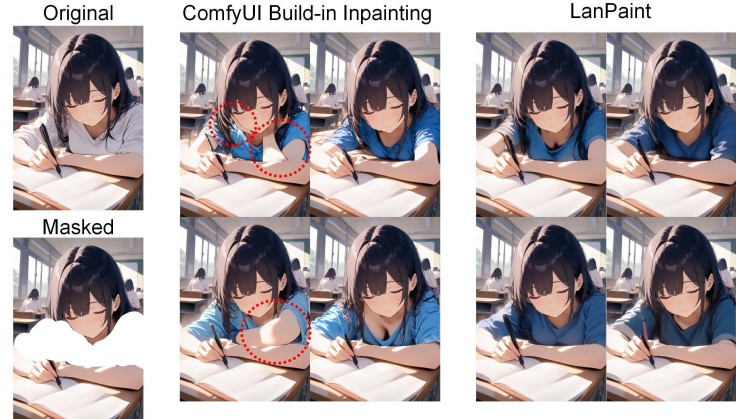

Figure 11: Model: animagineXL40_v4Opt, Prompt: "1girl, blue shirt, masterpiece, high score, great score, absurdres", Steps: 30, CFG Scale: 5.0, Sampler: Euler, Scheduler: Karras, LanPaint Iteration Steps: 5, Seed: 0, Batch Size: 4

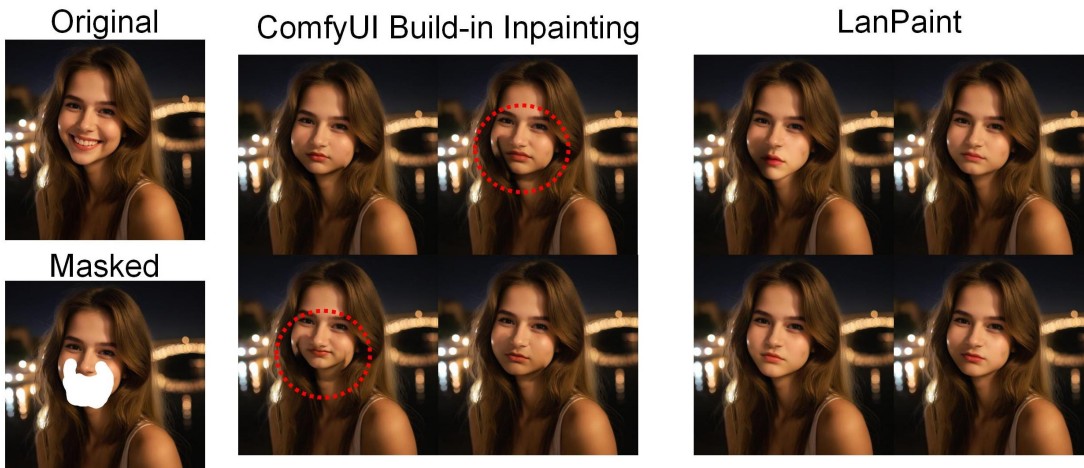

Figure 12: Model: juggernautXL_juggXIByRundiffusion, Prompt: "1girl, sad, beautiful girl, night, masterpiece", Steps: 30, CFG Scale: 5.0, Sampler: Euler, Scheduler: Karras, LanPaint Iteration Steps: 5, Seed: 0, Batch Size: 4

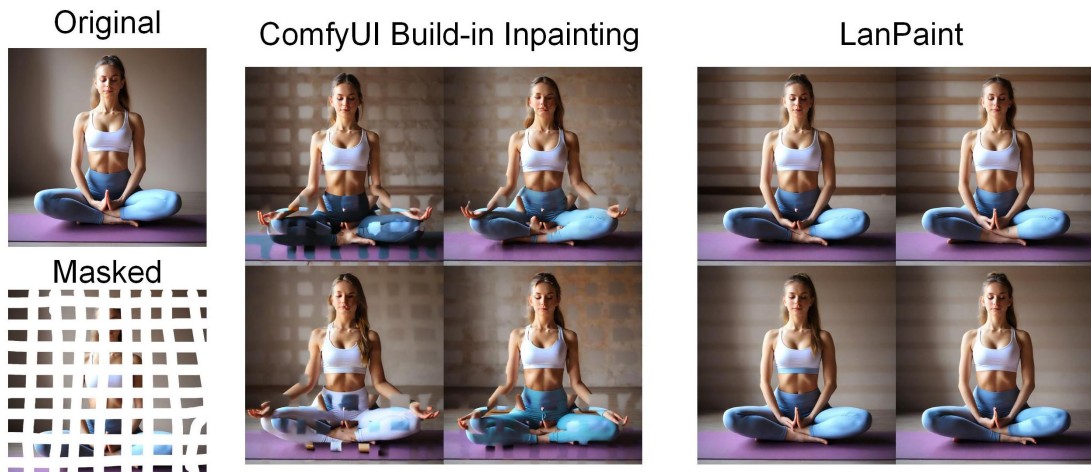

Figure 13: Model: juggernautXL_juggXIByRundiffusion, Prompt: "1girl, yoga, beautiful, masterpiece", Steps: 30, CFG Scale: 5.0, Sampler: Euler, Scheduler: Karras, LanPaint Iteration Steps: 5, Seed: 0, Batch Size: 4

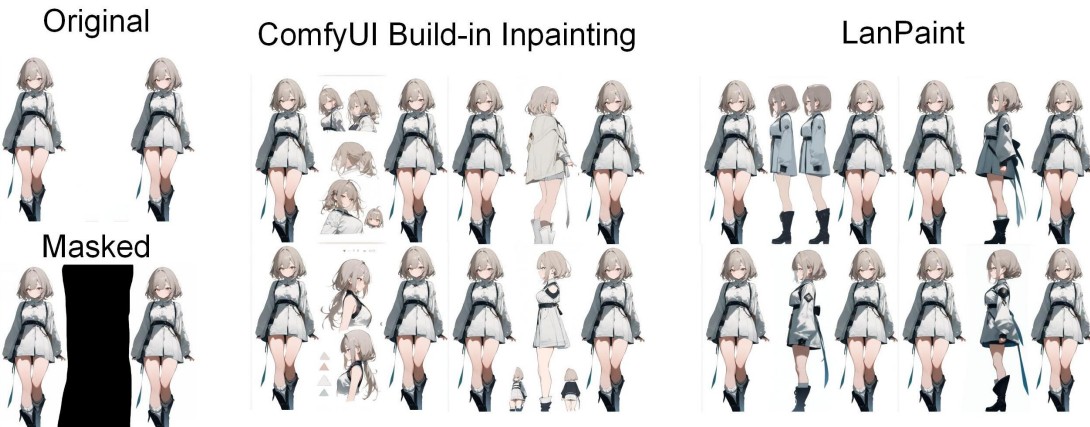

Figure 14: Model: animagineXL40_v4Opt, Prompt: "1girl, multiple views, multiple angles, clone, turnaround, from side, masterpiece, high score, great score, absurdres", Steps: 30, CFG Scale: 5.0, Sampler: Euler, Scheduler: Karras, LanPaint Iteration Steps: 5, Seed: 0, Batch Size: 4

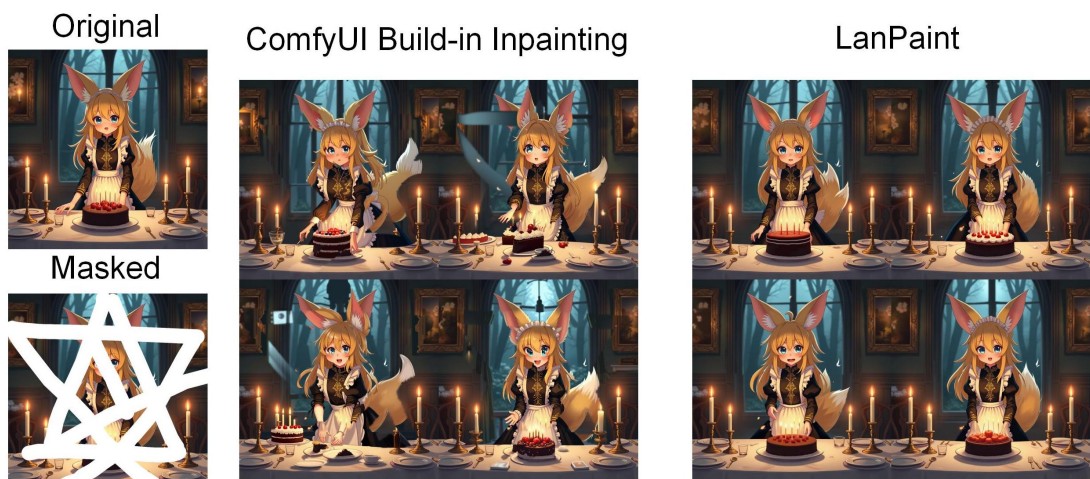

Figure 15: Model: flux1-dev-fp8, Prompt: "cute anime girl with massive fluffy fennec ears and a big fluffy tail blonde messy long hair blue eyes wearing a maid outfit with a long black gold leaf pattern dress and a white apron mouth open placing a fancy black forest cake with candles on top of a dinner table of an old dark Victorian mansion lit by candlelight with a bright window to the foggy forest and very expensive stuff everywhere there are paintings on the walls", Steps: 30, CFG Scale: 1.0, Sampler: Euler, Scheduler: Simple, LanPaint Iteration Steps: 5, Seed: 0, Batch Size: 4

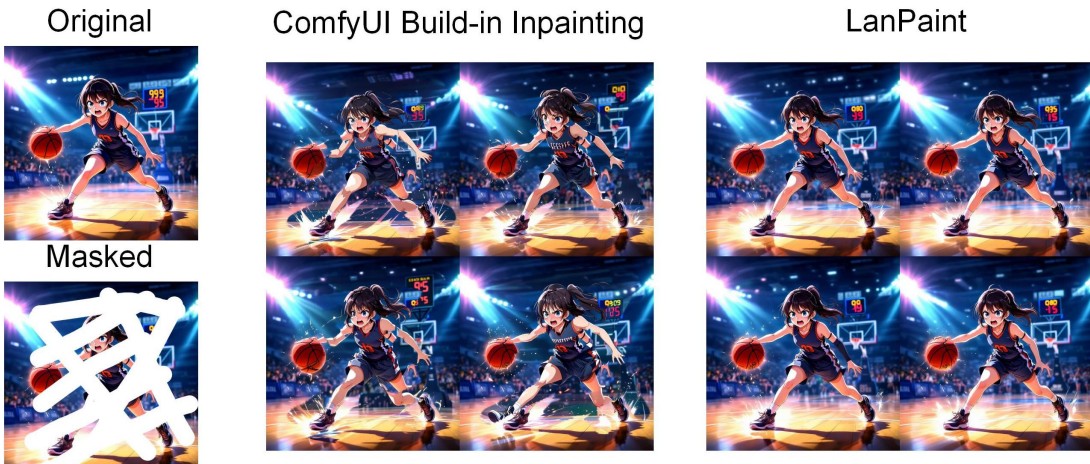

Figure 16: Model: hidream_i1_dev_fp8, Prompt: "An anime-style girl intensely playing basketball, mid-dribble with sweat glistening under the court lights. The scoreboard shows 98-95, highlighting the close match. She wears a sleek jersey and shorts, sneakers gripping the polished floor. Dynamic motion, vibrant colors, ultra-detailed (absurdres), with dramatic lighting and a glowing energy—like a high-stakes anime sports moment.", Steps: 28, CFG Scale: 1.0, Sampler: Euler, Scheduler: Normal, LanPaint Iteration Steps: 5, Seed: 0, Batch Size: 4

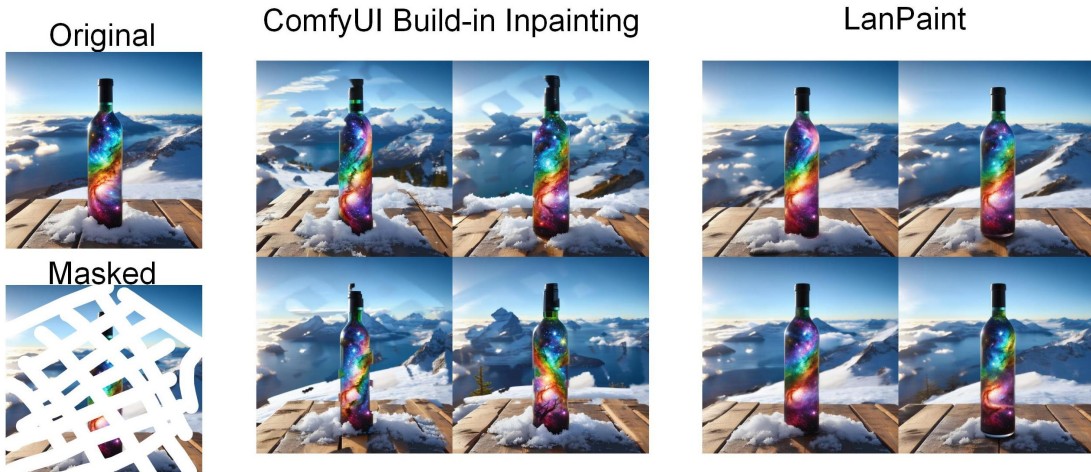

Figure 17: Model: sd3.5_large, Prompt: "a bottle with a rainbow galaxy inside it on top of a wooden table on a snowy mountain top with the ocean and clouds in the background", Steps: 30, CFG Scale: 5.5, Sampler: Euler, Scheduler: sgm_uniform, LanPaint Iteration Steps: 5, Seed: 0, Batch Size: 4

