# OpenReview forum: "LanPaint: Training-Free Diffusion Inpainting with Asymptotically Exact and Fast Conditional Sampling"
_TMLR — Accepted by TMLR_

### Review · Reviewer_mmMZ · 2025-08-08

**Summary Of Contributions:**

This paper presents LanPaint, a novel training-free framework for partial conditional sampling in diffusion models, specifically targeting the inpainting problem. The key contributions lie in two innovations: the Bidirectional Guided (BiG) Score, which introduces a mutual feedback mechanism between the known and inpainted regions to effectively avoid local maxima trapping issues common in fast ODE-based samplers; and the Fast Langevin Dynamics (FLD), an accelerated Langevin sampling method with momentum that converges quickly within just a few inner iterations while rigorously preserving the stationary distribution of the original Langevin dynamics. The authors provide theoretical guarantees and verify them on synthetic Gaussian benchmarks. Extensive experiments on standard datasets like CelebA-HQ and ImageNet demonstrate that LanPaint outperforms existing training-free methods in perceptual and distributional metrics while being more memory efficient. Moreover, LanPaint generalizes seamlessly to modern production-scale diffusion architectures, including Stable Diffusion XL and rectified flow models, proving its practical usability and broad applicability without requiring retraining.

**Audience:**

Yes

**Audience Explanation:**

The work addresses a longstanding challenge of training-free conditional sampling in diffusion models with a theoretically principled and computationally efficient solution. Its broad applicability across various diffusion architectures and the potential for immediate use in real-world systems further increases its appeal.

**Broader Impact Concerns:**

LanPaint makes high-quality image inpainting easier and more accessible, which raises concerns about potential misuse for creating deepfakes or misleading content. The paper should briefly address these risks and suggest mitigation measures like watermarking or provenance tracking.

**Claims And Evidence:**

Yes

**Claims Explanation:**

The paper supports its claims with solid theoretical proofs, which are backed by empirical validation on both synthetic benchmarks—where exact KL divergence to the target distribution is computed—and real-world datasets. Quantitative results on CelebA-HQ and ImageNet using standard metrics such as LPIPS and FID consistently show improvements over strong baselines. Qualitative examples on large-scale diffusion models further illustrate the practical effectiveness of the approach. Overall, the evidence presented is thorough, convincing, and clearly tied to the main contributions.

**Requested Changes:**

- Include high-resolution experiments at 512×512 and 1024×1024 resolutions, testing various mask types (box, free-form, outpainting), and report both perceptual quality metrics (LPIPS/FID) and computational costs to demonstrate scalability.
- Evaluate the method using diverse ODE solvers beyond Euler under comparable step budgets to assess solver-agnostic robustness and efficiency.
- Provide a comprehensive sensitivity analysis of critical hyperparameters, including the guidance scale λ, friction coefficient Γ, inner iteration count, and step size, with clear recommendations for practical settings.
- Conduct a user study with a sufficient number of participants (e.g., 30) performing blind A/B comparisons on challenging mask types to assess subjective perceptual quality and preference.
- It would also be beneficial to provide empirical analysis validating assumptions made in the BiG score approximation, such as measuring the magnitude of the discarded gradient terms during sampling.

---

> ### Author Response · Authors · 2025-08-14
> **Thanks to the Reviewer for Valuable Feedback - I**
>
> We thank the reviewer for feedback on our submission, particularly for the suggestions to validation and broader impact concerns. We appreciate the opportunity to refine our work based on your recommendations. Below is our response to each point:
>
> **C1. Include high-resolution experiments at 512×512 and 1024×1024 resolutions, testing various mask types (box, free-form, outpainting), and report both perceptual quality metrics (LPIPS/FID) and computational costs to demonstrate scalability.**
>
> A1. We understand the concern about scaling image resolution. However, scaling from 256×256 to 512×512 or 1024×1024 increases computational costs by 4× and 8×, respectively, which is prohibitive given our limited budget.
>
> Higher resolutions (e.g., 256×256 to 512×512 or 1024×1024) demands more training data and model parameters, requiring robust backbone scaling in large model training. LanPaint, however, is a model-agnostic sampling algorithm. It operates independently of the backbone architecture and is thus irrelevant to both data and model parameters. Hence, its performance does not rely on architectural scaling. We think testing resolution scaling is expensive but not critical for validating its performance. We kindly ask the reviewer to excuse us for not being able to conduct such validation at the current stage.
>
>
> **C2. Evaluate the method using diverse ODE solvers beyond Euler under comparable step budgets to assess solver-agnostic robustness and efficiency.**
>
> A2. We have added Table 4 (in Appendix A of the revised draft)  to evaluate LanPaint's robustness across different samplers on box mask.
>
> | Method              | CelebA |       |       |       |       |       | ImageNet |       |       |       |       |       |
> |---------------------|--------|-------|-------|-------|-------|-------|-------|-------|-------|-------|-------|-------|
> |                     | Euler  |       | DPM++ |       | DDIM  |       | Euler    |       | DPM++ |       | DDIM  |       |
> |                     | LPIPS  | FID   | LPIPS | FID   | LPIPS | FID   |LPIPS | FID   | LPIPS | FID   | LPIPS | FID   |
> | **Heuristic Methods** |        |       |       |       |       |       |          |       |       |       |       |       |
> | Replace             | 0.131  | 31.7  | 0.119 | 26.7  | 0.130 | 31.5  | 0.229    | 75.7  | 0.234 | 75.1  | 0.228 | 75.6  |
> | CoPaint-2           | 0.180  | 43.6  | 0.186 | 45.8  | 0.169 | 41.1  | 0.234    | 85.1  | 0.233 | 84.5  | 0.297 | 89.1  |
> | CoPaint-3           | 0.172  | 41.7  | 0.181 | 44.3  | 0.163 | 38.9  | 0.228    | 76.9  | 0.228 | 78.9  | 0.288 | 82.1  |
> | DDRM                | 0.128  | 32.4  | 0.135 | 34.8  | 0.127 | 32.2  | 0.216    | 67.2  | 0.215 | 68.9  | 0.211 | 65.3  |
> | MCG                 | 0.130  | 31.6  | 0.113 | 25.9  | 0.124 | 29.7  | 0.225    | 83.5  | 0.266 | 88.1  | 0.253 | 86.7  |
> | DPS                 | 0.181  | 44.1  | 0.166 | 40.2  | 0.179 | 43.7  | 0.252    | 107.5 | 0.248 | 103.0 | 0.249 | 108.2 |
> | **Asymptotically Exact Methods** | |       |       |       |       |       |          |       |       |       |       |       |
> | Repaint-5           | 0.115  | 34.5  | 0.127 | 41.7  | 0.114 | 34.2  | 0.216    | 62.8  | 0.342 | 133.6 | 0.211 | 60.8  |
> | Repaint-10          | 0.112  | 34.6  | 0.409 | 223.3 | 0.111 | 34.4  | 0.215    | 61.0  | 0.604 | 174.5 | 0.211 | 59.7  |
> | TFG-5               | 0.119  | 31.9  | 0.113 | 31.9  | 0.118 | 31.4  | 0.235    | 69.3  | 0.281 | 83.7  | 0.234 | 69.7  |
> | TFG-10              | 0.114  | 33.2  | 0.108 | 33.2  | 0.112 | 31.2  | 0.234    | 66.5  | 0.305 | 89.5  | 0.235 | 65.8  |
> | LanPaint-5 (ours)   | 0.105  | **27.9** | 0.097 | **23.7** | 0.104 | **28.0** | 0.180 | 49.3  | **0.201** | 56.2 | 0.178 | 48.6  |
> | LanPaint-10 (ours)  | **0.103** | 29.5 | **0.097** | 26.9 | **0.103** | 29.3 | **0.171** | **46.4** | 0.205 | **55.1** | **0.172** | **45.6** |
>
> It shows that, although different ODE solvers yield varying performance, LanPaint consistently outperforms other baseline models.
>
> **C3. Provide a comprehensive sensitivity analysis of critical hyperparameters, including the guidance scale λ, friction coefficient Γ, inner iteration count, and step size, with clear recommendations for practical settings.**
>
> A3. Thanks for raising concerns about hyperparameters. We kindly remind the reviewer that we have provided a sensitivity analysis and recommendations in Appendix G (Currently Appendix A in revised draft).

---

> > ### Author Response · Authors · 2025-08-14
> > **Thanks to the Reviewer for Valuable Feedback - II**
> >
> > **C4. Conduct a user study with a sufficient number of participants (e.g., 30) performing blind A/B comparisons on challenging mask types to assess subjective perceptual quality and preference.**
> >
> >
> > A4. We appreciate the reviewer’s suggestion for a rigorous blind A/B user study to evaluate subjective perceptual quality and preference. While a user study could serve as a more convincing validation than metrics, it requires funding for participant recruitment and obtainning the Institutional Review Board (IRB) approval. Due to current budget and manpower constraints, we are unable to conduct such a study at this stage, but we plan to pursue it in future work.
> >
> > Instead, we wish to mention that LanPaint, as an open-source project, has received positive feedback from its user community. We have engaged diverse users on GitHub and iterated the algorithm multiple times based on their feedback regarding image quality. We hope this provides the reviewer with some confidence in LanPaint’s perceptual quality.
> >
> > **C5. It would also be beneficial to provide empirical analysis validating assumptions made in the BiG score approximation, such as measuring the magnitude of the discarded gradient terms during sampling.**
> >
> > A5. We have now plotted the magnitude of the discarded gradient terms during sampling in Fig.8, appendix F. The figure demonstrates that the discarded gradient term is indeed scale as a small quantity of $\mathcal{O}(\sqrt{1 - \bar{\alpha}_t})$.
> >
> > **C6. Broader Impact Concerns: LanPaint makes high-quality image inpainting easier and more accessible, which raises concerns about potential misuse for creating deepfakes or misleading content. The paper should briefly address these risks and suggest mitigation measures like watermarking or provenance tracking.**
> >
> > A6. We have now added a broader impact section addressing misuses.

---

> ### Author Response · Authors · 2025-08-22
>
> Dear Reviewer,
>
> Thank you again for your valuable feedback. I wonder if our response has addressed your concerns? Looking forward to your reply.
>
> Best regards,
>
> The Authors

---

> > ### Comment · Reviewer_mmMZ · 2025-08-23
> > **Dear Authors**
> >
> > Thank you for your detailed response and the additional experiments. Most of my concerns have been addressed.

---

### Review · Reviewer_bWWV · 2025-08-08

**Summary Of Contributions:**

The authors present an inpainting method for ODE-based diffusion that is training-free and asymptotically exact. They focus on partial conditioning sampling to preserve the exact values of known pixels. It consists of two main contributions: (1) The bi-directional guided score ("BiG") which "loosens" the constraint by exchanging information between both the known and inpainted pixels during generation, in order to make generation asymptotically exact and have the final result match the true posterior distribution given $y_0$. It also helps the generation escape local likelihood maximas; (2) Fast Langevin dynamics, a "stabilized" version of underdamped Langevin dynamics that allows for faster generation with less inner and diffusion steps. They demonstrate these advantages, i.e., asymptotically exact conditioning, reduced cost, general applicability, and local maxima escape, on diverse experiments, which include synthetic Gaussian data, masked versions of popular image datasets, and usage on large well-known models such as StableDiffusion-XL.

### Strengths
- The methodology is clear, theoretically grounded and well-motivated.
- The authors say they have implemented their method into the popular ComfyUI software, which displays great reproducibility efforts.
- The targeted problems are quite important and have wide applications.
- The method can be used with a wide variety of models, which further strengthens its use case as a training-free conditioning technique.
- I appreciate the range of experiments, both synthetic data for demonstrating their theoretical claims as well as more concrete use cases on real large-scale models. The ablation study also highlights well the method.
### Weaknesses

- I think presentation could be strengthened.
	- There are some typos and inconsistencies in the paper. I think it requires better proofreading before publishing.
	- The structure lacks a bit of connection and coherence. Maybe Sections 2 and 5 could be merged, and 5 could better highlight summary results, paths for future work, more detailed limitations, parallel lines of work, etc.
	- Table and figure placement could be improved.
- The limitation section is too short in my opinion, as stated above. Do the authors have recommendations for future work to adapt this method for distilled models, as reduced cost is part of the targeted issues?
- Although the claim for asymptotically exact generation is well-supported theoretically, I think more recent methods that adopt a different approach could be good benchmark candidates. For example, moment-matching posterior [1] sampling can be used in a training-free manner to produce high-quality posterior samples. I understand your discussion that those methods do not exactly preserve the known parts, but as far as I understand it, especially in a discrete real setting, the BiG score kind of relaxes this constraint and allows $y$ to change during generation. Also, contrary to these methods, LanPaint and similar methods consider no uncertainty on the known parts, which can be an issue in some posterior sampling settings, and could maybe be discussed.
- Therefore, the claim for asymptotical exactness might be a bit too strong considering the possible drifts in practice? Did you observe that, especially on real-world images? Do you expect your method to be used in more diverse domains?

[1] : Rozet, F., Andry, G., Lanusse, F., & Louppe, G. (2024). Learning diffusion priors from observations by expectation maximization. _Advances in Neural Information Processing Systems_

**Additional Comments:**

Here are some of the typos I found during my readings:
- Abstract: exact partial conditional sampling methods **(with s)** for ODE-based
- End of introduction, 5th line before the end: you write in-painting while you do not use a hyphen in the rest of the work.
- End intro: missing "and" in last paragraph
- 2.1: "is a sampling technique", no s.
- End of 2.2 : "also fall" instead of fail.
- Title of 2.3: "InPainting", why is P uppercase?
- End of page 3: "The only problem **that** remains" (missing that)
- end of page 3: approximately sample (with no s)
- First line of page 4: "but fails to ensure" fails with s.
- End of page 4, "at t = 0" twice
- Page 5, right before "Theorem 3.1": duplicate "the".
- BiG sometimes written BIG? End of page 4
	- 4.4 : "bidirectional generative" instead of guided for BiG also.
	- Please be consistent with your acronyms.
- Beginning page 5: "then discard**ing**"
- Many occurrences of "equation" without uppercase E and without a reference to the equation. Please be consistent.
- Page 9, first paragraph: "qualitative" instead of "quanlititive"
- Table 3 caption: "share similar trends" → should be "shares similar trends".
- 4.5: "build in ComfyUI" -> "built in ComfyUI"
- Title of section 5.1: "Recitified Flow" -> "Rectified Flow"
- Other inconsistent hyphenation: training free, training-free, in-painting vs inpainting, etc.

I definitely could not write all of them, so please proofread carefully.

**Audience:**

Yes

**Audience Explanation:**

Generative modeling, inpainting, and overall conditioning on masked areas are extremely important topics of clear interest to the community, both for image generation as highlighted by the experiments here, but also in areas like scientific generative modeling. I think training-free methods are a particularly important topic for diffusion models, considering the numerous use cases, such as building upon large foundation models. The targeted issues, efficiency, exactly matching (asymptotically) the distribution, and local minima, are also quite important to tackle. The method is well theoretically grounded, has public implementation in ComfyUI as mentioned by the authors in the paper, and its advantages are well demonstrated throughout the experiments.

**Broader Impact Concerns:**

Like all generative modeling, especially inpainting applied to real images like in this case with popular tools, there are large societal impacts, both negative and positive. I think it is important to at least mention the impact and refer to ethical resources, such as [2] and [3]. Especially as this method is targeted toward efficient generation, which poses more risks for spreading misinformation, deepfakes, etc. I think for example the last figure of the appendix might already be borderline, especially to include in a paper, and clearly highlights some risks of such methods. If not removed, it should maybe be part of a small discussion about this.

[2] Emily Denton. Ethical considerations of generative ai. AI for Content Creation Workshop, CVPR, 2021. 9

[3] Franks, Mary Anne, and Ari Ezra Waldman. "Sex, lies, and videotape: Deep fakes and free speech delusions." Md. L. Rev. 78 (2018): 892.

**Claims And Evidence:**

Yes

**Claims Explanation:**

The major claims of the paper are in my opinion well-supported by the experiments and evidence. The synthetic Gaussian experiments demonstrate the sampling method can reach good results, both qualitatively and quantitatively in a few iterations and escape local maximas. Qualitative and quantitative results on CelebA and ImageNet support the claims for more complex datasets, providing consistent and plausible samples as well as low FID and LPIPS scores. Finally, I think the examples on large-scale pre-trained models truly strengthens the claims by showing that it can be applied on top of any popular large-scale model as a real-world use case. Overall, the method outperforms the inpainting baselines on these different benchmarks, and also reduces computational cost compared to most candidates. The ablation study also supports the importance of both components, the BiG score and Fast Langevin Dynamics.

**Requested Changes:**

Beside answering the questions above, here are a few more precise changes. In my opinion, most presentation changes are required for acceptance.
### Presentation
I think a better presentation would strengthen the paper and its reading.
- Please proofread thoroughly the paper. There are many typos and inconsistencies. I noted some I found below in the additional comments.
- As mentioned in the weaknesses, the structure is a bit hard-to-follow in my opinion, and the end feels like it lacks something:
	- Related works at the end is nice, but feels quite disconnected. I think it should be, and limitations too, part of a Discussion section.
	- A discussion section with a summary, future work recommendations, deeper discussions around limitations, use cases, and more would be very much appreciated to end the paper.
	- Is 5.1 really needed in the main text?
- Pay attention to references to figures in the text. For example, in Section 4.1, second paragraph, "the image" is a bit confusing. Also, be consistent between the use of "Equation ..." vs "Eq. ...", same for Figure vs Fig.
- Maybe the position of the figures could be improved a bit, but this is more minor.
### Other
- I think the discussion and opposition with "inexact" methods could be a bit different, as mentioned in the weaknesses above. I appreciate you still included it as "supplementary reference" but I think it can truly serve as a true benchmark and more recent comparisons could strengthen the paper.
- Could your method effectively consider uncertainty over known regions? If not, please include it in your limitations, even if you consider it to be a basic assumption considering the exact match requirement. If yes, please also discuss it.
- If possible, but this is not required at all, as mentioned above, comparisons with methods such as MMPS would be interesting. This also concerns quantitative results, which might also be interesting with higher numbers of steps for methods that require it.

---

> ### Author Response · Authors · 2025-08-14
> **Thanks to the Reviewer for Insightful Feedback - I**
>
> We thank the reviewer for detailed review of our submission and for the discussion on asymptotic exactness, which has greatly helped us refine our work. We have addressed your suggestions and clarified misleading phrases that caused misunderstandings. Below is our response to each point:
>
> ## Weaknesses
>
> **W1. The structure lacks a bit of connection and coherence. Maybe Sections 2 and 5 could be merged, and 5 could better highlight summary results, paths for future work, more detailed limitations, parallel lines of work, etc. Table and figure placement could be improved.**
>
> A1. Apologies for lacking connection and coherence. Section 5 was initially placed before Section 2 in our first draft but was later moved to the end of the paper. We now have moved it back to improve the flow. We have also improved the table and image placement.
>
>
> **W2. The limitation section is too short in my opinion, as stated above. Do the authors have recommendations for future work to adapt this method for distilled models, as reduced cost is part of the targeted issues?**
>
> A2. Thank you for your suggestion. We have further addressed distillation and cost reduction in the limitation section.
>
> **W3. Although the claim for asymptotically exact generation is well-supported theoretically, I think more recent methods that adopt a different approach could be good benchmark candidates. For example, moment-matching posterior sampling can be used in a training-free manner to produce high-quality posterior samples. I understand your discussion that those methods do not exactly preserve the known parts, but as far as I understand it, especially in a discrete real setting, the BiG score kind of relaxes this constraint and allows y to change during generation.**
>
> A3. The comment that "those methods do not exactly preserve the known parts" may reflect a possible misunderstanding. We wish to clarify that the difference between heuristic methods and asymptotic exact methods lies in their ability to "preserve the joint distribution between known and unknown parts," not only in "preserving the known parts." We have revised the introduction to address this at the end of the third paragraph: "The posterior $ q(\mathbf{z}|\mathbf{y}) $ is a heuristic approximation that aims to construct a visually plausible $\mathbf{z} = (\mathbf{x}, \mathbf{y})$ without requiring $\mathbf{z}$ to follow exactly the joint distribution $ p(\mathbf{z}) $ modeled by the pretrained diffusion model." The distinction between heuristic and asymptotically exact methods is also evident in the conditional Gaussian experiment (Section 5.1, previously Section 4.1), where asymptotically exact methods can achieve zero KL divergence, whereas heuristic methods cannot.
>
> The comment mentions "relaxes this constraint"; we assume this refers to the constraint that the sampled known part $\mathbf{y}\_0$ should equal the observed known part $\mathbf{y}\_o$. We apologize for the unclear terminology: our use of 'relax' before equation 11 mistakenly suggested the BiG score imposes a weaker constraint. We have revised the sentence from "we observe that equation 7 can be relaxed to (equation 11 and BiG score)" to "we observe that equation 7 is a special case of the following equivalent but more general form (equation 11 and BiG score)". In fact, equation 11 is a stronger constraint compared to equation 7 for $\lambda > 0$. Therefore the BiG score strengthens the constraint $\mathbf{y}\_0 = \mathbf{y}\_o$ instead of weaken it. Although the BiG score allows $\mathbf{y}\_{t>0}$ to change during generation, it ensures $\mathbf{y}\_0$ exactly matches the observed $\mathbf{y}\_o$ (see discussion below Equation 11 about $p_0(\mathbf{y} \mid \mathbf{y}\_o) = \delta(\mathbf{y} - \mathbf{y}\_o)$, hence $\mathbf{y}_0 = \mathbf{y}\_o$ no matter what value $\lambda>0$ takes). Thus, the asymptotic exactness remains intact.
>
>
>
>
> **W4. Also, contrary to these methods, LanPaint and similar methods consider no uncertainty on the known parts, which can be an issue in some posterior sampling settings, and could maybe be discussed.**
>
> A4. Thank you for your suggestion! We wish to point out that general posteior sampling is not the current aim of LanPaint. We have further addressed uncertainty in the revised Limitation and Future work section.

---

> > ### Author Response · Authors · 2025-08-14
> > **Thanks to the Reviewer for Insightful Feedback - II**
> >
> > **W5. Therefore, the claim for asymptotical exactness might be a bit too strong considering the possible drifts in practice? Did you observe that, especially on real-world images?**
> >
> >
> > A5. This doubt about exactness still questions whether the constraint $\mathbf{y}_0 = \mathbf{y}_o$ is satisfied. This concern stems from our misleading use of the term "relax," which we clarified in A3. In fact, the BIG score strengthens, rather than relaxes, the constraint. Thus, the constraint $\mathbf{y}_0 = \mathbf{y}_o$ and the claim of asymptotic exactness $\mathbf{x}_0 \sim p_0(\mathbf{x} \mid \mathbf{y}_o)$ in Theorem 4.1 remain valid.
> >
> >
> > In real-world applications, the constraint $\mathbf{y}_0 = \mathbf{y}_o$ is critical, as any deviation can accumulate over multiple inpainting rounds, degrading image quality. In practice, users must copy and paste the known part of the original image, $\mathbf{y}_o$, onto the inpainted image, overwriting the sampled known part $\mathbf{y}_0$. Failure to enforce $\mathbf{y}_0 = \mathbf{y}_o$ results in noticeable discontinuities or unnatural seams. With LanPaint, this copy-and-paste process is built in, with no discontinuities or seams are observed, indicating that deviations from the constraint $\mathbf{y}_0 = \mathbf{y}_o$ are not observed in practice.
> >
> >
> > **W6. Do you expect your method to be used in more diverse domains?**
> >
> > A6. Yes, LanPaint is a general conditional sampling method. It is not limited to image but can be extended to diverse domains, such as audio, video, and scientific data. We have now discussed it in the revised limitation section.
> >
> >
> > ## Requested Changes
> >
> > ### Presentation
> >
> > **CP1. I think a better presentation would strengthen the paper and its reading**
> >
> > - Please proofread thoroughly the paper. There are many typos and inconsistencies. I noted some I found below in the additional comments ...
> >
> > AP1. Thank you for the suggestion! We have moved the related works section to the front for better flow and condensed section 5.1 (now 2.1 in the revised version). We have expanded the limitations into a dedicated section on limitations and future work. Other formatting issues and typographical errors have been corrected.
> >
> >
> > ### Other
> >
> > **CO1. I think the discussion and opposition with "inexact" methods could be a bit different, as mentioned in the weaknesses above. I appreciate you still included it as "supplementary reference" but I think it can truly serve as a true benchmark and more recent comparisons could strengthen the paper.**
> >
> > AO1. Thank you for your appreciation. We prefer the term "heuristic" methods over "inexact" methods, as they address distinct problems rather than implying inferiority.
> >
> > Heuristic methods, such as DPS, tackle ill-posed inverse problems. Such problems could also be tackled with GANs or other models instead of diffusion models. Heuristic methods typically do not emphasize exactness in the joint distribution, as mentioned in A3. In other words, heuristic methods aim to obtain a visually plausible solution instead of a Monte Carlo sample that follows exactly the joint distribution modeled by the pretrained diffusion model.
> >
> > In contrast, LanPaint focuses on conditional sampling given a joint distribution modeled by a pretrained diffusion model, which is a well-posed problem. The output of LanPaint is an asymptotically exact Monte Carlo sample of the joint distribution. Therefore, they are fundamentally different from each other in purpose.
> >
> > A critical limitation of heuristic methods in practice is their reliance on a loss and corresponding gradient, which requires 2–4 times more GPU memory (also demonstrated in Tabs. 1 and 2) than gradient-free inference like LanPaint. This makes them nearly impractical for models like Stable Diffusion or Flux, which already consume most GPU memory during model loading. This issue persists in more recent methods as long as they require gradients.
> >
> > Moreover, heuristic methods usually adopt stochastic sampling. Comparing LanPaint with heuristic methods would require adapting them to ODE sampling, which is time-consuming and sometimes infeasible. Such adaptations may also compromise the performance of heuristic methods, as they are not designed for ODE sampling. We kindly ask the reviewer to understand our inability to compare LanPaint with a complete list of heuristic methods.
> >
> > **CO2. Could your method effectively consider uncertainty over known regions? If not, please include it in your limitations, even if you consider it to be a basic assumption considering the exact match requirement. If yes, please also discuss it.**
> >
> > AO2. We have not considered uncertainty in known regions, as this paper focuses on conditional inference: given $p(\mathbf{x}, \mathbf{y})$, how to infer from $p(\mathbf{x}|\mathbf{y})$. In future work, uncertainty can be addressed by modifying the current form of $p(\mathbf{y}_t | \mathbf{y}_0)$ appropriately. We have included this in the revised future work section.

---

> > > ### Author Response · Authors · 2025-08-14
> > > **Thanks to the Reviewer for Insightful Feedback - III**
> > >
> > > **CO3. If possible, but this is not required at all, as mentioned above, comparisons with methods such as MMPS would be interesting. This also concerns quantitative results, which might also be interesting with higher numbers of steps for methods that require it.**
> > >
> > >
> > > AO3. We acknowledge that MMPS, a heuristic method we initially missed, significantly enhances posterior inference by incorporating EM, addressing DPS’s instability and support cover issues. We have now included it in the related work section. In response to the reviewer's interest, we tested the model on box masks using the 20 steps Euler sampler with parameters aligned with the FFHQ box inpainting setup described in the paper of MMPS. The results are presented below:
> > >
> > >
> > > | Method             | CelebA-HQ-256  |        |        |         | ImageNet  |        |        |         |
> > > |--------------------|---------------------|--------|--------|---------|----------------|--------|--------|---------|
> > > |                    | LPIPS               | FID    | Time (s/image) | MemOver (MB/image) | LPIPS  | FID    | Time (s/image) | MemOver (MB/image) |
> > > | **Heuristic Methods** |                     |        |        |         |                |        |        |         |
> > > | Replace            | 0.131               | 31.7   | 0.3    | 81      | 0.229          | 75.7   | 1.9    | 581     |
> > > | CoPaint-2          | 0.180               | 43.6   | 1.7    | 248     | 0.234          | 85.1   | 15.7   | 5444    |
> > > | CoPaint-3          | 0.172               | 41.7   | 2.5    | 248     | 0.228          | 76.9   | 22.4   | 5445    |
> > > | DDRM               | 0.128               | 32.4   | 0.3    | 81      | 0.216          | 67.2   | 1.9    | 583     |
> > > | MCG                | 0.130               | 31.6   | 0.8    | 248     | 0.225          | 83.5   | 6.4    | 5445    |
> > > | DPS                | 0.181               | 44.1   | 0.8    | 247     | 0.252          | 107.5  | 6.4    | 5440    |
> > > | MMPS               | 0.117               | **27.6**   | 1.1    | 298     | 0.236          | 88.0   | 7.2    | 6340    |
> > > | **Asymptotically Exact Methods** |         |        |        |         |                |        |        |         |
> > > | Repaint-Euler-5    | 0.115               | 34.5   | 1.4    | 81      | 0.216          | 62.8   | 11.8   | 581     |
> > > | Repaint-Euler-10   | 0.112               | 34.6   | 2.6    | 81      | 0.215          | 61.0   | 20.5   | 581     |
> > > | TFG-5              | 0.119               | 31.9   | 1.5    | 81      | 0.235          | 69.3   | 11.9   | 595     |
> > > | TFG-10             | 0.114               | 33.2   | 2.6    | 81      | 0.234          | 66.5   | 21.7   | 595     |
> > > | LanPaint-5         | 0.105               | 27.9   | 1.6    | 81      | 0.180          | 49.3   | 11.3   | 599     |
> > > | LanPaint-10        | **0.103**           | 29.5 | 2.9  | 81      | **0.171**      | **46.4** | 20.8 | 599     |
> > >
> > > We observed that MMPS significantly outperforms MCG and DPS, achieving the best FID on CelebA. However, its performance degrades on ImageNet, where DDRM performs the best among heuristic methods. We observed that MMPS incurs the highest GPU memory overhead, a common bottleneck that limits the practical implementation of heuristic methods requiring gradients. In comparison, LanPaint is more robust, consistently achieving the best LPIPS across CelebA and ImageNet with significantly lower  memory overhead.
> > >
> > >
> > > Regarding number of steps, 20–30 steps using ODE fast samplers are standard in practice, taking approximately 0.5 (i.e Stable Diffusion 1.5) to 5 minutes (i.e Qwen Image) depending on model size on a high end GPU. While higher step counts (e.g., above 50) are academically interesting, they are often impractical. That is why we focusing only on ODE fast samplers.
> > >
> > > **CO4. Broader Impact Concerns: Like all generative modeling, especially inpainting applied to real images like in this case with popular tools, there are large societal impacts, both negative and positive. I think it is important to at least mention the impact and refer to ethical resources, such as [2] and [3]. Especially as this method is targeted toward efficient generation, which poses more risks for spreading misinformation, deepfakes, etc. I think for example the last figure of the appendix might already be borderline, especially to include in a paper, and clearly highlights some risks of such methods. If not removed, it should maybe be part of a small discussion about this.**
> > >
> > > AO4. Thanks for the suggestion! We have now added a broader impact section addressing misuses and removed the last figure of the appendix.

---

> ### Author Response · Authors · 2025-08-22
>
> Dear Reviewer,
>
> Thank you again for your valuable feedback. I wonder if our response has addressed your concerns? Looking forward to your reply.
>
> Best regards,
>
> The Authors

---

> > ### Comment · Reviewer_bWWV · 2025-08-23
> > **Follow-up answer to my review**
> >
> > Dear authors,
> >
> > Thank you for your detailed replies to all of my concerns. Most of them have been addressed and I really appreciate both the more precise explanations as well as the additional experiments you ran. I do not have further major comments to make and consider the paper can be accepted with the revisions.
> >
> > Best regards.

---

### Review · Reviewer_TfeF · 2025-08-09

**Summary Of Contributions:**

This paper proposes a training-free algorithm to inpaint/outpaint an image with diffusion. The method (coined LanPaint) extends the Langevin dynamics Monte Carlo approaches, introducing two innovations: the BiG score and the fast Langevin dynamics (FLD). The BiG score is designed to prevent the Langevin dynamics from falling in local minima by introducing a mechanism to incorporate the feedback of inpainted pixels to correct the posterior for the seen pixels. The FLD is a variant of underdamped LD that can use a larger step size.

**Additional Comments:**

- Please also fix the grammatical error (if I am correct): In the second paragraph of the introduction, there is a phrase "as iteratively denoise and renoise" which I believe should be "as iteratively denoising and renoising."
- Also, please fix many \citet which should be \citep instead.
- In table 5, the order of Repaint-Euler-10 and Repaint-Euler-5 should be the opposite?

**Audience:**

Yes

**Audience Explanation:**

The problem this paper addresses---inpainting with diffusion models---is practically important. I am not sure if the Langevin dynamics approach will be the dominant in the near future, but it is definitely relevant and many people will be interested in improving and refining the approach.

**Broader Impact Concerns:**

I do not have any concerns.

**Claims And Evidence:**

Yes

**Claims Explanation:**

The claims are mostly backed up by empirical results, except for the theoretical claim on the asymptotic exactness of the conditional sampler (Thm 3.1) and the stationarity (Thm 3.2).

The empirical claims are backed-up well through both experiments on both synthetic and realistic data, with a concrete ablation study. While I believe that some of the claims should be narrowed down and backed up slightly more rigorously, I find that the submission to contain mostly satisfactory evidences.

**Requested Changes:**

- One thing that is not very clear to me is whether the authors claim the FLD to be a generally better variant of LD (or ULD). From what is stated in section 3.2, it seems like that FLD indeed is a better version, as none of the arguments seem to pertain to the task of diffusion-based inpainting. If this is what authors intend to argue (which is a bold claim), please be a little bit more straightforward and empirically prove this point more rigorously. If not, please explain why this is a specialized solution for the task at hand.
- Related to the prior point, the effectiveness of the FLD has been somewhat under-validated in the ablation studies. In particular, the table 3 shows that FLD works as intended for ImageNet, but not for CelebA. If a method works on only one dataset out of two, I am not really sure if it is indeed working. I request the authors to add back up this point, perhaps by adding a new dataset or testing with a finer grid of step sizes.
- Also, I wonder how the hyperparameters have been tuned for both the proposed LanPaint and the baselines (esp. the step sizes). LanPaint has additional hyperparameters, so this point should be clarified to ensure the fairness of the comparison.
- Authors say that eq.11 is a "relaxation" of eq.7, but I fail to see why it is technically a relaxation. Relaxation (as I know it) usually refers to a looser lower or upper bound which is usually easier to handle. However, as both equations are written with a hand-wavy \approx, I am not should which one is looser.
- As a minor comment, the manuscript was quite difficult to read for a person who reads on a laptop, because the locations where the figures are mentioned and where the figures actually appear at was quite far away from each other. For instance, Figure 3 has been referred in page 4, but appears in page 7. I recommend changing the orders and locations accordingly.
- As another minor suggestion, the experiments sections will be easier to read if the authors could give a brief overview of the baselines they consider, and how they match to the categories of inpainting methods described in the introduction.
- Finally, there are some typos and mistakes. Please see "additional comments."

---

> ### Author Response · Authors · 2025-08-14
> **Thanks to the Reviewer for Thorough Feedback - I**
>
> We thank the reviewer for effort to thoroughly understand our method, which has greatly assisted us in clarifying points that were not clearly articulated previously. We have addressed your questions and revised misleading phrases that caused confusion and misunderstandings. Below is our response to each point:
>
> **C1. One thing that is not very clear to me is whether the authors claim the FLD to be a generally better variant of LD...please explain why this is a specialized solution for the task at hand.**
>
>
> A1. We have already clarified FLD’s role as a specialized solution for diffusion models in the paragraph beginning "To understand how decomposition enhances stability" (Section 3.2 in the original text, now Section 4.2 in the revised draft). However, the placement of Fig. 2 may have disrupted the reading experience. We apologize for the figure placement, which may have caused reading difficulties for the reviewer.
>
>
> The key to FLD’s adaptation for diffusion sampling lies in the score decomposition $\mathbf{s} = \mathbf{C} - A \mathbf{z}$, where the term $-A \mathbf{z}$ acts as a regularizer, enabling larger time steps $\Delta \tau$ to accelerate convergence without instability. Specifically, we choose $A = (1 - \bar{\alpha}\_t)^{-1}$ to ensure that the asymptotic behavior of Langevin dynamics as $\Delta \tau \to \infty$ matches the forward diffusion process (Equation 2):
> $$
> \lim_{\Delta \tau \to \infty} \mathbf{z}_{\tau + \Delta \tau} \sim \mathcal{N}\left(\sqrt{\bar{\alpha}\_t} \hat{\mathbf{z}}\_0, 1 - \bar{\alpha}\_t\right),
> $$
> where $t$ is diffusion time and $\tau$ is Langevin time. In contrast, both LD and ULD treat $\mathbf{s}$ as constant, so $\Delta \tau \to \infty$ leads to exploding results. To emphasize this, we have named this decomposition as **diffusion damping force** in the revised Sec 4.2.
>
> We also wish to mention that a detailed discussion of the problems with LD and ULD in diffusion sampling, and how we address them, is provided in Appendix F (Appendix G in the revised ver), as referenced in Section 3.2 (Section 4.2 in the revised ver).
>
> **C2. Related to the prior point, the effectiveness of the FLD has been somewhat under-validated in the ablation studies...I request the authors to add back up this point, perhaps by adding a new dataset or testing with a finer grid of step sizes.**
>
> A2. Following the previous response, a key improvement of FLD is its ability to support larger step sizes. However, as stated in the main text, "for the CelebA-HQ-256 dataset, performance metrics exhibit low sensitivity to step size variations." This explains why FLD has a limited impact on CelebA performance.
>
> In response to the reviewer’s request, we have updated the ablation results in Table 3 with an expanded grid of step sizes. Note that we tested both CelebA and ImageNet using the **same** set of parameters to assess generalization ability. Therefore, *the evaluation of LanPaint and its components should also account for both datasets.*
>
> | Method                    | Step Size 0.02 |       | Step Size 0.05 |       | Step Size 0.1 |       | Step Size 0.15 |       | Step Size 0.2 |       |
> |---------------------------|----------------|-------|----------------|-------|---------------|-------|----------------|-------|---------------|-------|
> | **CelebA-HQ-256**        | **LPIPS** | **FID** | **LPIPS** | **FID** | **LPIPS** | **FID** | **LPIPS** | **FID** | **LPIPS** | **FID** |
> | None (Langevin)           | 0.121     | 28.9  | 0.115     | 28.4  | 0.111     | 29.2  | 0.114     | 30.9  | 0.108     | 30.1  |
> | + BiG score               | 0.110     | 26.1 | 0.104 | 26.5  | 0.102 | 28.5  | 0.103     | 28.7  | 0.103     | 30.0  |
> | + FLD                     | 0.121     | 28.6  | 0.115     | 28.9  | 0.112     | 29.9  | 0.116     | 31.7  | 0.109     | 30.5  |
> | + (BiG score + FLD)       | 0.111     | 26.7  | 0.105     | 26.6  | 0.103     | 28.5  | 0.103     | 29.5  | 0.103     | 30.2  |
> | **ImageNet**              |           |       |           |       |           |       |           |       |           |       |
> | None (Langevin)           | 0.220     | 66.7  | 0.223     | 65.1  | 0.314     | 81.6  | 0.441     | 125.9 | 0.475     | 141.1 |
> | + BiG score               | 0.205     | 58.6  | 0.213     | 58.0  | 0.303     | 76.0  | 0.431     | 121.4 | 0.474     | 140.0 |
> | + FLD                     | 0.217     | 68.3  | 0.205     | 60.7  | 0.195     | 56.9  | 0.188     | 54.4  | 0.181     | 51.4  |
> | + (BiG score + FLD)       | 0.201     | 58.9  | 0.190     | 52.7  | 0.179     | 48.1  | 0.171     | 46.4  | 0.167 | 45.1 |
>
> As shown in the revised Table 3, FLD does not degrade performance on CelebA (a difference in LPIPS of 0.001) and significantly enhances results on ImageNet (a difference in LPIPS of 0.03). Moreover, applying FLD on ImageNet at a step size of 0.2 further improves results, even surpassing those reported in Table 2. This demonstrates that FLD is a valuable component of LanPaint with a net benefit.

---

> > ### Author Response · Authors · 2025-08-14
> > **Thanks to the Reviewer for Thorough Feedback - II**
> >
> > **C3. Also, I wonder how the hyperparameters have been tuned for both the proposed LanPaint and the baselines (esp. the step sizes). LanPaint has additional hyperparameters, so this point should be clarified to ensure the fairness of the comparison.**
> >
> > A3. We have described it in appendix G and mentioned it in the main text (first paragraph of the section "Latent and Pixel Space Model"). LanPaint has additional hyperparameters, which is tuned on a separate validation set of 100 image. As for other baselines, we follow their recommended hyperparameters. We emphasize that all methods use a single set of hyperparameters across both CelebA and ImageNet and all mask types to compare their generalization ability.
> >
> >
> >
> > **C4. Authors say that eq.11 is a "relaxation" of eq.7, but I fail to see why it is technically a relaxation. Relaxation (as I know it) usually refers to a looser lower or upper bound which is usually easier to handle. However, as both equations are written with a hand-wavy \approx, I am not should which one is looser.**
> >
> > A4. We apologize for the misleading term "relax." We have revised the sentence from "we observe that equation 7 can be relaxed to (equation 11 and BiG score)" to "we observe that equation 7 is a special case of the following equivalent but more general form (equation 11 and BiG score)". Equation 11 offers a more flexible and general form of Equation 7 without loosen any constraint. For $\lambda > 0$, equation 11 is a stronger constraint than equation 7.
> >
> > In addition, we wish to point out that the $\approx$ in both equations becomes $=$ at $t = 0$, as explained in the text immediately following them. This ensures that both Equation 7 and Equation 11 have the capability to sample the conditional distribution $p(\mathbf{x} | \mathbf{y})$.
> >
> >
> > **C5. As a minor comment, the manuscript was quite difficult to read for a person who reads on a laptop, because the locations where the figures are mentioned and where the figures actually appear at was quite far away from each other. For instance, Figure 3 has been referred in page 4, but appears in page 7. I recommend changing the orders and locations accordingly.**
> >
> > A5. We apologize again for the figure placement. We have now rearranged the figure placement accordingly.
> >
> >
> > **C6. As another minor suggestion, the experiments sections will be easier to read if the authors could give a brief overview of the baselines they consider, and how they match to the categories of inpainting methods described in the introduction.**
> >
> > A5. We apologize for the lack of an overview. The baselines we consider and how they correspond to the categories were discussed in the "Related Works", section 5 at the end of the paper. We have now moved this section to section 2 of the paper to improve the paper's flow.

---

> ### Author Response · Authors · 2025-08-22
>
> Dear Reviewer,
>
> Thank you again for your valuable feedback. I wonder if our response has addressed your concerns? Looking forward to your reply.
>
> Best regards,
>
> The Authors

---

### Decision · Action_Editor_9jU4 · 2025-09-10

**Recommendation:** Accept as is

**Additional Comments:**

**Reviews**
The reviews present a range of perspectives. One reviewer expressed concern about the lack of strong evidence that FLD consistently improves over LD, while another commended the authors for thoroughly addressing feedback, adding new experiments, and strengthening the technical soundness of the work. A third reviewer viewed the contribution as incremental but nonetheless solid, practically relevant, and of clear interest to the community.

**Outcome**
Balancing these views, I find that the paper is technically correct, timely, and relevant, with meaningful contributions despite some limitations in generalizability. Therefore, I am pleased to recommend acceptance.

**Audience:**

Yes

**Audience Explanation:**

As mentioned by one of the reviewers: *I think this is clearly of strong interest to the audience, as generative AI and in-painting is very popular, and the integration into a popular framework will be appreciated by a large audience.*

I cannot agree more, I find the paper interesting to the GenAI audience.

**Claims And Evidence:**

Yes

**Claims Explanation:**

**Claims**
The paper makes a clear claim that LanPaint offers a significant methodological advance, namely, it is the first training-free, asymptotically exact, and efficient conditional sampler compatible with fast ODE-based diffusion models.

**Evidence** The evidence is rather broad and convincing, spanning theory, synthetic tests, real datasets, large-scale production models, and ablation studies. As a result, the claims are rather generally well-supported. While some limitations remain (distillation, noisy data, scalability), the overall contribution is strong, timely, and practically impactful.